# Smoking changes adaptive immunity with persistent effects

Violaine Saint-André[1,2 ✉], Bruno Charbit[3], Anne Biton[2], Vincent Rouilly[4], Céline Possémé[1], Anthony Bertrand[1,5], Maxime Rotival[6], Jacob Bergstedt[6,7,8], Etienne Patin[6], Matthew L. Albert[9], Lluis Quintana-Murci[6,10], Darragh Duffy[1,3 ✉] & The Milieu Intérieur Consortium*

Individuals differ widely in their immune responses, with age, sex and genetic factors having major roles in this inherent variability[1–6]. However, the variables that drive such differences in cytokine secretion—a crucial component of the host response to immune challenges—remain poorly defined. Here we investigated 136 variables and identified smoking, cytomegalovirus latent infection and body mass index as major contributors to variability in cytokine response, with effects of comparable magnitudes with age, sex and genetics. We find that smoking influences both innate and adaptive immune responses. Notably, its effect on innate responses is quickly lost after smoking cessation and is specifically associated with plasma levels of CEACAM6, whereas its effect on adaptive responses persists long after individuals quit smoking and is associated with epigenetic memory. This is supported by the association of the past smoking effect on cytokine responses with DNA methylation at specific signal *trans*-activators and regulators of metabolism. Our findings identify three novel variables associated with cytokine secretion variability and reveal roles for smoking in the short- and long-term regulation of immune responses. These results have potential clinical implications for the risk of developing infections, cancers or autoimmune diseases.

High levels of variability exist among individuals and populations in relation to responses to immune challenges[2,7]. This has been highlighted by the COVID-19 pandemic through the diverse clinical outcomes observed after infection with SARS-CoV-2[6,8]. Variables such as age, sex and genetics have a major effect on the way individuals respond to infection[2–6,9,10]. However, such immune variability is generally not considered in the design of treatments or vaccines, and there is a need to better identify the variables associated with immune response variation[11].

The Milieu Intérieur project was developed to assess the factors that contribute to variable 'healthy' immune responses[12]. The cohort is equilibrated in terms of age and sex and comprises individuals of a homogenous genetic background, to facilitate identification of novel immune determinants, in addition to age, sex and genetic variants. The Milieu Intérieur project has already advanced our understanding of the variables that regulate immune homeostasis. In particular, by quantifying the effects of age, sex, genetics and cellular composition on the transcript levels of immune-related genes[4], and the effects of age, sex, cytomegalovirus (CMV) latent infection and smoking on blood leukocyte composition[3].

To identify new environmental factors associated with variability in response to immune stimulation, we focused on cytokine protein secretion as an immune response phenotype. The concentrations of 13 disease- and medically-relevant cytokines (CXCL5, CSF2, IFNγ, IL-1β,

TNF, IL-2, IL-6, IL-8, IL-10, IL-12p70, IL-13, IL-17 and IL-23) were measured with Luminex technology, after 22 h of standardized whole-blood stimulation with 11 immune agonists for the 1,000 Milieu Intérieur donors (Supplementary Table 1), as well as in a non-stimulated control (null condition). The stimulations are classified into 4 categories: microbial (Bacillus Calmette-Guérin (BCG), *Escherichia coli* (*E. coli*), lipopolysaccharide (LPS) and *Candida albicans* (*C. albicans*)) and viral (influenza and polyinosinic–polycytidylic acid (poly I:C)) agents, which are predominantly recognized by receptors on innate immune cells; T cell activators (*Staphylococcus aureus* enterotoxin B superantigen (SEB) and anti-CD3 and anti-CD28 antibodies (anti-CD3 + CD28)), which induce adaptive immune responses; and cytokines (TNF, IL-1β and IFNγ).

## Smoking, CMV and BMI associations

Principal component analysis (PCA) (Extended Data Fig. 1) and heat maps (Extended Data Fig. 2) of the 13 cytokines in the 12 immune stimulations highlight the specific cytokines that were induced in each independent condition. Hierarchical clustering of the standardized log mean differences of the cytokine levels (Fig. 1a) clearly distinguishes groups that broadly correspond to stimulation type. Immune responses induced by innate (*E. coli* and LPS) and adaptive (SEB and anti-CD3 + CD28) stimulations cluster separately, and show greater

[1]Translational Immunology Unit, Department of Immunology, Institut Pasteur, Université Paris Cité, Paris, France. [2]Institut Pasteur, Université Paris Cité, Bioinformatics and Biostatistics Hub, Paris, France. [3]Cytometry and Biomarkers UTechS, Center for Translational Research, Institut Pasteur, Université Paris Cité, Paris, France. [4]DATACTIX, Paris, France. [5]Frontiers of Innovation in Research and Education PhD Program, LPI Doctoral School, Université Paris Cité, Paris, France. [6]Institut Pasteur, Université Paris Cité, CNRS UMR2000, Human Evolutionary Genetics Unit, Paris, France. [7]Institute of Environmental Medicine, Karolinska Institutet, Stockholm, Sweden. [8]Department of Medical Epidemiology and Biostatistics, Karolinska Institutet, Stockholm, Sweden. [9]Octant Biosciences, San Francisco, CA, USA. [10]Chair Human Genomics and Evolution, Collège de France, Paris, France. *A list of authors and their affiliations appears at the end of the paper. ✉e-mail: violaine.saint-andre@pasteur.fr; darragh.duffy@pasteur.fr

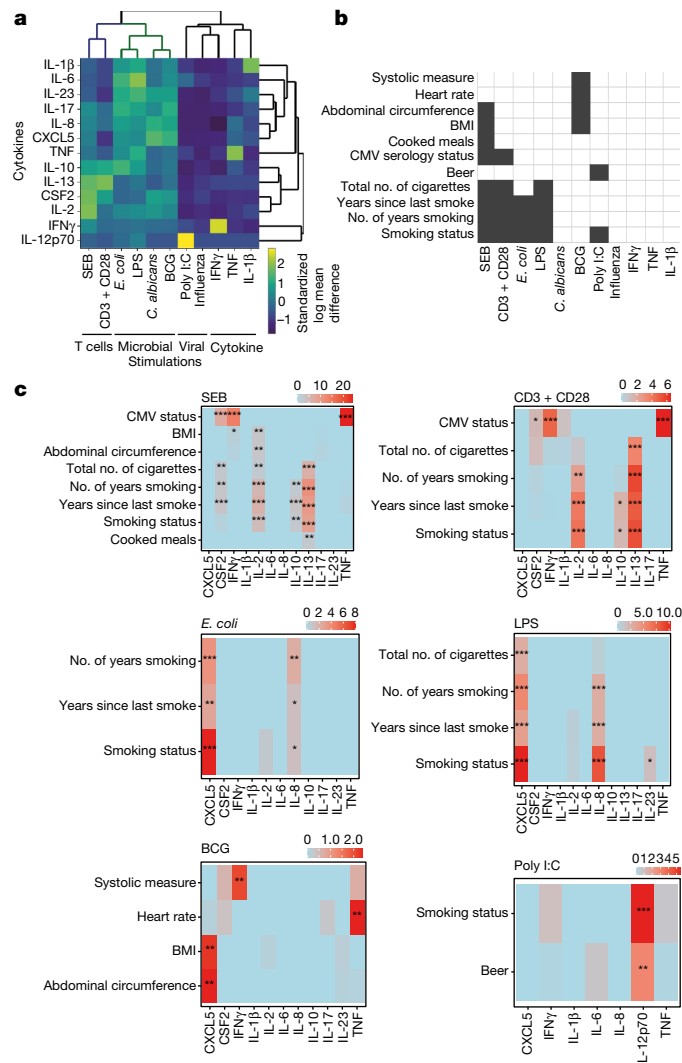

**Fig. 1 | Variables associated with cytokine levels in diverse immune stimulations. a**, Standardized log mean differences of 13 cytokines in 12 immune stimulations. **b**, Significant associations (Benjamini–Yekutieli adjusted *P* value of likelihood ratio test (LRT) < 0.01) of variables with at least one induced cytokine for each immune stimulation are coloured in black. **c**, Heat maps showing −log10(Benjamini–Yekutieli adjusted *P* value of LRT) for the eCRF variables associated with at least one cytokine in each stimulation (Benjamini–Yekutieli adjusted *P* value of LRT < 0.01). *P < 0.05, **P < 0.01, ***P < 0.001.

inter-individual variability in the measured cytokines compared with the other stimulation types.

From the electronic case report form (eCRF), we compiled 136 socio-demographic, environmental, clinical and nutritional variables (Supplementary Table 2) and tested their association with the induced cytokines in each stimulation through likelihood ratio tests (LRTs), using age, sex and experimental batch as covariates. Eleven variables are associated with at least one cytokine in at least one stimulation (Benjamini–Yekutieli adjusted *P* value <0.01) (Fig. 1b). These are related mostly to body mass index (BMI) in SEB and BCG stimulations, CMV latent infection in adaptive immune stimulations and smoking, which shows the most associations across stimulations.

Smoking-related variables are associated with IL-2 and IL-13 in SEB and anti-CD3 + CD28 stimulations, and they are associated with CXCL5 in innate immune stimulations (Fig. 1c). These observations are supported by previous findings showing that smoking favours inflammation and compromises immunity to bacterial infection[13], and that IL-2 and IL-13 are involved in modulating effects of exposure to tobacco[14,15]. CMV

latent infection is associated with CSF2, IFNγ and TNF upon adaptive immune stimulations, in line with our previous work showing strong associations between CMV seropositivity and increased numbers of T cell effector memory subsets[3]. We also observed that BMI-related variables are associated with CXCL5 after BCG stimulation, and with IL-2 after SEB stimulation, which is consistent with the dysregulation of CXCL5 and IL-2 in obesity[16,17]. As potential interactions may exist between our tested variables and age, we performed the same analysis considering age and smoking interactions in the models. The results are very similar to the ones obtained without considering interactions, and some smoking-related variables are associated with even higher significance in SEB, anti-CD3 + CD28, *E. coli* and LPS stimulation conditions (Extended Data Fig. 3). Notably, by including these interactions, smoking-related variables are significantly associated with IL-2 responses after BCG stimulation. This IL-2 response may reflect a long-lived antigen-specific T cell response to BCG vaccination, which all of the cohort received at birth owing to mandatory BCG vaccination in France prior to 2007, further strengthening the associations of smoking with T cell immunity. Individual effects of age and sex have also been tested, and corresponding LRT results and effects sizes are shown on Extended Data Figs. 4 and 5. In addition, as human leukocyte antigen (HLA) is a well-known determinant of immune response variability, which is mostly relevant for antigen-specific responses, we tested associations between previously identified HLA types[3] and induced cytokine responses following the same procedure that we used for the other donor variables. We detected only one significant association, between the major histocompatibility complex class II, DQ beta 1 HLA.DBQ1.1P and IL-6 in the non-stimulated control condition. However no associations were observed with induced cytokine responses after stimulation.

## The smoking effect is persistent

To assess the biological effect of smoking on cytokine secretion, we plotted the effect sizes for the smoking variables from the linear models. We observed that current smoking affects both innate and adaptive immune responses (Fig. 2a and Extended Data Fig. 6). Smoking is associated with stronger induction of CXCL5 after *E. coli* stimulation, and stronger induction of IL-2 and IL-13 after SEB stimulation (Benjamini–Yekutieli adjusted *P* value of LRT < 0.001). The smoking-related variables— number of years smoking, number of years since last smoked, and total number of cigarettes—show consistent associations (Fig. 2b,c).

Of note, in contrast to current smokers (Fig. 2a), past smokers show no significant increase in CXCL5 secretion after innate immune stimulation, whereas they show an increase in IL-2 and IL-13 secretion after adaptive immune stimulation, compared with non-smokers (Fig. 2d). Box plots of cytokine concentrations show that the levels of CXCL5 in past smokers and non-smokers are not significantly different, but differ from those in current smokers after innate immune stimulations (Fig. 2e and Extended Data Fig. 6a,d). Conversely, the levels of IL-2 in current smokers and past smokers are not significantly different but differ from those in non-smokers in adaptive immune stimulations (Fig. 2f and Extended Data Fig. 6b,e). Cytokine production for both current and past smokers correlates with the number of years smoking in adaptive immune stimulations, but this is not the case for past smokers in innate immune stimulations (Fig. 2g and Extended Data Fig. 6c,f). These results collectively show a short-term effect of smoking on innate immune responses, and a long-term effect of smoking on adaptive immune responses.

## Immune cells and plasma proteins

As induced cytokine levels are associated with numbers of specific subsets of circulating immune cells, we tested whether the smoking–cytokine associations remain when considering these cell subsets in our models (Fig. 3 and Extended Data Fig. 7a). The associations of CMV

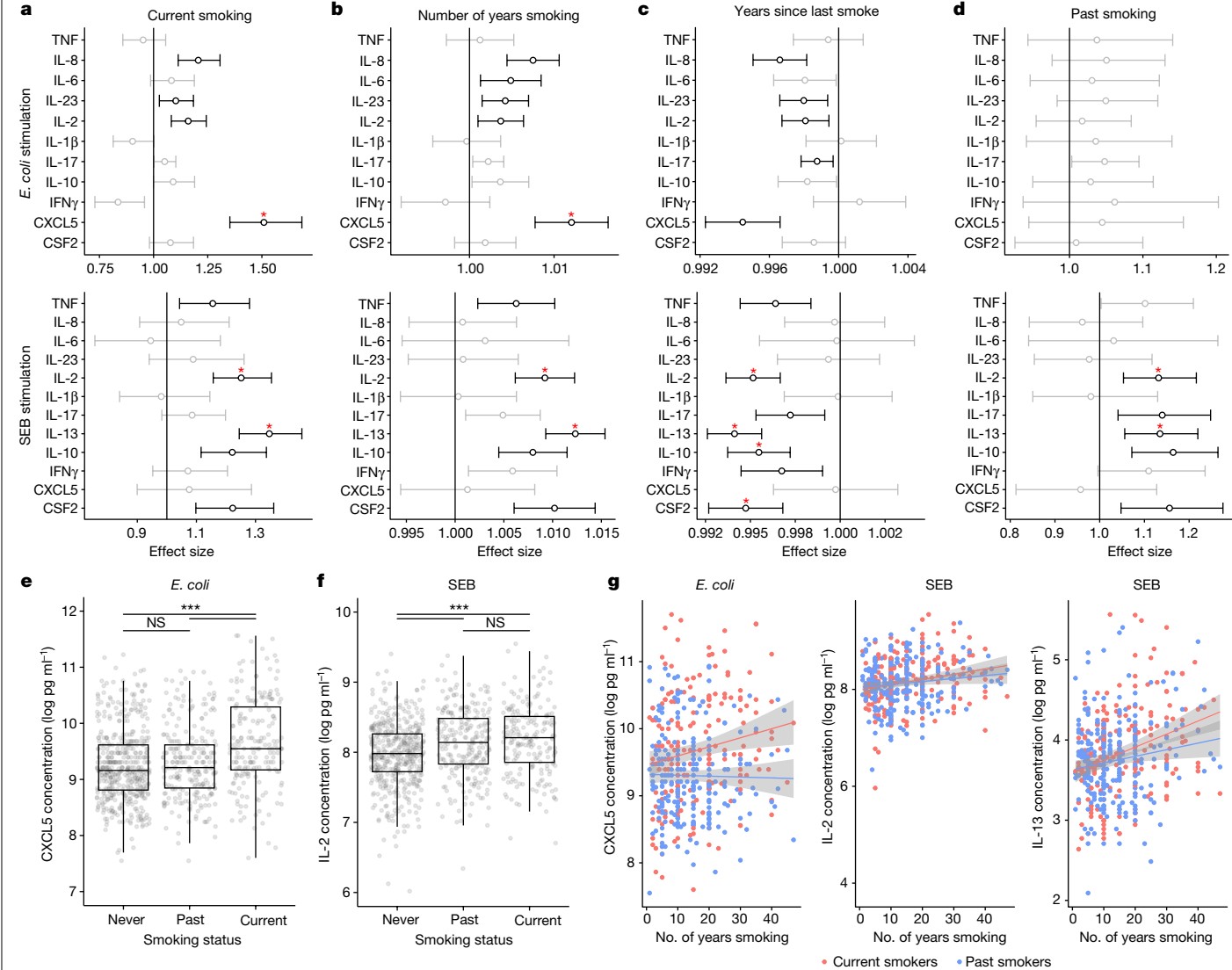

**Fig. 2 | Smoking effects on innate and adaptive immune responses, represented by *E. coli* and SEB stimulations, respectively. a**–**d**, Two-sided effect size plots representing effect on each induced cytokine using *n* = 955 independent individuals with 0.95 confidence interval for current smoking compared with non-smoking (**a**), for number of years smoking (**b**), for years since last smoke (**c**), and for past smoking (**d**) in *E. coli* or SEB stimulation. Significant effect sizes (*P* < 0.01) are in black, others are in grey. Those that also have Benjamini–Yekutieli adjusted *P* value of LRT < 0.01 are labelled with a red star. Exact *P* values are provided in the . **e**,**f**, CXCL5 concentration following *E. coli* stimulation (**e**) and IL-2 concentration following SEB stimulation (**f**) for

never, past and current smokers. Box plots represent *n* = 955 independent individuals. The centre line shows the median, hinges represent the 25th and 75th percentiles and whiskers extend from the hinge to the largest or smallest values no further than 1.5 interquartile range. Two-sided Wilcoxon rank sum tests adjusting for multiple comparisons are shown. *P* values (left to right): 0.18, 7.3 × 10⁻¹² and 2.1 × 10⁻⁷ (**e**); 1.1 × 10⁻⁵, 1.3 × 10⁻⁷ and 0.32 (**f**). **g**, CXCL5 concentration following *E. coli* stimulation and IL-2 and IL-13 concentration following SEB stimulation versus numbers of years smoking for current or past smokers. Grey areas depict the 0.95 confidence intervals of the linear regression lines.

serostatus with the levels of CSF2, IFNγ and TNF in SEB stimulation are lost when specific memory T cell numbers are used as covariates (Extended Data Fig. 7a), which is consistent with the reported association of these cells with CMV latent infection[3], and suggests that the effects of CMV latent infection on cytokines are mediated by changes in blood cell composition. The same analysis performed on the smoking variable for the innate immune stimulations show no clear cell subsets affecting the association of smoking with CXCL5 levels (Fig. 3a). Conversely, upon adaptive immune stimulations, the numbers of multiple B cell and regulatory T cell subsets eliminate the association of smoking with protein levels (Fig. 3a), suggesting that these subsets mediate the smoking effect in adaptive immune responses. We also assessed a role for 326 soluble blood proteins measured in the plasma of a subset of 400 donors[18]. We identified that the levels of carcinoembryonic

antigen-related cell adhesion molecule 6 (CEACAM6), when included in the models, eliminates the association of smoking with CXCL5 after innate immune stimulations, suggesting that CEACAM6 is involved in CXCL5 regulation in smokers (Fig. 3b). We interpret this effect as a biological interaction rather than a shift in immune cell populations, owing to the short stimulation period of 22 h and because the regression analysis including cell numbers as covariates (Fig. 3a) did not identify any cellular associations with the bacteria-induced CXCL5–smoking association.

## DNA methylation changes

To test the hypothesis that an epigenetic mechanism mediates the persistent effect of smoking on adaptive immune responses,

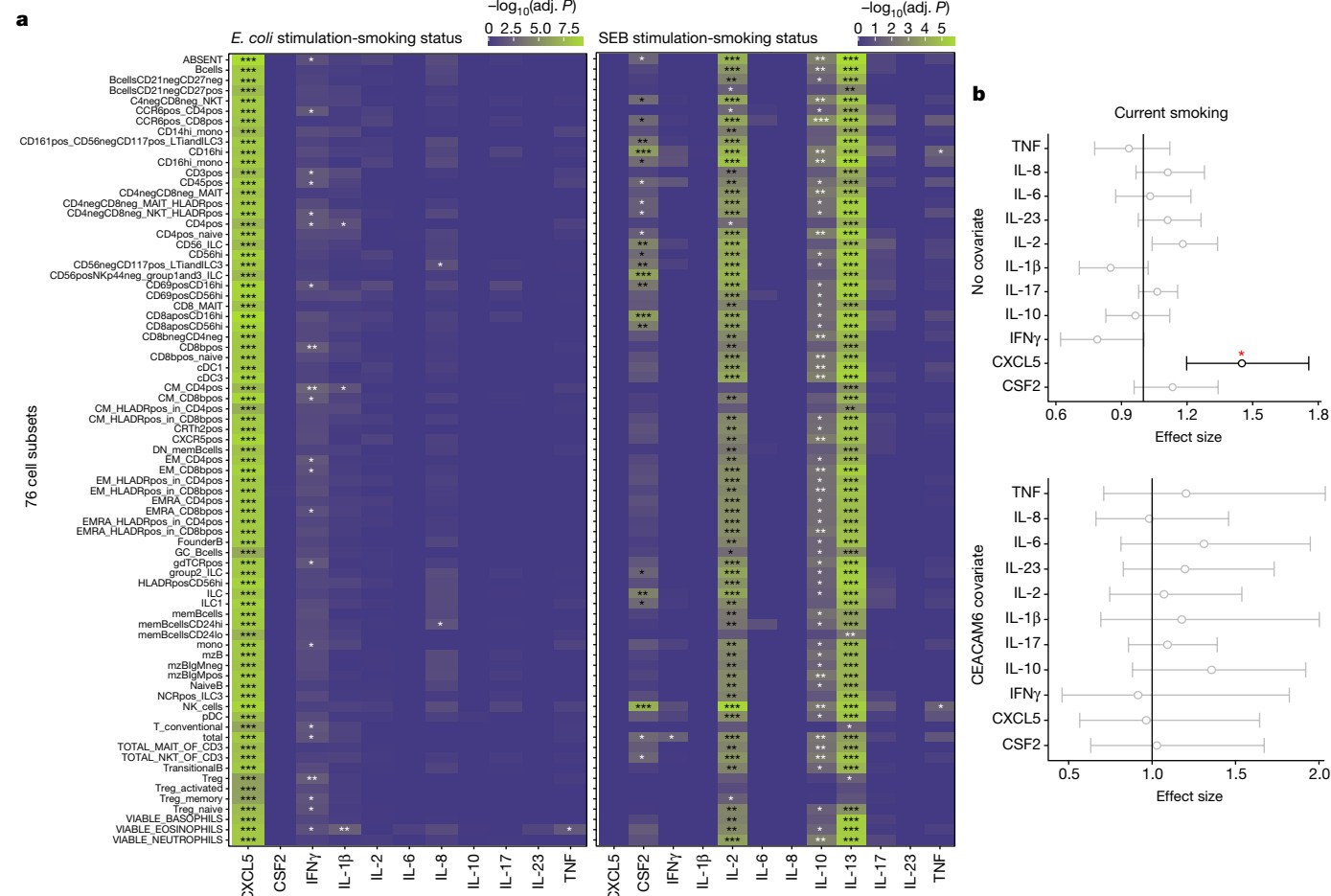

**Fig. 3 | Effects of smoking on induced cytokines is modified by blood cell subsets and plasma proteins. a**, Heat maps showing associations ($-\log_{10}$(Benjamini–Yekutieli adjusted $P$ value of LRT (adj. $P$))) of the smoking status with induced cytokines in *E. coli* and SEB stimulations with either no cell subset count covariate (top line) or each of 76 cell subset counts passed as covariates in the models. **b**, Two-sided effect size plots representing $n = 955$ independent individuals with 0.95 confidence interval for current smokers compared with non-smokers in *E. coli* stimulation when no plasma protein (top) or CEACAM6 (bottom) is passed as a covariate in the models. Significant ($P < 0.01$) effect sizes are in black and others are in grey. The red star indicates a Benjamini–Yekutieli adjusted $P$ value of LRT < 0.001.

we determined whether smoking affected cytokine levels through DNA methylation changes. We measured baseline DNA methylation at more than 850,000 CpG sites using the MethylationEPIC array and identified 2,416 CpG sites that are directly associated with smoking—that is, not mediated by blood cell composition—in the Milieu Intérieur cohort[19]. Among those sites, 129 are significantly associated with IL-2 in SEB stimulation (Benjamini–Yekutieli adjusted $P$ value of the LRT < 0.001). We tested whether the DNA methylation levels of these CpG sites could be related to the association of smoking with cytokine levels after SEB stimulation. We observed that 11 CpGs, when passed as covariates in the models, eliminate the association of smoking with IL-2 and IL-13 (Benjamini–Yekutieli adjusted $P$ value of LRT < 0.001) (Extended Data Fig. 7b). Among these, 3 relate to the aryl hydrocarbon receptor repressor (*AHRR*) gene. The other CpG sites correspond to *F2RL3* and G-protein-coupled receptor (*GPR15*), retinoic acid receptor (*RARA*) and serine protease (*PRSS23*) genes. At all of these loci, we observe a strong DNA hypomethylation in current smokers compared with non-smokers, with past smokers showing an intermediate state of methylation (Fig. 4a). Consistently, in past smokers, the number of years individuals smoked (Fig. 4b), the total number of cigarettes they smoked (Fig. 4c) and IL-2 levels in SEB stimulation (Fig. 4e) negatively correlate with the level of DNA methylation of these genes, whereas the number of years since they

stopped smoking generally positively correlate with their level of DNA methylation (Fig. 4d). These results provide evidence that a persistent effect of smoking on adaptive immune responses is associated with DNA methylation at signal *trans*-activators and metabolism regulators.

## Genomic associations

As some genetic variants manifest their regulatory roles upon stimulation only, our study is particularly well suited to identify response protein quantitative trait loci (pQTLs). We tested a total of 5,699,237 high-quality imputed single nucleotide polymorphisms (SNPs) for associations with the cytokines induced in each stimulation, adjusting for age, sex, technical variables and major immune cell population counts (Supplementary Table 3) and report 44 reponse pQTLs (Table 1). The Somalogic and Olink databases are the main resources of plasma pQTLs, which have identified pQTLs for some of our tested cytokines at steady state[20–24]. However, to our knowledge, only one study—the 500FG study—tested for response pQTLs for some of our tested cytokines in whole blood[25]. Among the common tested cytokine–stimulation couples, both studies identify pQTLs for IL-6 and IL-1β in poly I:C, TNF in *C. albicans* and IL-1β in LPS. Of note, the pQTL (rs3775291) we identify for IL-1β and IL-6 in poly I:C is located in

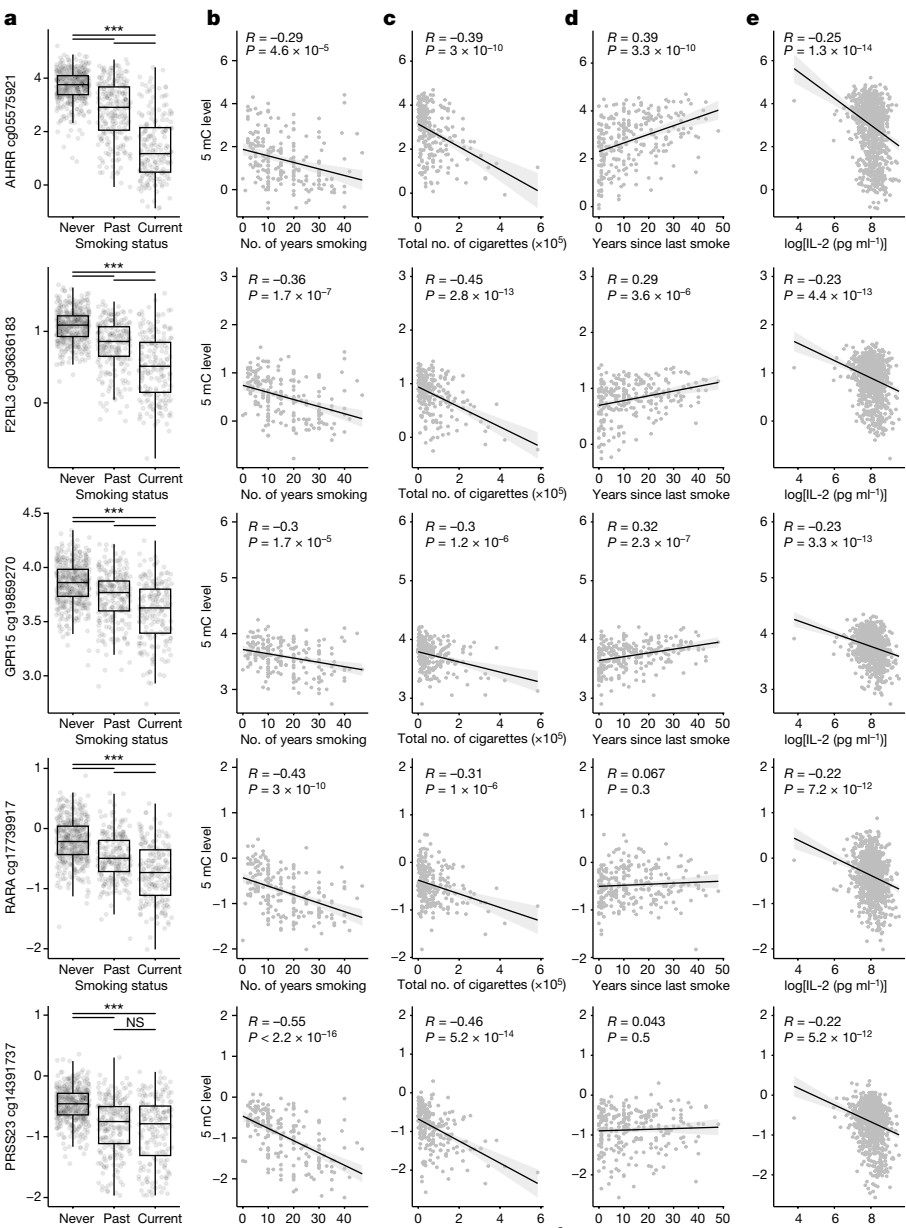

**Fig. 4 | Persistent effect of smoking on adaptive immune responses correlates with DNA methylation at signal *trans*-activators and metabolism regulators. a**, DNA methylation levels for the top probe that removes association of smoking status with IL-2 in SEB stimulation, when passed as a covariate in the models, for the indicated genes and for never, past and current smokers. The centre line shows the median, hinges represent the 25th and 75th percentiles and whiskers extend from the hinge to the largest or smallest values no further than 1.5 interquartile range and *n* = 955. Two-sided Wilcoxon rank sum tests adjusting for multiple comparisons. Left to right: $P < 2.22 \times 10^{-16}$,

$P < 2.22 \times 10^{-16}$, $P < 2.22 \times 10^{-16}$ (cg05575921); $P < 2.22 \times 10^{-16}$, $P < 2.22 \times 10^{-16}$, $9 \times 10^{-14}$ (cg03636183); $7.9 \times 10^{-12}$, $P < 2.22 \times 10^{-16}$, $2 \times 10^{-7}$ (cg19859270); $P < 2.22 \times 10^{-16}$, $P < 2.22 \times 10^{-16}$, $9 \times 10^{-9}$ (cg17739917); $P < 2.22 \times 10^{-16}$, $P < 2.22 \times 10^{-16}$, 0.31 (cg14391737). **b**–**e**, Methylation (5mC level) for each probe depending on the number of years individuals smoked (for current smokers) (**b**), the total lifetime number of cigarettes smoked (for current smokers) (**c**), the number of years since last smoke (for past smokers) (**d**) and IL-2 concentration following SEB stimulation (**e**) for the indicated genes. *R* values and two-sided Pearson correlations are shown.

the TLR3 exon locus, whereas the pQTLs reported by the 500FG study for these cytokines (rs28393318 for IL-1β and rs6831581 for IL-6) are located in the *TLR1–TLR6–TLR10* (*TLR1/6/10*) locus. We performed conditional analysis between the two loci by passing rs28393318 as a covariate in our pQTL identification, and show that the association we report between IL-1β and rs3775291 is maintained, indicating that it is independent of the one reported in the 500FG study (Supplementary Table 4). Among the potential new *trans*-pQTLs we identified, 3 are for CXCL5. One was identified upon BCG stimulation and is located in the *TLR1/6/10* locus. Another one was identified in *C. albicans*

stimulation and is located in the *CR1* locus. The third was observed in poly I:C, IL-1β and IFNγ stimulations, and is in an intron of *JMJD1C*, which encodes a candidate histone H3K27me3 demethylase. We also identified a *trans*-pQTL for IL-2 in anti-CD3 + CD28 stimulation, which is close to the immunoglobulin Fc receptor IIa (*FCGR2A*) gene—identified as an eQTL by the GTEx consortium[26]—that has been associated with multiple autoimmune diseases[27] and conditions the response to anti-CD3 + CD28 stimulation[28]. Another *trans*-pQTL of interest is for IL-1β, IL-12, TNF, IL-6 and IFNγ in poly I:C stimulation, which is located in a *TLR3* exon and associated with age-related macular degeneration[29]

**Table 1 | *cis*-pQTLs and *trans*-pQTLs for induced cytokines in each stimulation**

| Stimulus | Cytokine | rsID | Adjusted *P* value | *cis* or *trans* | Locus |
|---|---|---|---|---|---|
| *E. coli* | IFNγ | rs4833095 | $5.45 \times 10^{-26}$ | *trans* | *TLR1/6/10* locus |
| | IL-2 | rs9306967 | $2.83 \times 10^{-33}$ | *trans* | *TLR1/6/10* locus |
| | IL-6 | rs5743614 | $1.07 \times 10^{-34}$ | *trans* | *TLR1/6/10* locus |
| | IL-8 | rs4833095 | $5.64 \times 10^{-52}$ | *trans* | *TLR1/6/10* locus |
| | IL-17a | rs9306967 | $9.24 \times 10^{-21}$ | *trans* | *TLR1/6/10* locus |
| | IL-23a | rs6815814 | $4.50 \times 10^{-30}$ | *trans* | *TLR1/6/10* locus |
| LPS | CXCL5 | rs352045 | $5.13 \times 10^{-15}$ | *cis* | *CXCL5* promoter |
| | IL-1β | rs3764613 | $1.96 \times 10^{-10}$ | *trans* | *PPP5C* enhancer |
| | IL-6 | rs62449491 | $7.58 \times 10^{-10}$ | *cis* | *IL6* enhancer |
| | IL-10 | rs1518110 | $3.39 \times 10^{-08}$ | *cis* | *IL10* intronic enhancer |
| | IL-12a/b | rs143060887 | $1.37 \times 10^{-10}$ | *cis/trans* | *IL12A* putative regulatory regions |
| SEB | IL-6 | rs11936050 | $1.30 \times 10^{-27}$ | *trans* | *TLR1/6/10* locus |
| Anti-CD3+CD28 (responders) | CXCL5 | rs352045 | $1.33 \times 10^{-36}$ | *cis* | *CXCL5* promoter |
| | IL-2 | rs1801274 | $2.09 \times 10^{-22}$ | *trans* | Close to *FCGR2A* |
| BCG | CXCL5 | rs10013453 | $6.64 \times 10^{-12}$ | *trans* | *TLR1/6/10* locus |
| | IL-2 | rs4833095 | $1.01 \times 10^{-66}$ | *trans* | *TLR1/6/10* locus |
| | IL-6 | rs72636686 | $8.92 \times 10^{-12}$ | *trans* | *KAZN* intron |
| | IL-8 | rs4833095 | $7.21 \times 10^{-109}$ | *trans* | *TLR1/6/10* locus |
| *C. albicans* | CXCL5 | rs10779330 | $9.21 \times 10^{-16}$ | *trans* | *CR1* locus |
| | IL-1β | rs10863358 | $9.56 \times 10^{-15}$ | *trans* | *CR1* locus |
| | IL-23a | rs10779330 | $2.70 \times 10^{-10}$ | *trans* | *CR1* locus |
| | TNF | rs11117956 | $9.46 \times 10^{-15}$ | *trans* | *CR1* locus |
| Influenza | CXCL5 | rs352045 | $3.24 \times 10^{-59}$ | *cis* | *CXCL5* promoter |
| | CXCL5 | rs10822168 | $1.88 \times 10^{-14}$ | *trans* | JMJD1C intron |
| | IL-6 | rs35345753 | $1.61 \times 10^{-07}$ | *cis* | *IL6* enhancer |
| Poly I:C | CXCL5 | rs352045 | $7.15 \times 10^{-59}$ | *cis* | *CXCL5* promoter |
| | CXCL5 | rs10822168 | $5.01 \times 10^{-15}$ | *trans* | *JMJD1C* intron |
| | IFNγ | rs3775291 | $1.36 \times 10^{-13}$ | *trans* | *TLR3* exon |
| | IL-1β | rs3775291 | $1.76 \times 10^{-24}$ | *trans* | *TLR3* exon |
| | IL-6 | rs62449491 | $3.37 \times 10^{-07}$ | *cis* | *IL6* enhancer |
| | IL-6 | rs3775291 | $1.46 \times 10^{-17}$ | *trans* | *TLR3* exon |
| | IL-12a/b | rs3775291 | $1.92 \times 10^{-22}$ | *trans* | *TLR3* exon |
| | TNF | rs113845942 | $1.57 \times 10^{-07}$ | *cis* | *HLA-DRB* intergenic region |
| | TNF | rs3775291 | $4.78 \times 10^{-19}$ | *trans* | *TLR3* exon |
| TNF | CSF2 | rs112997843 | $1.11 \times 10^{-09}$ | *trans* | Intergenic region |
| | CXCL5 | rs352045 | $1.95 \times 10^{-22}$ | *cis* | *CXCL5* promoter |
| | IFNγ | rs352045 | $2.38 \times 10^{-14}$ | *trans* | *CXCL5* promoter |
| | IL-10 | rs6689179 | $1.58 \times 10^{-07}$ | *cis* | *IL10* enhancer |
| IL-1β | CXCL5 | rs352045 | $5.09 \times 10^{-32}$ | *cis* | *CXCL5* promoter |
| | CXCL5 | rs2393969 | $6.35 \times 10^{-10}$ | *trans* | *JMJD1C* intron |
| | IFNγ | rs2564594 | $1.07 \times 10^{-11}$ | *trans* | *CXCL5* promoter |
| IFNγ | CXCL5 | rs352045 | $8.06 \times 10^{-58}$ | *cis* | *CXCL5* promoter |
| | CXCL5 | rs10822168 | $1.49 \times 10^{-14}$ | *trans* | *JMJD1C* intron |

Bonferroni adjusted *P* values of association ($P < 0.05$) obtained with MatrixEQTL using additive genotype effects (least squares model) are shown alongside the locus corresponding to each variant.

and resistance to viral infections[30], consistent with poly I:C stimulation acting through TLR3. Other *trans* associations were identified for IL-1β in LPS stimulation, with the corresponding variant rs3764613, located in a *PPP5C* enhancer, and for IL-6 in BCG stimulation, for which the corresponding variant—rs72636686—is in an intron of the *KAZN* gene, which encodes a cell adhesion protein. We also assessed potential interactions between the identified SNPs and smoking status. No significant associations were observed for *E. coli*, LPS, SEB and anti-CD3 + CD28

stimulations, but BCG stimulation showed significant genetic-smoking interaction (effect size = 1.58 [1.42–1.75], Benjamini–Yekutieli adjusted *P* value = $3.8 \times 10^{-13}$) (Extended Data Fig. 8a,b) between rs72636686 and smoking status for IL-8 levels. This interaction shows that smoking status can remove differences in response to some immune stimulation between individuals of different genotypes (Extended Data Fig. 8c), supporting modulation of genotype effects by the smoking status.

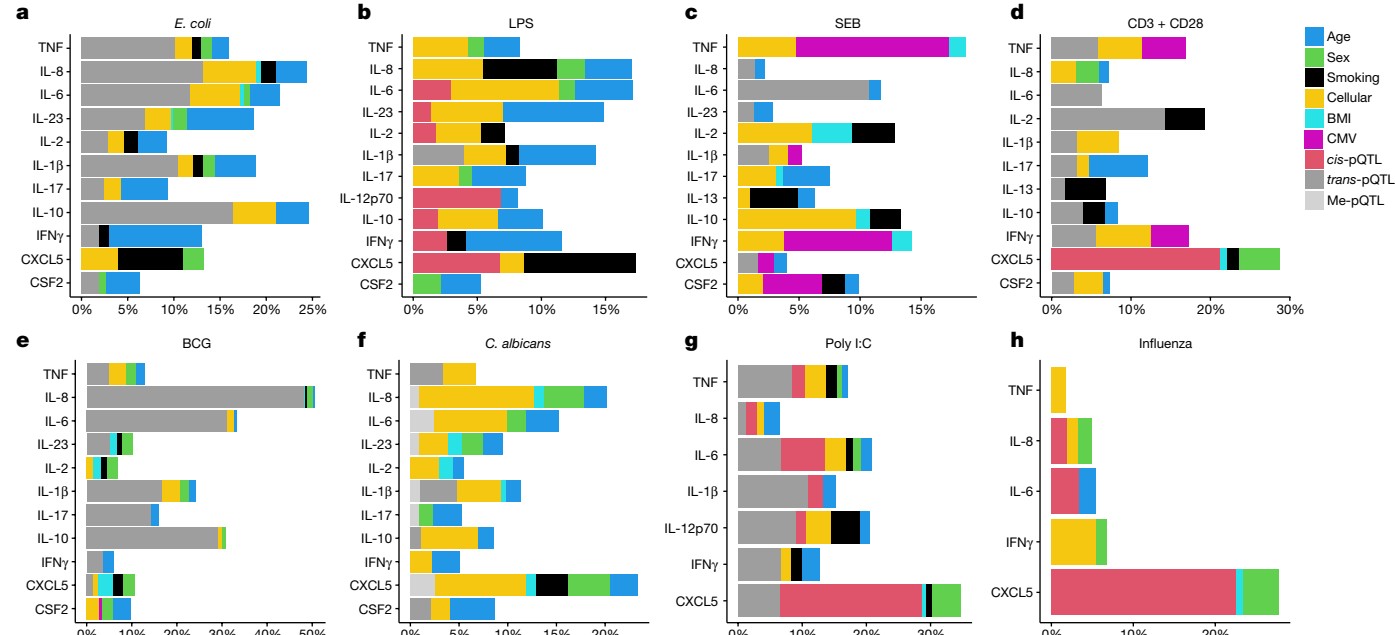

**Fig. 5 | Induced cytokine variance explained. a–h**, Percentages of variance explained by each variable associated with at least one induced cytokine in *E. coli* (**a**), LPS (**b**), SEB (**c**), anti-CD3 + CD28 (**d**), BCG (**e**), *C. albicans* (**f**), poly I:C (**g**) and influenza (**h**) stimulations. $R^2$ contributions averaged over orderings among regressors are represented on each plot.

## Cytokine variance explained

To assess the relative effect of the smoking associations identified, we quantified how inter-individual variance in each induced cytokine may be explained by each associated variable. The genetic and epigenetic loci, major immune cell subsets, age, sex, BMI, CMV latent infection and smoking status were combined in the same model and their respective effects on induced cytokine levels were computed (Fig. 5a–h). We observed that smoking explained between 4 and 9% of the inter-individual variance of the associated cytokines, a level equivalent to age, sex or genetic effects when present.

## Discussion

Here we show that smoking status, CMV latent infection and BMI, in addition to age, sex, genetic variation, DNA methylation levels and immune cell subsets, are the variables most associated with variation in cytokine secretion upon immune challenge. Our approach to study induced immune responses to diverse stimuli was validated by the identification of new environmental and genetic determinants for 11 of the 13 cytokines studied, associations that were detected across the different stimulation conditions. The effects of smoking are of particular interest, as they have effects on both innate and adaptive immune responses (Extended Data Fig. 10c). The variance explained by the smoking status for some cytokines upon stimulation reaches a level equivalent to those of age, sex and genetic variants, all of which are known to have implications for disease risk. Current smokers showed an increased inflammatory response following bacterial stimulation, which is quickly lost upon smoking cessation. Conversely, the smoking effects on T cell responses persist years after individuals quit smoking.

The association of the smoking effect with long-lived B and T cell subsets and DNA methylation at signal *trans*-activators and metabolism regulators highlights a mechanism for the persistent effects in the adaptive response. *AHRR*, *F2RL3*, *GP15*, *PRSS23* and *RARA* CpG sites were previously identified as candidate smoking-related loci in whole blood[31,32]. Of note, the CpG site most associated with smoking—within the *AHRR* gene—is a quantitative biomarker of smoking cessation[33] and

is associated with increased risk for chronic obstructive pulmonary disease and lung cancer[34]. Although the interplay of AHRR in the AHR pathway is complex and depends on the cell-type specific balancing of AHR and AHRR, the repressive activity of AHRR is due to displacement or inhibition of AHR binding to xenobiotic response elements in the promoter of its targets[35]. Decreased expression of xenobiotic metabolizing genes may, in turn, compromise the ability of the body to metabolize harmful agents, potentially leading to impaired lung function. Our work thus suggests that smoking can induce specific epigenetic modifications that could subsequently lead to altered immune responses.

By contrast, we did not identify putative cellular mediators for the increased inflammatory response to innate stimulation in smokers. We found a strong association between induced levels of CXCL5 with circulating levels of CEACAM6, a glycosylphosphatidylinositol cell surface protein that belongs to the CEACAM immunoglobulin supergene family. CXCL5 regulates neutrophil trafficking to the lung via CXCR2 and has been implicated in asthma and multiple cancers[36]. CEACAM6 is expressed on the surface of neutrophils, macrophages and lung and intestinal epithelial cells and its levels are increased in multiple cancers, such as breast and gastric cancer where it has been proposed as a clinical biomarker[37]. Although previous studies have suggested that levels of CXCL5[38] and CEACAM6[39] may be increased in smokers, most studies have focused on pulmonary sites and in patients with respiratory disease (such as cancer, chronic obstructive pulmonary disease and asthma), which can confound the results. Our study identifies a strong link between these previously proposed disease biomarkers and response to immune challenges in smokers versus non-smokers. Furthermore, our findings in healthy donors open avenues for further exploration into understanding how smoking acts as a risk factor for cancers beyond the lungs.

Although our study has revealed novel effects of environmental variables on immune responses, it does present some limitations. One is the absence of a replication cohort to validate these findings. Furthermore, our analyses were conducted on a population of similar genetic background, but ongoing efforts are underway to include other populations from diverse ancestries. Future studies will also aim to

identify the transcriptional regulatory networks[40] that underly the persistent effect of smoking on adaptive immune responses. These findings provide new understanding on the effects of smoking on human health, and the role of modifiable environmental effects on immune response variability.

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

**The Milieu Intérieur Consortium**

Laurent Abel[11,12], Andres Alcover[13], Hugues Aschard[14], Philippe Bousso[15], Nollaig Bourke[16], Petter Brodin[17], Pierre Bruhns[18], Nadine Cerf-Bensussan[19], Ana Cumano[20], Christophe D'Enfert[21], Caroline Demangel[22], Ludovic Deriano[23], Marie-Agnès Dillies[2], James Di Santo[24], Gérard Eberl[25], Jost Enninga[26], Jacques Fellay[27], Ivo Gomperts-Boneca[28], Milena Hasan[3], Gunilla Karlsson Hedestam[29], Serge Hercberg[30], Molly A. Ingersoll[31,32], Olivier Lantz[33], Rose Anne Kenny[34], Mickaël Ménager[35], Frédérique Michel[36], Hugo Mouquet[37], Cliona O'Farrelly[38,39], Etienne Patin[6], Antonio Rausell[40], Frédéric Rieux-Laucat[41,42], Lars Rogge[43], Magnus Fontes[44], Anavaj Sakuntabhai[45], Olivier Schwartz[46], Benno Schwikowski[47], Spencer Shorte[48], Frédéric Tangy[49], Antoine Toubert[50], Mathilde Touvier[30], Marie-Noëlle Ungeheuer[51], Christophe Zimmer[52], Matthew L. Albert[9], Darragh Duffy[1,3] & Lluis Quintana-Murci[6,10]

[11]Laboratory of Human Genetics of Infectious Diseases, Necker Branch, INSERM UMR1163, Paris, France. [12]Necker Hospital for Sick Children, Paris, France. [13]Institut Pasteur, Université Paris Cité, INSERM-U1224, Unité Biologie Cellulaire des Lymphocytes, Ligue Nationale Contre le Cancer-Équipe Labellisée Ligue 2018, Paris, France. [14]Institut Pasteur, Université Paris Cité, Department of Computational Biology, Paris, France. [15]Dynamics of Immune Responses Unit, Institut Pasteur, INSERM U1223, Université de Paris Cité, Paris, France. [16]Department of Medical Gerontology, School of Medicine, Trinity Translational Medicine Institute, Trinity College Dublin, The University of Dublin, Dublin, Ireland. [17]Department of Women's and Children's Health, Karolinska Institutet, Solna, Sweden. [18]Unit of Antibodies in Therapy and Pathology, INSERM UMR1222, Institut Pasteur, Université de Paris Cité, Paris, France. [19]Université Paris-Cité, Institut Imagine, Laboratory of Intestinal Immunity, INSERM U1163, Paris, France. [20]Unit of Lymphocytes and Immunity, Immunology Department, Institut Pasteur, INSERM U1223, Université de Paris Cité, Paris, France. [21]Institut Pasteur, Université Paris Cité, INRAE USC2019, Unité Biologie et Pathogénicité Fongiques, Paris, France. [22]Institut Pasteur, Université Paris Cité, Inserm U1224, Immunobiology and Therapy Unit, Paris, France. [23]Institut Pasteur, Université Paris Cité, INSERM U1223, Équipe Labellisée Ligue Contre Le Cancer, Genome Integrity, Immunity and Cancer Unit, Paris, France. [24]Institut Pasteur, Innate Immunity Unit, Université Paris Cité, Inserm U1223, Paris, France. [25]Institut Pasteur Université de Paris Cité, Inserm U1224, Microenvironment and Immunity Unit, Paris, France. [26]Dynamics of Host-Pathogen Interactions, Institut Pasteur, Université de Paris Cité, CNRS UMR3691, Paris, France. [27]School of Life Sciences, École Polytechnique Fédérale de Lausanne, Lausanne, Switzerland. [28]Institut Pasteur, Université Paris Cité, CNRS Unité Mixe de Recherche 6047, INSERM U1306, Unité de Biologie et génétique de la paroi bactérienne, Paris, France. [29]Department of Microbiology, Tumor and Cell Biology, Karolinska Institutet, Stockholm, Sweden. [30]Université Sorbonne Paris Nord and Université Paris Cité, Inserm, INRAE, CNAM, Centre de Recherche in Epidemiology and StatisticS (CRESS), Nutritional Epidemiology Research Team (EREN), Bobigny, France. [31]Mucosal Inflammation and Immunity, Department of Immunology, Institut Pasteur, Paris, France. [32]Université Paris Cité, Institut Cochin, INSERM U1016, CNRS UMR 8104, Paris, France. [33]Center for Cancer Immunotherapy, INSERM U932,

PSL Research University, Institut Curie, Paris, France. [34]The Irish Longitudinal Study on Ageing, School of Medicine, Trinity College Dublin, Dublin, Ireland. [35]Université Paris Cité, Imagine Institute, Laboratory of Inflammatory Responses and Transcriptomic Networks in Diseases, Atip-Avenir Team, INSERM UMR1163, Paris, France. [36]Cytokine Signaling Unit, INSERM U1224, Institut Pasteur, Paris, France. [37]Institut Pasteur, Université Paris Cité, INSERM U1222, Humoral Immunology Unit, Paris, France. [38]School of Biochemistry and Immunology, Trinity Biomedical Sciences Institute, Trinity College Dublin, Dublin, Ireland. [39]School of Medicine, Trinity College Dublin, Dublin, Ireland. [40]Université Paris Cité, Institut Imagine, Inserm U1163, Paris, France. [41]University of Paris Cité, Paris, France. [42]Laboratory of Immunogenetics of Pediatric Autoimmune Diseases, Imagine Institute, INSERM UMR 1163, Paris, France. [43]Immunoregulation Unit, Institut Pasteur, Université Paris Cité, Paris, France. [44]Institut Roche, Paris, France. [45]Institut Pasteur, Université de Paris, Functional Genetics of Infectious Diseases Unit, Department of Global Health, Paris, France. [46]Virus and Immunity Unit, Institut Pasteur, Université de Paris Cité, Paris, France. [47]Computational Systems Biomedicine Lab, Institut Pasteur, Université Paris Cité, Paris, France. [48]UTechS Photonic BioImaging/C2RT, Institut Pasteur, Université Paris Cité, Paris, France. [49]Vaccines Innovation Laboratory, Institut Pasteur, Université de Paris Cité, Paris, France. [50]Université Paris Cité, Institut de Recherche Saint Louis, EMiLy, INSERM UMR S1160, Paris, France. [51]Investigational Clinical Service and Access to Research Bio-Resources (ICAReB), Institut Pasteur, Paris, France. [52]Institut Pasteur, Université Paris Cité, CNRS UMR 3691, Imaging and Modeling Unit, Paris, France.

## Methods

### Human samples

Human samples came from the Milieu Intérieur cohort, which was approved by the Comité de Protection des Personnes–Ouest 6 on 13 June 2012, and by the French Agence Nationale de Sécurité du Médicament (ANSM) on 22 June 2012. The study is sponsored by Institut Pasteur (Pasteur ID-RCB Number: 2012-A00238-35) and was conducted as a single-centre interventional study without an investigational product. The original protocol was registered under ClinicalTrials.gov (study no. NCT01699893). The samples and data used in this study were formally established as the Milieu Intérieur biocollection (NCT03905993), with approvals by the Comité de Protection des Personnes–Sud Méditerranée and the Commission Nationale de l'Informatique et des Libertés (CNIL) on 11 April 2018. Donors gave written informed consent. The 1,000 donors of the Milieu Intérieur cohort were recruited by BioTrial to be composed of healthy individuals of the same genetic background (Western European) and to have 100 women and 100 men from each decade of life between 20 and 69 years of age. Donors were selected based on various inclusion and exclusion criteria that were previously described[12]. In brief, donors were required to have no history or evidence of severe, chronic or recurrent pathological conditions, neurological or psychiatric disorders, alcohol abuse, recent use of drugs, recent vaccine administration and recent use of immune modulatory agents. To avoid the influence of hormonal fluctuations in women, pregnant and peri-menopausal women were not included. To avoid genetic stratification in the study population, the recruitment of donors was restricted to individuals whose parents and grandparents were born in Metropolitan France. Additionally, we formally checked how the genetic background of the donors could affect cytokine levels by performing association tests between the first 20 genetic principal components out of the PCA on the individual genotypes and each of the induced cytokines in each stimulation. Although PC1 had significant association with IL-10 (Benjamini–Yekutieli adjusted $P$ value < 0.05), we found that the first 20 principal components showed no significant associations with cytokine responses at the $P$ value threshold (Benjamini–Yekutieli adjusted $P$ value < 0.01) we use throughout this study. To illustrate the homogeneity of the genetic structure of the 1,000 individuals of the Milieu Intérieur cohort, a PCA was performed with EIGENSTRAT[41] on 261,827 independent SNPs and 1,723 individuals, which include the 1,000 Milieu Intérieur donors together with 723 individuals from a selection of 36 populations originating from North Africa, the Near East, as well as western and northern Europe[42] is shown, similarly to what was previously performed[3]. PC1 versus PC2, PC1 versus PC3 and PC2 versus PC3 are displayed as well as a bar plot of the variance explained by the first 20 components of the PCA (Extended Data Fig. 9b). Unless otherwise stated, all displayed results have been performed on the 955 individuals of the cohort who gave consent to share their data publicly, in order to ensure easy reproducibility of the results.

### TruCulture whole-blood stimulations

TruCulture whole-blood stimulations were performed in a standardized way as previously described[4,43]. Briefly, tubes were prepared in batch with the indicated stimulus, resuspended in a volume of 2 ml buffered medium, and maintained at −20 °C until time of use. Stimuli used in this study were LPS derived from *E. coli* O111:B4 (Invivogen), *E. coli* O111:B4 (Invivogen), *C. albicans* (Invivogen), vaccine-grade poly I:C (Invivogen), live Bacillus Calmette-Guerin (Immucyst, Sanofi Pasteur), live H1N1 attenuated influenza A/PR8 (IAV) (Charles River), SEB (Bernhard Nocht Institute), CD3 + CD28 (R&D Systems and Beckman Coulter), and cytokines TNF (Miltenyi Biotech), IL-1β (Peprotech) and IFNγ (Boehringer Ingelheim). One millilitre of whole blood was distributed into each of the prewarmed TruCulture tubes, inserted into a dry block incubator, and maintained at 37 °C room air for 22 h. At the end of the incubation period, tubes were opened, and a valve was inserted in order to separate the sedimented cells from the supernatant and to stop the stimulation reaction. Liquid supernatants were aliquoted and immediately frozen at −80 °C until the time of use.

### Luminex multi-analyte profiling

Supernatants from TruCulture tubes were analysed by Rules Based Medicine using the Luminex xMAP technology. Samples were analysed according to the Clinical Laboratory Improvement Amendments (CLIA) guidelines. The lower limit of quantification (LLOQ) was determined as previously described[43], and is the lowest concentration of an analyte in a sample that can be reliably detected and at which the total error meets CLIA requirements for laboratory accuracy. The 13 cytokines (CXCL5, CSF2, IFNγ, IL-1β, TNF, IL-2, IL-6, IL-8, IL-10, IL-12p70, IL-13, IL-17 and IL-23), which were measured in this study, were selected to best capture broad immune response variability. Among 109 analytes initially tested, these are the ones that captured the maximum variance following stimulation with the 4 stimuli (LPS, BCG, poly I:C and SEB) that showed the most distinct immune responses among 27 stimuli tested on a subset of 25 individuals of the Milieu Intérieur cohort.

### Principal components analysis

The PCA in Extended Data Fig. 1 was created in R 4.2.1 using the FactoMineR 2.8 package. The data were log-transformed and by default scaled to unit and missing values were imputed by the mean of the variable.

### Cytokine induction visualization

Cytokines were considered induced if the absolute value of their median concentration in the stimulated condition was 30%-fold of their concentration in the null condition. Standardized log mean differences were computed as follows (mean(concentration of the cytokine in the stimulated condition) − mean(concentration of the cytokine in the null condition))/s.d.((concentration of the cytokine in the stimulated condition) − (concentration of the cytokine in the null condition)) and the corresponding heat map was generated with heatmaply 1.0.0 and dendextend 1.13.12 with 'complete' clustering method and 'euclidean' distance in R version 4.2.1.

### Identification of CD3 + CD28 non-responders

Levels of cytokines that we focused on are low to undetectable in the non-stimulated condition, and cytokine induction is generally homogenous within this healthy population of individuals, with no clear distinguishable groups of responders and non-responders, except for anti-CD3 + CD28 stimulation (Extended Data Fig. 2). For the anti-CD3 + CD28 stimulation, we identified through $k$-means clustering a group of 705 individuals that responded to the stimulation and a group of 295 individual did not. This lack of response of 295 individuals is explained by a FcγRIIA polymorphism (rs1801274) that was previously described as preventing response to this anti-CD3 + CD28 stimulation[28] (Extended Data Fig. 9). All statistical analyses on anti-CD3 + CD28 stimulations in this study were thus performed on the responders only.

### eCRF criteria association tests with induced cytokines

Variables were extracted from the eCRF filled by the donors with the help of a physician. To limit biases in associations, categorical variables had to have at least 5% of individuals in at least half of the categorical levels to be considered for association tests. Such categorical variables or numerical ones were tested for associations with the log-transformed induced cytokine levels in each stimulation through LRTs, using age, sex and the technical variable batchID (corresponding to two batches of TruCulture tubes produced at different periods of time) as covariates: the LRT compared the models lm(cytokine ~ variable + age + sex + batchID) with lm(cytokine ~ age + sex + batchID), followed by Benjamini–Yekutieli multiple testing correction applied

to the whole heat maps, so taking into account the tests made for the 136 CRF variables with all the induced cytokines in a specific stimulation. For Extended Data Figs. 4 and 5, the models compared were lm(cytokine ~ age + sex + batchID) with lm(cytokine ~ sex + batchID) for age and lm(cytokine ~ sex + batchID) with lm(cytokine ~ age + batchID) for sex. $P$ values of association tests were represented using ggplot2 3.2.1 in R 3.6.0. Adjusted $P$ values on the box plots were computed with the wilcox.test function, correcting for multiple testing. Versions of the box plots and scatter plots made on the residuals after regression on age, sex and batchID are displayed on Extended Data Fig. 6d–f.

### Effect size plots

Linear regression models were estimated in each stimulation using the log-transformed induced cytokine levels as outcome and age, sex, batchID, and the covariates of interest (for example, smoking status) as predictor variables. Interactions with the covariates of interest were considered when indicated. Exponential of the regression coefficient estimates, and their 95% confidence interval were plotted. When the covariate of interest is of categorical nature, each level of the variable is shown independently, considering the one specified as the reference. When the $P$ value of the $t$-test testing if the coefficient estimate is different from zero is <0.01, it is plotted in black, otherwise it is plotted in grey. If the LRT comparing the regression with and without the variable of interest in the model with a Chi-square test is significant with a Benjamini–Yekutieli adjusted $P$ value < 0.01, a red star is added above the effect size value and interval.

### Cell subset association tests

Acquisition of flow cytometry data was detailed previously[3]. Association tests with log-transformed values of induced cytokines in each stimulation were performed as for the eCRF criteria association tests using log-transformed raw counts of cell subsets for each donor. $P$ values of significance are indicated with asterisks as follows: *$P$ < 0.05; **$P$ < 0.01; ***$P$ < 0.001.

### DNA methylation association tests

CpG methylation profiles were generated using the Infinium MethylationEPIC BeadChip (Illumina) on genomic DNA treated with sodium bisulfite (Zymo Research) for 958 individuals of the Milieu Intérieur cohort as described[19]. Associations between the DNA methylation levels for the CpG sites located within 1 Mb of each cytokine gene transcription start site (TSS) and the levels of log-transformed induced cytokines in each stimulation, adjusting for age, sex, technical variable batchID and major immune cell population counts for each stimulation, were tested through LRT and identified CpG sites weakly associated with IL-17 in LPS (cg09582880), IL-2 in $C.$ $albicans$ (cg17850932 and cg25065535) and IL-8 in influenza (cg16468729) stimulations (FDR adjusted $P$ value of LRT < 0.05) (Extended Data Fig. 10). These effects were mild compared with the identified associated genetic variants and the other associated variables identified in this study but are considered in the final global models (Fig. 5). CpG sites with DNA methylation levels that are directly affected by smoking have been selected as described[19].

### Heat maps showing effects of covariates

To test if the levels of some covariates, such as cell subsets, plasma proteins or DNA methylation probes, could modify the observed association of a variable, such as smoking status, with the log-transformed induced levels of cytokines in each stimulation, we compared with a LRT for each cytokine in each stimulation the model considering both the variable of interest and the covariate of interest (with interactions) plus the usual covariates age, sex and the technical covariate batchID, with a model containing all the covariates but not the variable of interest, followed by a Benjamini–Yekutieli multiple testing adjustment on the whole heat maps. For example, for Fig. 3a, the variable of interest was smoking status and the covariate of interest was each cell subset, so we compared lm(cytokine ~ smoking status × cell subset + age + sex + batchID) with lm(cytokine ~ cell subset + age + sex + batchID). When the LRT is significant, it means adding the variable of interest to the model improves the fit to the cytokine levels. For BMI-related variables, these do not improve the fit to both IL-2 and CXCL5 when T cell subsets are passed as covariates, showing that our approach is powered to identify cellular associations with effects on CXCL5 levels when present.

### pQTL analyses

Protocols and quality-control filters for genome-wide SNP genotyping are detailed in ref. 3. In brief, the 1,000 Milieu Intérieur donors were genotyped on both the HumanOmniExpress-24 and the HumanExome-12 BeadChips (Illumina), which include 719,665 SNPs and 245,766 exonic SNPs, respectively. Average concordance rate between the two genotyping arrays was 99.99%. The final dataset included 732,341 high-quality polymorphic SNPs. After genotype imputation and quality-control filters, 11,395,554 SNPs were further filtered for minor allele frequencies > 5%, yielding a dataset composed of 1,000 donors and 5,699,237 SNPs for pQTL mapping. pQTL analyses were performed using the MatrixEQTL[44] 2.2 R package. SNPs were considered as $cis$-acting pQTLs if they were located within 1 Mb of the TSS of the gene, otherwise they were considered as $trans$-pQTLs. Protein expression data of the 1,000 individuals were log-transformed prior to pQTL analysis. Bonferonni correction for multiple testing (adjusted $P$ value < 0.05) was applied to the results. We used detection thresholds of $10^{-3}$ for $cis$-pQTLs and $10^{-5}$ for $trans$-pQTLs and age, sex and the technical covariate batchID, as well as a main associated cell subset (monocytes for LPS, $E.$ $coli$ and $C.$ $albicans$ stimulations, CD4pos for SEB, CD8posEMRA for anti-CD3 + CD28, CD45pos for BCG, cDC3 for poly I:C, CD3pos for influenza, CD45pos for TNF, none for null, IL-1β and IFNγ) as covariates. SNPs associated with IFNγ in IFNγ stimulation, with IL-1β in IL-1β stimulation and with TNF in TNF stimulation were disregarded because each of these cytokines were respectively added to the TruCulture tubes and thus do not reflect endogenous secretion. To test the novelty of our pQTL results, we studied the SomaLogic plasma protein pQTL database[20], for both $cis$- and $trans$-pQTLs listed in Table 1. This dataset allowed testing associations for CXCL5, IFNγ, IL-1β, IL-2, IL-6, IL-10 and IL-12a. Significant associations were identified between the variants rs352045 ($cis$), rs2393969 ($trans$), rs10822168 ($trans$) and the protein CXCL5 (respective FDR adjusted $P = 3.02 × 10^{-10}$, $P = 0.01$ and $P = 0.022$), between rs35345753 ($cis$), rs62449491 ($cis$) and IL-6 (respective FDR adjusted $P = 4.17 × 10^{-3}$ and $P = 0.017$) and between rs3775291 ($trans$) and IL-12A (FDR adjusted $P = 0.049$). To test associations for SNPs in linkage disequilibrium with the SNPs originally referenced in Table 1, we used a dataset of linkage disequilibrium from the ensemble database with similar ancestries as the Milieu Intérieur cohort (1000GENOMES:phase_3:CEU: Utah residents with Northern and Western European ancestry). To be inclusive, SNPs with a $r^2 > 0.2$ were selected as associated alleles and underwent the same analysis as the one performed with the SNPs of reference. SNPs that came out as significant are those in linkage disequilibrium with the SNP referenced in Table 1 that is significantly associated with the corresponding protein. In addition, we also screened eQTL results. We compared our pQTL results with the eQTLs reported in our previous work based on nanostring transcriptomic data for common cytokines (CSF2, IFNγ, IL-1β, TNF, IL-2, IL-6, IL-8, IL-10, IL-12p70, IL-13, IL-17 and IL-23) and stimulations ($E.$ $coli$, $C.$ $albicans$, influenza, BCG, and SEB)[4], which identified 2 main loci: the $TLR1/6/10$ locus and the $CR1$ locus. Association of variants referenced in Table 1 were found in the GTEx consortium database for rs1518110 and IL-10 (FDR adjusted $P = 4.3 × 10^{-9}$), for rs352045 ($cis$) and CXCL5 (FDR adjusted $P = 9.2 × 10^{-23}$) in whole blood and for rs143060887 ($cis$) and IL-12A (FDR adjusted $P = 0.000076$). Significant associations between rs352045 and CXCL5 and between rs1518110 and IL-10 were also found in the eQTLgen catalogue.

## Computation of variance explained

For each stimulation, all the variables associated with at least one induced cytokine were considered to compute the percentage of each induced cytokine variance explained by each associated variable ($q$ value < 0.05) with the R package relaimpo 2.2.3 and plotted with the R package ggplot2 3.2.1. The $R^2$ contribution averaged over orderings among regressors was computed using the lmg type in the calc.relimp function of the relaimpo R package. For this analysis log-transformed induced cytokine data and log-transformed raw counts of cell subsets were used, as well as data for *cis*- and *trans*-associated SNPs and methylation probes. For each stimulation, all associated *cis*-pQTLs (rs352045, rs143060887, rs62449491 and rs1518110 for LPS; rs352045 for anti-CD3 + CD28, rs352045, rs62449491 and rs113845942 for poly I:C; rs352045 and rs35345753 for influenza), and *trans*-pQTLs (rs3764613 for LPS; rs4833095 for *E. coli*; rs11936050 for SEB; rs1801274 for anti-CD3 + CD28; rs4833095, rs72636686 and rs10013453 for BCG; rs10779330 and rs11117956 for *C. albicans*; rs3775291 and rs10822168 for poly I:C), as well as methylation probes (cg09582880 for LPS; cg25065535 for *C. albicans*, cg17850932 for poly I:C and cg16468729 for influenza) and a main associated cell subset (monocytes for LPS, *E. coli* and *C. albicans* stimulations, CD4pos for SEB, CD8posEMRA for anti-CD3 + CD28, CD45pos for BCG, cDC3 for poly I:C, CD3pos for influenza) were considered in the models.

## Reporting summary

Further information on research design is available in the Nature Portfolio Reporting Summary linked to this article.

## Data availability

SNP array data can be accessed from the European Genome-Phenome Archive EGA under accession EGAS00001002460, the Infinium MethylationEPIC raw intensity data can be accessed via https://dataset.owey.io/doi/10.48802/owey.f83a-1042. Cytokine data are provided in Supplementary Data 1 and eCRF data are provided in Supplementary Data 2. Source data are provided with this paper.

## Code availability

All the scripts related to this study are available on Github https://github.com/ViolaineSaint-Andre/ and have a tagged copy on Zenodo: https://doi.org/10.5281/zenodo.10198929.

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

**Acknowledgements** This programme is managed by the Agence Nationale de la Recherche. This work benefited from support of the French government's Invest in the Future programme; reference ANR-10-LABX-69-01. A. Bertrand is a student from the FIRE PhD programme funded by the Bettencourt Schueller Foundation and the EURIP Graduate Program (ANR-17-EURE-0012).

**Author contributions** Conceptualization: M.L.A., L.Q.-M, D.D., V.S.-A. and E.P. Methodology: V.S.-A., A. Biton and D.D. Software: V.S.-A. and A. Biton Validation: V.S.-A., A. Biton and D.D. Formal analysis: V.S.-A. and A. Biton. Investigation: V.S.-A., B.C., C.P., A. Bertrand and D.D. Resources: M.R., J.B., E.P., L.Q.-M. and D.D. Data curation: V.S.-A.,V.R., B.C., E.P. and D.D. Writing, original Draft: V.S.-A. and D.D. Writing, review and editing: V.S.-A, D.D., A. Biton, M.R., J.B., E.P. and L.Q.-M. Visualization: V.S.-A. and D.D. Supervision: D.D., L.Q.-M., V.S.-A. and E.P. Project administration: D.D. Funding acquisition: M.L.A., L.Q.-M., D.D. and the Milieu Intérieur Consortium.

**Competing interests** M.L.A. is an employee of Octant Biosciences, CA, USA. D.D. previously received grant support from Rules Based Medicine. The other authors declare no competing interests.

**Additional information**
**Correspondence and requests for materials** should be addressed to Violaine Saint-André or Darragh Duffy.

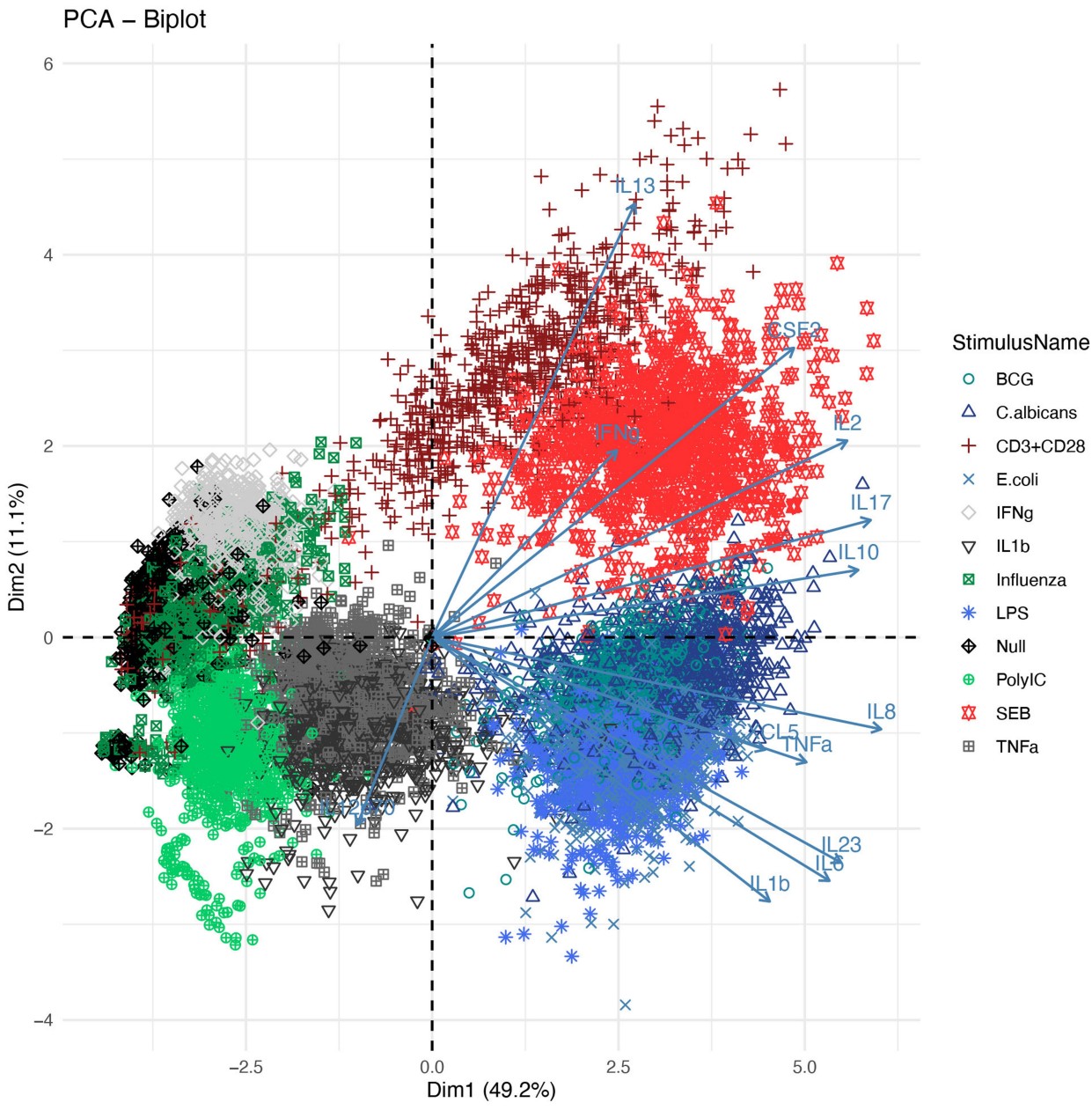

**Extended Data Fig. 1 | Principal component analysis (PCA) of individuals for the 13 cytokines in the 12 stimulation conditions.** Each dot represents one individual in a specified stimulation condition. Contributions of the cytokines to the first two dimensions are represented with arrows.

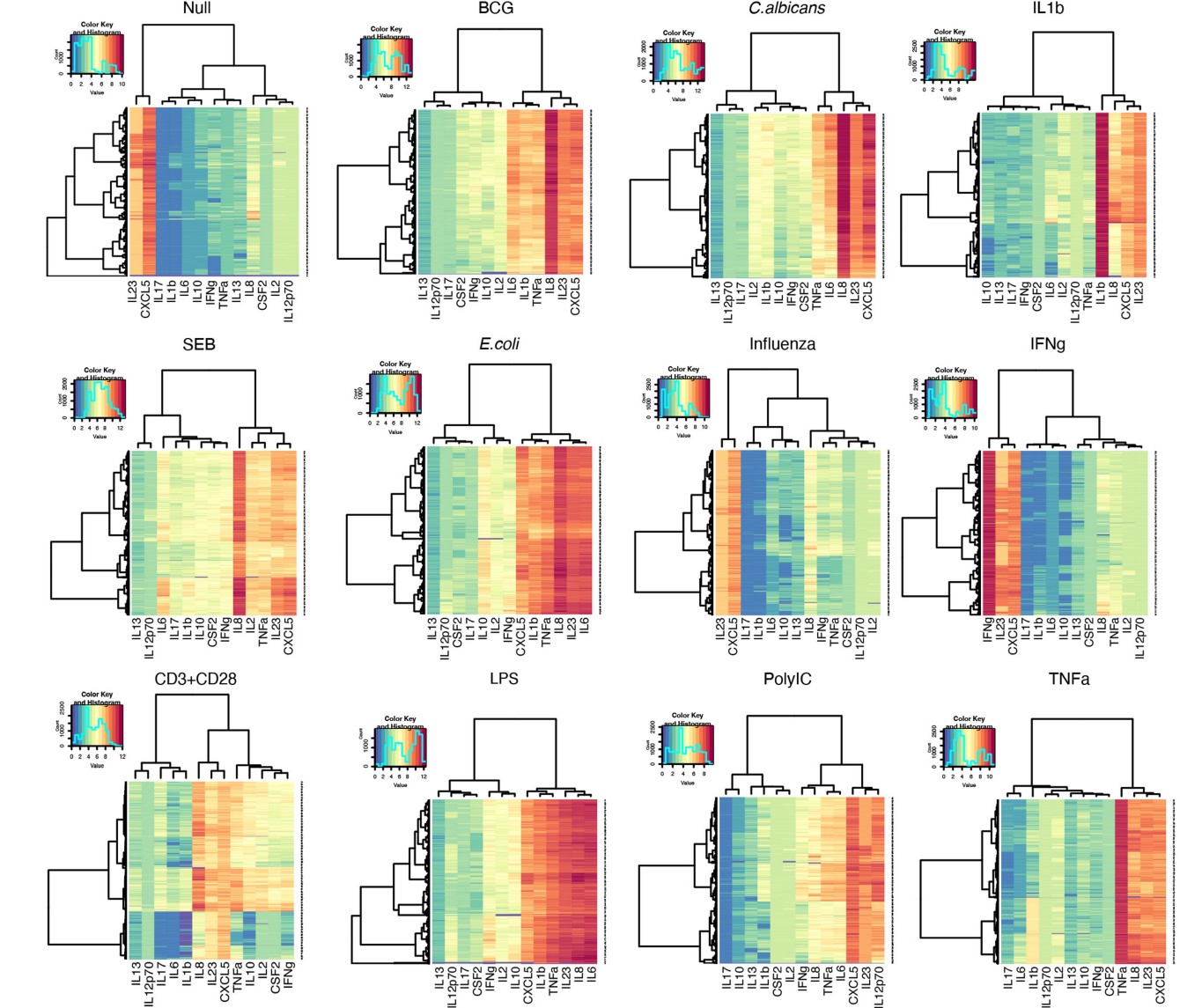

**Extended Data Fig. 2 | Heatmaps showing cytokine concentration levels in each stimulation condition for the 1,000 individuals of the Milieu Intérieur cohort.** Hierarchical clustering is performed with Ward method and euclidean distance on both lines and columns.

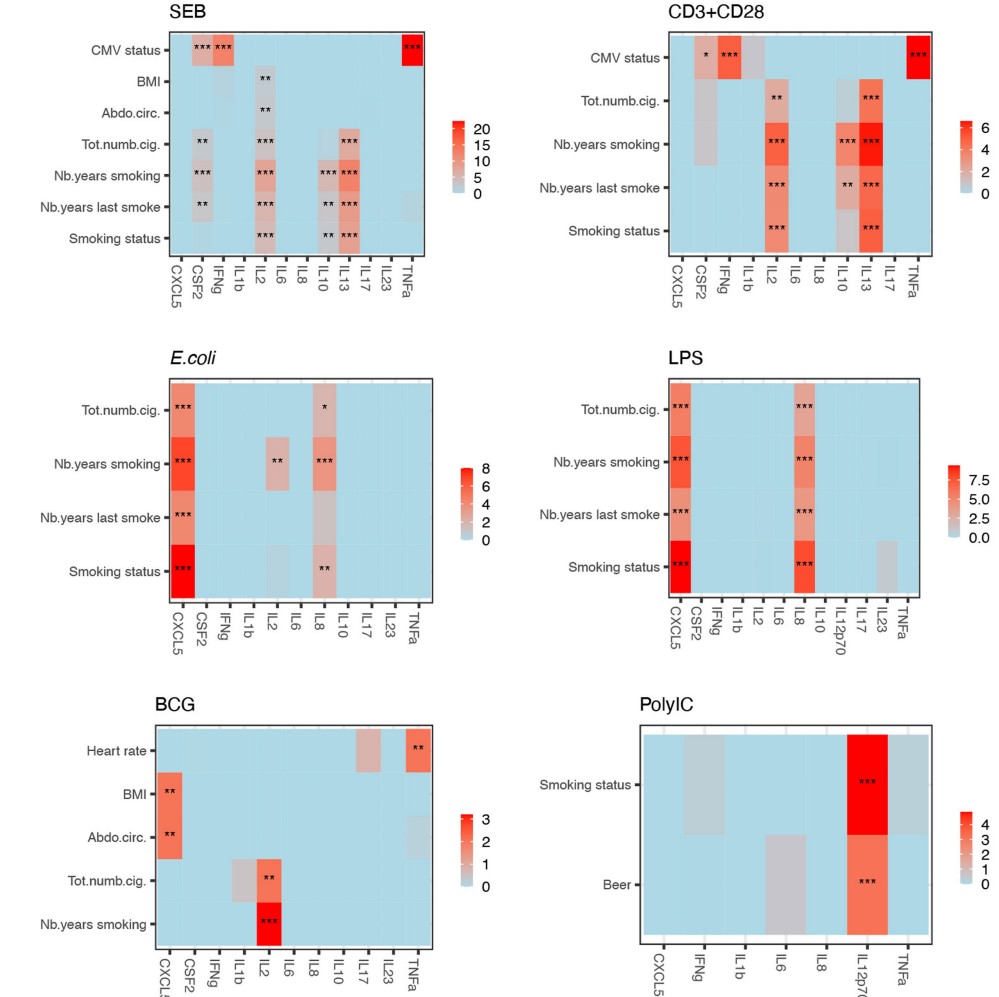

**Extended Data Fig. 3 | Heatmaps showing -log10(BY adj. pval of likelihood ratio test) of association for the eCRF variables associated with at least one cytokine in each stimulation considering smoking status and age** interactions in the compared models. These are coloured according to the colour key on the side of each heatmap and stars are shown depending on the strength of association (*P < 0.05; **P < 0.01; ***P < 0.001).

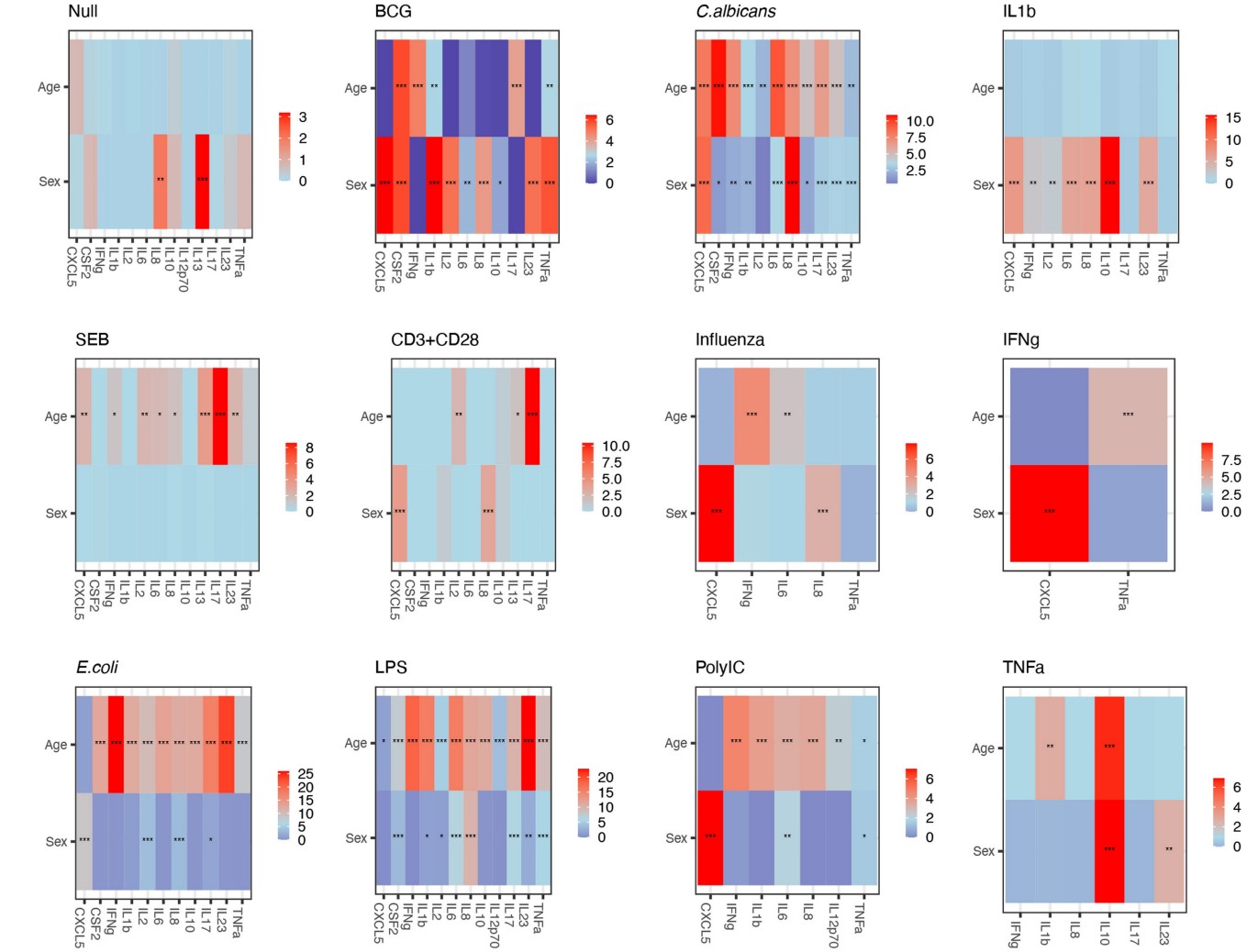

**Extended Data Fig. 4 | Heatmaps showing associations of age and sex variables corrected for sex or age respectively and for batchId, on induced cytokines in each stimulation condition.** Significance of the BY adj-pval likelihood ratio test is marked with stars (*P < 0.05; **P < 0.01; ***P < 0.001).

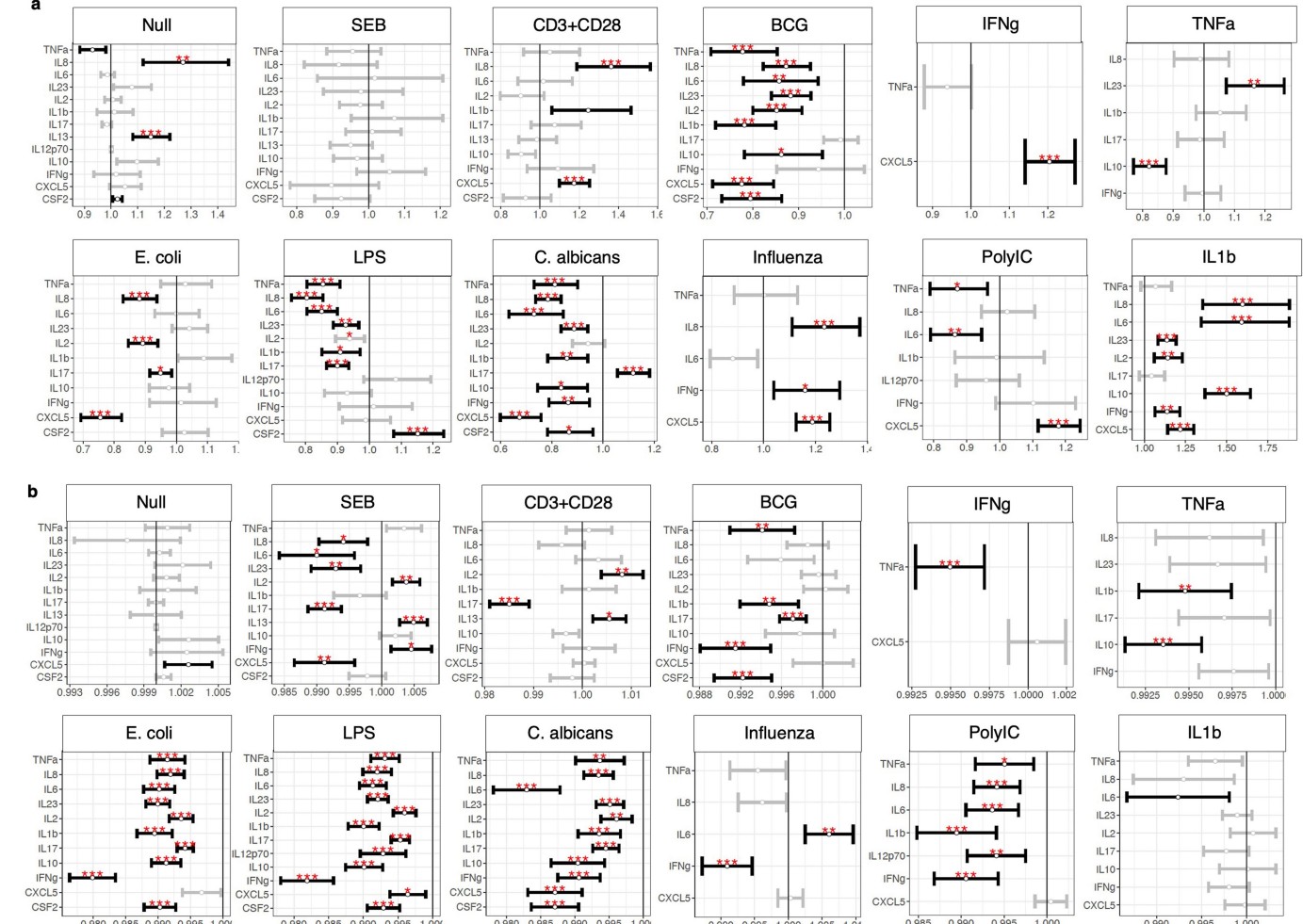

**Extended Data Fig. 5 | Effect of sex and age on on induced cytokines.**
Two-sided effect size plots made from $n$ = 955 independent individuals with
0.95 confidence interval for sex (**a**) and age (**b**) corrected for age or sex
respectively and batchId, on induced cytokines in each stimulation condition
for the 1,000 individuals of the Milieu Intérieur cohort. Significant effect sizes
are in black, others are in grey. Significance of the BY adj-pval likelihood ratio
test is marked with stars (*$P$< 0.05; **$P$< 0.01; ***$P$< 0.001).

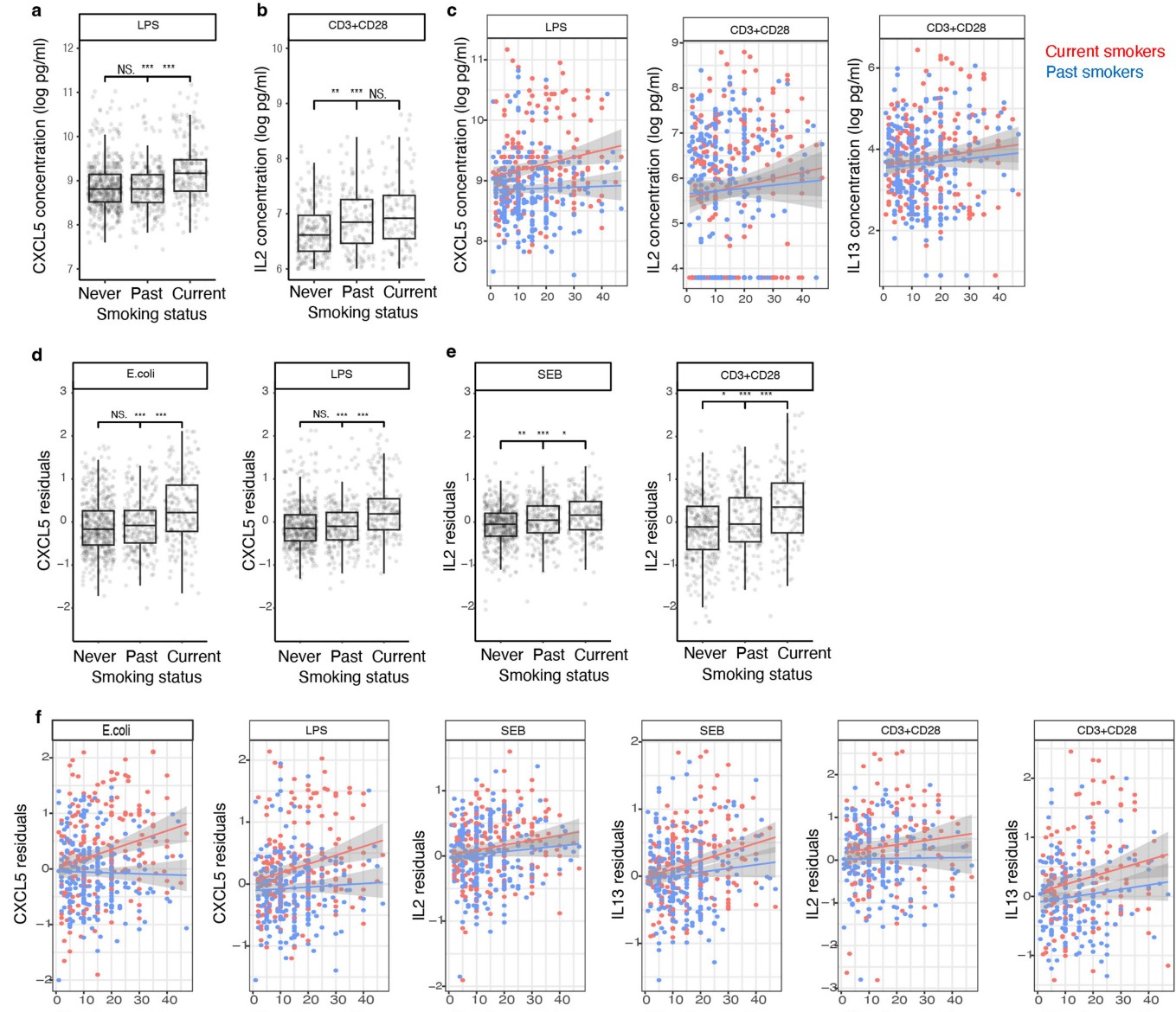

**Extended Data Fig. 6 | Smoking effect on innate and adaptive immune responses, represented by LPS and CD3 + CD28 stimulations respectively and same plots on residuals, after regression on age, sex and batchId.** **a,b,** CXCL5 concentration following LPS stimulation (**a**) and IL-2 concentration following anti-CD3 + CD28 stimulation (**b**) for never, past and current smokers. Boxplots represent *n* = 955 independent individuals. The centre line shows the median, hinges represent the 25th and 75th percentiles whiskers extend from the hinge to the largest or smallest value no further than 1.5 interquartile range. Significance of two-sided Wilcoxon rank sum tests adjusting for multiple comparisons is indicated with stars above the boxes on the left (between never and past smokers), in the middle (between never and current smokers) and on the right (between past and current smokers) (N.S, not significant; *P < 0.05, **P < 0.01, ***P < 0.001). P-values are from left to right, for CXCL5 in LPS: 0.74, 4.9e-13 and 4.7e-10 and for IL-2 in anti-CD3 + CD28: 3.8e-03, 2.4e-05 and 0.17. **c,** CXCL5 concentrations following LPS stimulation or IL-2 and IL-13 concentrations following anti-CD3 + CD28 stimulation versus numbers of years smoking for current smokers (red) or past smokers (blue). Grey areas depict the 0.95 confidence intervals of the linear regression lines. **d–f,** Residuals of CXCL5

following *E. coli* and LPS stimulations (**d**) or of IL-2 and IL-13 following SEB and anti-CD3 + CD28 stimulations (**e**) after regression on age, sex and batchId variables. Boxplots represent *n* = 955 independent individuals. The centre line shows the median, hinges represent the 25th and 75th percentiles, and whiskers extend from the hinge to the largest or smallest value no further than the 1.5 interquartile range. Significance of two-sided Wilcoxon Rank Sum tests adjusting for multiple comparisons is indicated with stars above the boxes on the left (between never and past smokers), in the middle (between never and current smokers) and on the right (between past and current smokers) (N.S, not significant; *P < 0.05, **P < 0.01, ***P < 0.001). P-values are from left to right, for CXCL5 in E.coli: 0.38, 7.1e-10 and 9.9e-07; for CXCL5 in LPS: 0.34, 7.5e-12 and 5.6e-08; for IL-2 in SEB: 1.4e-03,1.2e-07 and 4.2e-02 and for IL-2 in anti-CD3 + CD28: 0.026, 6.7e-09 and 9.8e-04. **f,** CXCL5 residuals following *E.coli* or LPS stimulation or IL-2 and IL-13 concentrations following SEB or anti-CD3 + CD28 stimulation versus the numbers of years smoking for current smokers (red) or past smokers (blue). Grey areas depict the 0.95 confidence intervals of the linear regression lines.

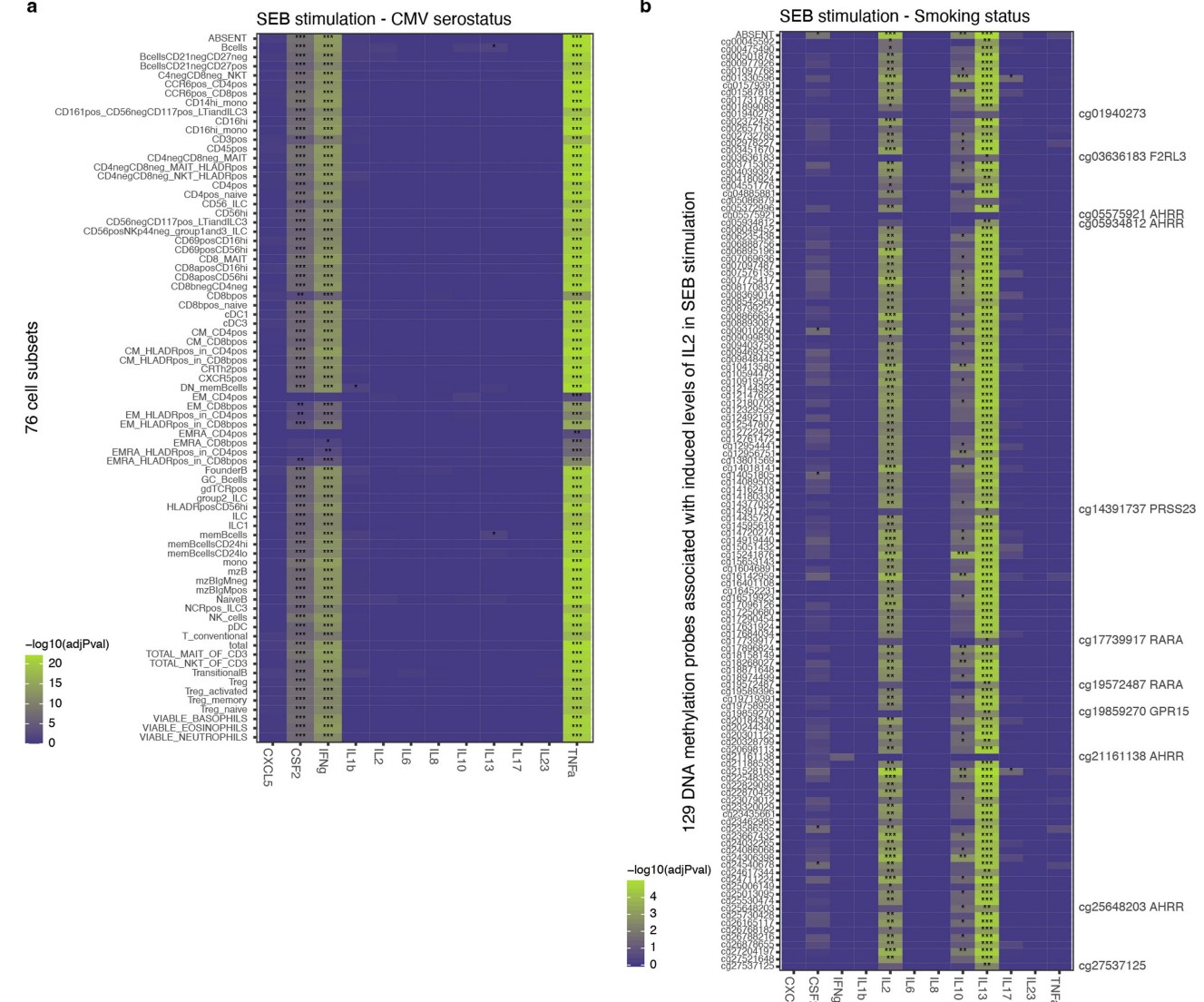

**Extended Data Fig. 7 | Effects of CMV status on induced cytokines is modified by blood cell subsets and effect of smoking on induced cytokines is modified by specific CpGs associated with induced levels of IL2 following SEB stimulation. a**, Heatmap showing associations of the cytomegalovirus (CMV) serostatus with each induced cytokine following SEB stimulation, with either no cellular covariate (top line) or each of 76 cell subsets passed as covariates in the models. Significance of the BY adj-pval likelihood ratio test

is marked with stars (*P < 0.05, **P < 0.01, ***P < 0.001). **b**, Heatmap showing associations of the smoking status with each induced cytokine following SEB stimulation when either no methylation probe (top line) or each individual methylation probe associated with IL-2 concentration levels in SEB stimulation is passed as a covariate in the models. Significance of the BY adj-pval likelihood ratio test is marked with stars (*P < 0.05, **P < 0.01, ***P < 0.001).

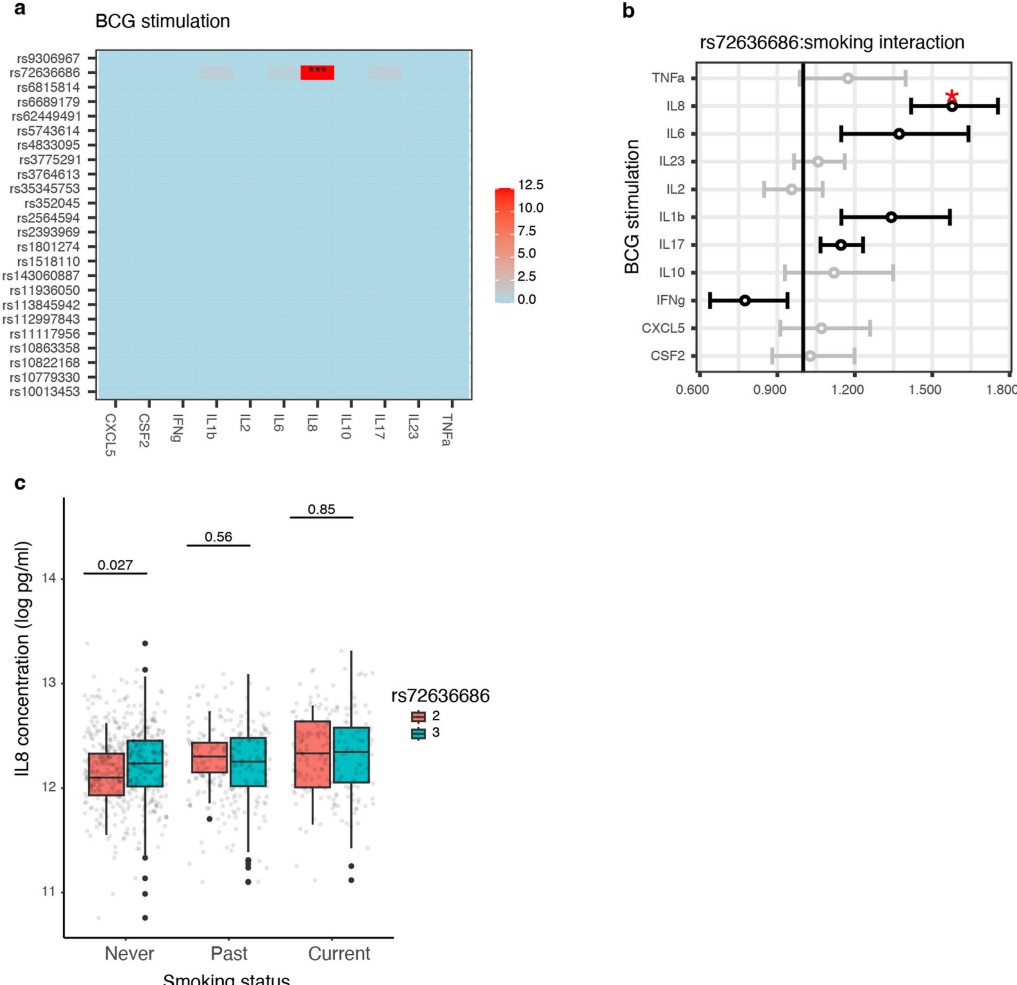

**Extended Data Fig. 8 | Smoking status and SNPs interactions. a**, Heatmaps showing -log10(BY adj.pval of likelihood ratio test) of interactions between genetic variants listed in Table 1 and smoking status for each induced cytokine following BCG stimulation considering age, sex and batchId as covariates. These are colored according to the color key on the side of the heatmap and stars are shown depending on the strength of association.*$P < 0.05$, **$P < 0.01$, ***$P < 0.001$. **b**, Two-sided effect size plot made representing n = 955 independent individuals with 0.95 confidence interval for the interaction of the SNP rs72636686 with the smoking status, corrected for age, sex and batchId, on induced cytokines following BCG stimulation. Significant effect sizes (p-val < 0.01) are in black, others are in grey. Those that also have BY adj.pval of the likelihood ratio test <0.01 are labelled with a red star. **c**, Boxplot representing *n* = 955 independent individuals for IL8 levels depending on the genotype for rs72636686 and the smoking status. The centre line shows the median, hinges represent the 25th and 75th percentiles, and whiskers extend from the hinge to the largest or smallest value no further than the 1.5 interquartile range. Significance of two-sided Wilcoxon rank sum tests adjusting for multiple comparisons are indicated above the compared data.

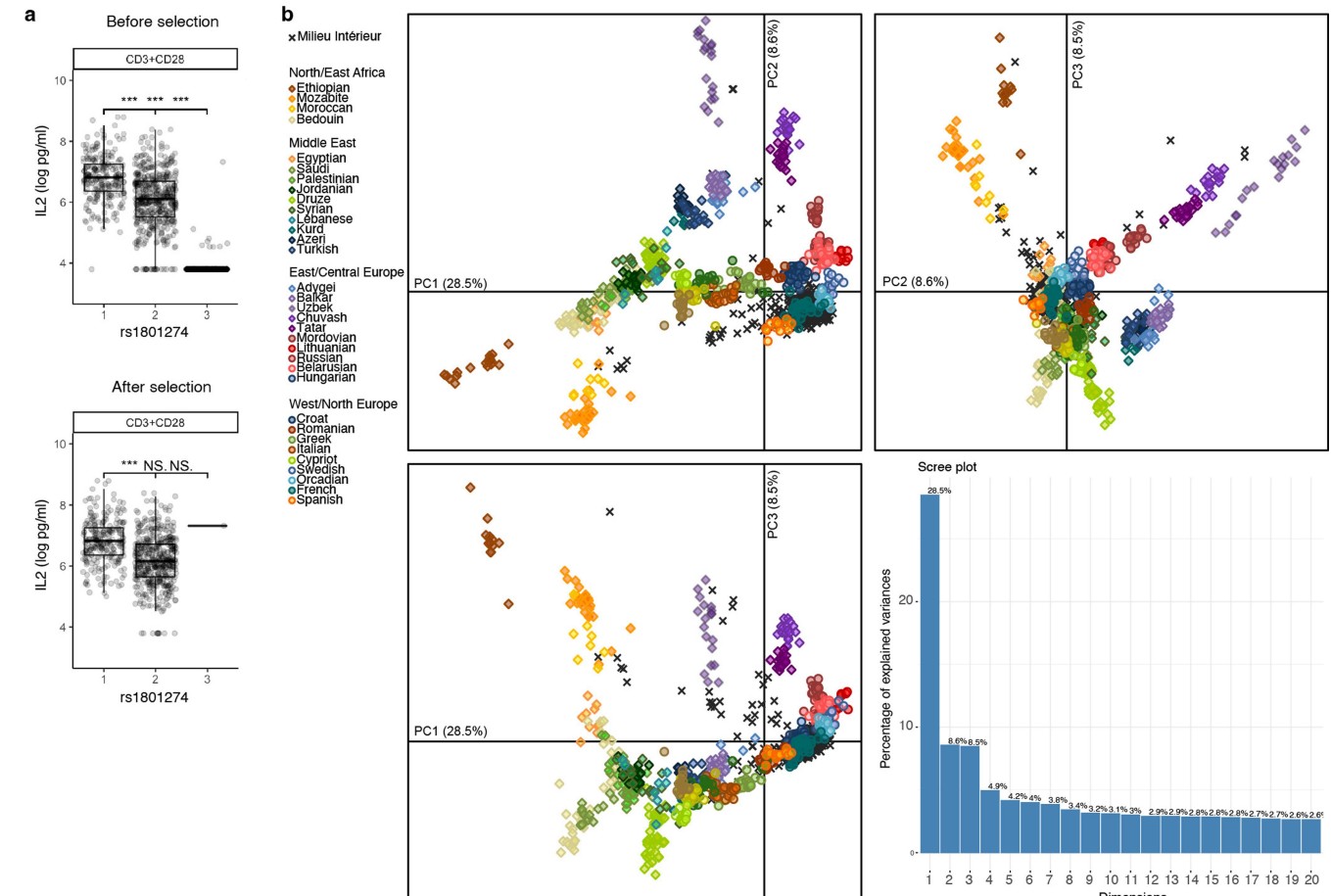

**Extended Data Fig. 9 | IL-2 protein levels depending on FcγRIIA polymorphism rs1801274 and genetic PCA showing homogeneity of the genetic background of Milieu Intérieur individuals. a**, Boxplots representing n = 1,000 independent individuals for IL-2 concentration levels depending on the genotype of the donors for the rs1801274 genetic variant before and after K-means clustering (2 groups) selection of responders. The centre line shows the median, hinges represent the 25th and 75th percentiles, and whiskers extend from the hinge to the largest or smallest value no further than the 1.5 interquartile range. Significance of two-sided Wilcoxon rank sum tests adjusting for multiple comparisons is indicated above the boxes (N.S, not significant; *P < 0.05, **P < 0.01, ***P < 0.001). P-values are from left to right on the top boxplot: <2.22e-16, <2.22e-16 and <2.22e-16 and on the bottom boxplot: <2.22e-16, 0.35 and 0.15. **b**, PCA performed on 261,827 independent SNPs and 1,723 individuals, which include the 1,000 MI donors together with 723 individuals from a selection of 36 populations from North Africa, the Near East, as well as western and northern Europe. PC1 versus PC2 (top left), PC1 versus PC3 (bottom left) and PC2 versus PC3 (top right) are displayed as well as a barplot (bottom right) of the variance explained by the first 20 components of the PCA.

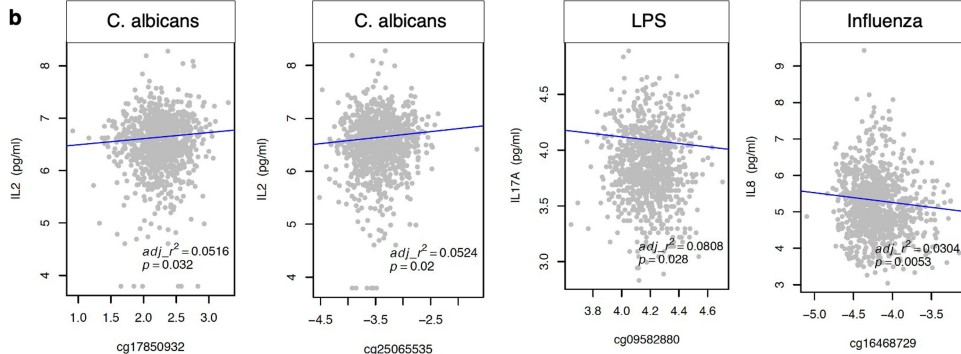

| Stimulus | mepQTL |
|---|---|
| LPS | IL17(cg09582880*) |
| C.albicans | IL2(cg17850932*, cg25065535*) |
| Influenza | IL8(cg16468729*) |

**Extended Data Fig. 10 | Association of DNA methylation probes with their cognate cytokines. a**, DNA methylation probes located within 1 Mb of the TSS of each cytokine that show associations (Benjamini-Hochberg false discovery rate adj-pval of likelihood ratio test <0.05) with the concentration levels of these cytokines in the specified stimulation condition (*$P$< 0.05). **b**, Plots showing the corresponding concentration levels depending on the methylation levels of the probes. Adj-$r^2$ values of the linear regressions and p-statistics of two-sided Pearson correlations are reported on each graph.

# Reporting Summary

## Statistics

For all statistical analyses, confirm that the following items are present in the figure legend, table legend, main text, or Methods section.

| n/a | Confirmed | |
|---|---|---|
| ☐ | ☒ | The exact sample size (*n*) for each experimental group/condition, given as a discrete number and unit of measurement |
| ☐ | ☒ | A statement on whether measurements were taken from distinct samples or whether the same sample was measured repeatedly |
| ☐ | ☒ | The statistical test(s) used AND whether they are one- or two-sided *Only common tests should be described solely by name; describe more complex techniques in the Methods section.* |
| ☐ | ☒ | A description of all covariates tested |
| ☐ | ☒ | A description of any assumptions or corrections, such as tests of normality and adjustment for multiple comparisons |
| ☐ | ☒ | A full description of the statistical parameters including central tendency (e.g. means) or other basic estimates (e.g. regression coefficient) AND variation (e.g. standard deviation) or associated estimates of uncertainty (e.g. confidence intervals) |
| ☐ | ☒ | For null hypothesis testing, the test statistic (e.g. *F*, *t*, *r*) with confidence intervals, effect sizes, degrees of freedom and *P* value noted *Give P values as exact values whenever suitable.* |
| ☒ | ☐ | For Bayesian analysis, information on the choice of priors and Markov chain Monte Carlo settings |
| ☒ | ☐ | For hierarchical and complex designs, identification of the appropriate level for tests and full reporting of outcomes |
| ☐ | ☒ | Estimates of effect sizes (e.g. Cohen's *d*, Pearson's *r*), indicating how they were calculated |

*Our web collection on statistics for biologists contains articles on many of the points above.*

## Software and code

Policy information about availability of computer code

| Data collection | Luminex xMAP technology software<br>FlowJo (Treestar)<br>Illumina Infinium MethylationEPIC BeadChip<br>Illumina HumanOmniExpress-24 and the HumanExome-12 BeadChips |
|---|---|
| Data analysis | R package relaimpo 2.2.3<br>R package ggplot2 3.2.1 (R 3.6.0)<br>MatrixEQTL 2.2 R package<br>heatmaply 1.0.0 and dendextend 1.13.12 with "complete" clustering method and "euclidien" distance  in (R 3.6.0)<br>R 4.2.1 using FactoMineR 2.8 |

For manuscripts utilizing custom algorithms or software that are central to the research but not yet described in published literature, software must be made available to editors and reviewers. We strongly encourage code deposition in a community repository (e.g. GitHub). See the Nature Portfolio guidelines for submitting code & software for further information.

# Data

Policy information about availability of data

All manuscripts must include a data availability statement. This statement should provide the following information, where applicable:
- Accession codes, unique identifiers, or web links for publicly available datasets
- A description of any restrictions on data availability
- For clinical datasets or third party data, please ensure that the statement adheres to our policy

SNP array data can be accessed from the European Genome-Phenome Archive EGA under accession EGAS00001002460, the Infinium MethylationEPIC raw intensity data can be accessed via the following link: https://dataset.owey.io/doi/10.48802/owey.f83a-1042, cytokine data is in Supplementary Data 1 and eCRF data is in Supplementary Data 2.

# Human research participants

Policy information about studies involving human research participants and Sex and Gender in Research.

| | |
|---|---|
| Reporting on sex and gender | The cohort was specifically designed to be equally balanced in terms of sex, 500 men and 500 women who were also stratified by age (20-69 years old). Sex (as opposed to gender) was confirmed by genetic chromosome analysis. For identification of environmental effects on cytokine responses we corrected for both age and sex. We also report specific sex effects on cytokine responses. |
| Population characteristics | The 1,000 donors of the Milieu Intérieur cohort were recruited by BioTrial (Rennes, France) to be composed of "healthy" individuals of the same genetic background (Western European) and to have 100 women and 100 men from each decade of life, between 20 and 69 years of age. Donors were selected based on various inclusion and exclusion criteria that were previously described (Thomas et al., 2015). Briefly, donors were required to have no history or evidence of severe/chronic/recurrent pathological conditions, neurological or psychiatric disorders, alcohol abuse, recent use of drugs, recent vaccine administration, and recent use of immune modulatory agents. To avoid the influence of hormonal fluctuations in women, pregnant and peri-menopausal women were not included. To avoid genetic stratification in the study population, the recruitment of donors was restricted to individuals whose parents and grandparents were born in Metropolitan France. |
| Recruitment | Human samples came from the Milieu Intérieur cohort, and were recruited by BioTrial (Rennes, France) during September 2011-July 2012. The study is sponsored by Institut Pasteur (Pasteur ID-RCB Number: 2012-A00238-35) and was conducted as a single center interventional study without an investigational product. The original protocol was registered under ClinicalTrials.gov (study# NCT01699893). The samples and data used in this study were formally established as the Milieu Intérieur biocollection (NCT03905993), with approvals by the Comité de Protection des Personnes – Sud Méditerranée and the Commission Nationale de l'Informatique et des Libertés (CNIL) on April 11, 2018. Donors were defined as healthy based on inclusion/exlcusion criteria previously described in Thomas et al, Clinical Immunology 2015. |
| Ethics oversight | The study was approved by Comité de Protection des Personnes – Ouest 6 on June 13th, 2012, and by the French Agence Nationale de Sécurité du Médicament (ANSM) on June 22nd, 2012. |

Note that full information on the approval of the study protocol must also be provided in the manuscript.

# Field-specific reporting

Please select the one below that is the best fit for your research. If you are not sure, read the appropriate sections before making your selection.

☒ Life sciences  ☐ Behavioural & social sciences  ☐ Ecological, evolutionary & environmental sciences

For a reference copy of the document with all sections, see nature.com/documents/nr-reporting-summary-flat.pdf

# Life sciences study design

All studies must disclose on these points even when the disclosure is negative.

| | |
|---|---|
| Sample size | The original sample size of the Milieu Interieur cohort (n=1000) was calculated to have sufficient power for strong to intermediate genetic effects. For this particular study no specific sample size was calculated and all available donors were included in each analysis step. Unless otherwise stated, all displayed results have been performed on the 956 individuals of the cohort who gave consent to share their data publicly in order to ensure easy reproducibility of the results |
| Data exclusions | For the CD3+CD28 stimulation, we identified through k-means clustering a group of 705 individuals that responded to the stimulation and a group of 295 individual did not. This lack of response of 295 individuals is explained by a FcγRIIA polymorphism (rs1801274) that was previously described as preventing response to this CD3+CD28 stimulation (Stein et al., Genes and Immunity, 2018) (Extended Data Fig. 9). All statistical analyses on CD3+CD28 stimulations in this study were thus performed on the 705 responders only. |
| Replication | Due to the specific nature of the cohort and associated data sets replication has not been performed. |

| Randomization | Samples were completely randomized prior to generation of cytokine data sets |
|---|---|
| Blinding | The generation of cytokine data sets was blinded as to the stimuli condition and donor ID |

# Reporting for specific materials, systems and methods

We require information from authors about some types of materials, experimental systems and methods used in many studies. Here, indicate whether each material, system or method listed is relevant to your study. If you are not sure if a list item applies to your research, read the appropriate section before selecting a response.

## Materials & experimental systems

| n/a | Involved in the study |
|---|---|
| ☐ | ☒ Antibodies |
| ☒ | ☐ Eukaryotic cell lines |
| ☒ | ☐ Palaeontology and archaeology |
| ☒ | ☐ Animals and other organisms |
| ☐ | ☒ Clinical data |
| ☒ | ☐ Dual use research of concern |

## Methods

| n/a | Involved in the study |
|---|---|
| ☒ | ☐ ChIP-seq |
| ☒ | ☐ Flow cytometry |
| ☒ | ☐ MRI-based neuroimaging |

## Antibodies

| Antibodies used | Supernatants from TruCulture tubes were analyzed by Rules Based Medicine (Austin, Texas, US) using the Luminex xMAP technology. Samples were analyzed according to the Clinical Laboratory Improvement Amendments (CLIA) guidelines. The lower limit of quantification (LLOQ) was determined as previously described (Duffy et al., Immunity 2014), and is the lowest concentration of an analyte in a sample that can be reliably detected and at which the total error meets CLIA requirements for laboratory accuracy. Information on the antibody clones used in the assay is the propriety of Rules Based Medicine. |
|---|---|
| Validation | Validation of each antibody pair used in the Luminex assay has been performed by Rules Based Medicine in respect of CLIA guidlines and testing for specificity, sensitivity and cross-reactivity. |

## Clinical data

Policy information about clinical studies

All manuscripts should comply with the ICMJE guidelines for publication of clinical research and a completed CONSORT checklist must be included with all submissions.

| Clinical trial registration | NCT01699893, NCT03905993 |
|---|---|
| Study protocol | Clinical trials.gov : NCT01699893, NCT03905993 |
| Data collection | Donors were recruited by BioTrial (Rennes, France) during September 2011-July 2012. Data has been generated from these samples since 2012. |
| Outcomes | NA |

