## [Peer Review File · Nature]

Manuscript Title: Smoking changes adaptive immunity with persistent effects

Reviewer Comments & Author Rebuttals

Reviewer Reports on the Initial Version:

Referees' comments:

Referee #1 (Remarks to the Author):

In this article, the authors subject blood from 1,000 healthy donors to immune stimulation directed towards a range of innate and adaptive responses, and assess cytokine responses. They complement this work with methylation and genomic profiling to present an impressively comprehensive assessment of environmental factors affecting the immune response. Most interestingly, they are able to dissect out the effects of smoking on innate and adaptive immune responses, showing short-term reduction in innate immunity in current smokers and longer-term reduction in adaptive immunity in ex-smokers. This has clear clinical implications for the risk of developing infections, and cancers.

I would highly commend the authors for data transparency, providing as they do full experimental data in their supplementary information, as well as genomic and methylation data, and analysis code. I would personally prefer to see the methylation data in an open repository such as GEO rather than the authors' institutional repository, but I understand they may be constrained by prior agreements.

It would be helpful if the authors could expand on the associations between smoking and age. This cohort is designed with 100 men and 100 women in each decade of life. With a wide spectrum of ages represented we would expect years smoking and total cigarettes smoked to strongly correlate with age. The patterns of expression shown for smoking (Figure 2) and age (extended data figure 4) do look fairly similar. It does appear that the authors are using multivariate linear models adjusted for age, but are not taking the same approach of adding covariates as they did with, for example, CEACAM6. Could they expand a little on this and justify that the smoking analyses presented are all appropriately corrected for age?

The authors present a detailed analysis of genomic and epigenetic array-based data. They do not mention HLA, which is perhaps the best known determinant of immune response variability. Do they have these data (or are they able to get it)? If not, I believe tools exist to infer HLA type from SNP

arrays. Even in this “genomically homogenous” cohort I would expect some variation in HLA, and would not be surprised if it were a major factor in the immune response.

Regarding CEACAM6, the authors note that this is expressed on immune cells including neutrophils and macrophages. When they say the effects negate the changes in CXCL5 after innate immune stimulation, are they confident that this is a genuine regulatory effect rather than a shift in immune cell populations?

There is a substantial body of literature examining the effects of smoking on DNA methylation. Have the authors validated their smoking-associated DNA methylation markers in external datasets? Regarding the selection of 11 CpGs which eliminate IL2 and IL13 effects, can the authors comment on multiple testing correction? I understand that their 850k methylation data was reduced to 129 CpGs using appropriate correction (BY method) but I am not clear that any correction was applied to reduce the 129 to 11 which eliminate the effects of smoking on immune stimulation?

I don't quite know what to make of the authors' genomic eQTL analysis. No insights from this are mentioned in the Discussion. How would the authors describe the impact of this work? Do they see this leading to identifying people at risk of a reduced immune response based on genotyping and attempting to intervene therapeutically? What would be the roadmap to this? This is a lot of work and it would be good to make the impact clearer to the reader.

Referee #2 (Remarks to the Author):

Summary of the key results

In the submitted work, Saint-André et al. investigated the relationships between cytokine responses and immune stimulations with respect to environmental factors in the *Milieu Intérieur* (MI) project. Even though the title only stated the effects of smoking on host immune response, the authors investigated and nominated a wide range of variables, such as age, sex, genetic background and CMV infection status, and their impact on cytokine responses.

Originality and significance: if not novel, please include reference

I don't know the literature in the field of cytokine responses with respect to environmental factors well. But a brief search suggested that previous studies have shown smoking alters methylation as well as cytokine responses. This has been shown in population cohorts such as UKB (Amador et al. 2021), TwinsUK cohort (Christiansen et al. 2021) and Lothian Birth Cohort 1936 (Corley et al. 2019). In the abstract, the authors stated “*Our findings describe **new factors** associated with cytokine*

secretion variability, identify a **new role** for smoking on immune response regulation. Please can the authors describe exactly what new factors and new role for smoking on immune response regulation did they identify. As far as I can read:

- No new data was generated for the MI project.
- No new methods were introduced for analyzing the dataset apart from looking at different environmental factors that were previously measured in the study.
- No novel biological mechanisms were identified through the study, only confirmation of previous findings.

Data & methodology: validity of approach, quality of data, quality of presentation

The main text is clearly written, but data presentation and method description have room for improvement with more details and consistency (suggested improvements listed below).

- Line 51-53 could be better written. I couldn't find anything describing adjusting genetic factors in their Method or justify the statement of "a homogenous genetic background". Are these individuals genotyped? If yes, a genetic PC would help to justify the statement.
- It is unclear in various places what regression model was used. For example, Line 100-101 compared the effect sizes of smoking status to age and sex, but did not provide details of the regression models used in Extended Data Fig. 3 and 4. For sex, is it $\text{lm}(\text{cytokine} \sim \text{sex} + \text{batchID})$ or $\text{lm}(\text{cytokine} \sim \text{age} + \text{sex} + \text{batchID})$? And how can the authors compare effect sizes when one is adjusted for age and sex $\text{lm}(\text{cytokine} \sim \text{sex} + \text{age} + \text{batchID} + \text{smoking})$ or does Figure 2 show $\text{lm}(\text{cytokine} \sim \text{smoking} + \text{batchID})$. In either case more clarity is needed.
- The authors need to describe how they define innate versus adaptive stimulation and which stimulus belongs to which.
- The quality control steps for processing the methylation steps are not described. This is important as methylation arrays can be noisy and subject to batch effects. If it is the same as Bergstedt et al. 2022, it also needs to be mentioned.
- The authors used the term "factor" for different variables (genetic factors, environmental factors, blood factors, etc). I think they all mean slightly different things. Did the authors perform a factor analysis to define the "factors" in the clinical variable association analysis? It would be better to use more precise words to describe them for clarity. E.g. instead of "blood factors" just call them proteins.
- It is unclear which scripts in the provided GitHub repository are related to this project.

Appropriate use of statistics and treatment of uncertainties

I have a few questions/suggestions regarding the statistics used in this manuscript:

- The authors used BY-adjust p-value to nominate significant associations, but this also needs to be adjusted for multiple-testing among 13 cytokines too. It would also be beneficial to state the actual BY-adjust p-value for the readers to understand the level of dependencies for all the environmental variables considered.
- A supplementary table needs to be provided for all reported associations presented in the main figures and in the text.

The principal component analysis needs to be described in the Method section. Are each cytokines standardized similar to what they did in the heatmap analysis?

- The authors seem to use different models for claiming significance in their heatmap and effect size

analysis. The statistical tests should be the same in these two presentations.

- Are the p-values presented in the boxplots also multiple-testing corrected and adjusted for age, sex and batch?
- There is no discussion of how genetic background will affect the results. Considering the authors have stated that genetic background does make a difference in cytokine responses, genetic PCs should be included as covariates in their linear models.
- Genome-wide PCs should also be included as covariates in the pQTL analysis.

Conclusions: robustness, validity, reliability

As the authors mentioned in the conclusion, there is no replication cohort in their study and it is currently achievable with population cohorts such as UK Biobank.

Suggested improvements: experiments, data for possible revision

Major

- Replicate main findings in an independent population cohort.
- A conditional analysis should be undertaken for the various smoking related phenotypes e.g. current smoker, previous smoker, number of years smoked, number of cigarettes smoked a day etc. as it is unclear whether these variables are correlated and which is the primary signal of association. This is important as it has different biological implications.
- Since the smoking effect on cytokine is the main focus, can the authors perform a further conditional analysis with $\text{lm}(\text{cytokine} \sim \text{age} + \text{sex} + \text{batchID} + \text{CMV} + \text{all other non-smoking related factors})$ and report if the smoking has an independent effect from all other environmental factors.
- Since genotyping information is available for this cohort, it would be good to see a SNP x tobacco interaction analysis, similar to Pisecka et al.
- The description of **novel** pQTL signals is inaccurate. A search on Open Targets suggested that rs35045 was previously reported in Sun et al. 2018. Furthermore, what are the LD relationships between the reported novel pQTLs in relationship with the previously top reported pQTL SNP in respect to the same protein. The authors need to perform a conditional analysis to convince the readers they are indeed independent novel regulatory SNPs compared to previously reported pQTL signals. For example, the claimed “new pQTL” for IL6 (rs35345753) is in LD with the lead pQTL SNP (rs2069840) reported in Pietzner*, Wheeler* et al., 2021 ($r^2 = 0.2$).

Minor

- Extended Data Fig.1 would be more clear if the authors chose a better color scheme among 4 different stimulation categories (different shades of red, blue, black and green say).
- It would be helpful to see separate PCA within each stimulations, and color them by age and sex. Based on the Extended Data Fig. 1, there seems to be a clear stratification in PolyIC, Influenza and Null, are they separated by smoking status or any other non-biological factors, e.g. batch, season?
- Fig. 1 has a missing legend for panel c. The color scale seems to be different for each stimulation panel. This should be unified with an added color legend.
- Don't really understand the grouping scheme for environmental factors. It is unclear to me why “cooked meals” and “heart rate” belong to the same group.
- The SI Tables are missing legends.
- Extended Data Fig.2 has confusing names for the cytokines (e.g. GM-CSF and ena_78).

- The formatting of Table 1 could be improved, with respect to the tables and exponentials.

References: appropriate credit to previous work?

- The results of CEACAM6 - CXCL5 association in smokers should be described more and reference earlier studies if it is already known.
- Similarly, for the methylation analysis, the authors need to describe what is known and reference previous studies.
- For the pQTL analysis, the authors need to describe why they think their results are novel.

Clarity and context: lucidity of abstract/summary, appropriateness of abstract, introduction and conclusions

As I mentioned before, I feel the authors need to describe better what is novel in the present work. Current descriptions of “new factors” and “new roles” are too vague.

Referee #3 (Remarks to the Author):

This is an extremely detailed and well executed study of the effects of various parameters on the outcome of cigarette smoking cessation, particularly with regards to the immune system.

The positives of this study include the cohort of 1000 patients that are matched with regards to age, sex and ethnicity. This cohort, developed by the Milieu Intérieur (MI) project, has previously been interrogated for variations in immune homeostasis, with respect to age, sex, cytomegalovirus (CMV) latent infection and smoking.

Another positive is the ability to link with sociodemographic and clinical details etc. The current study is comprehensive, examining the production of 13 cytokines in response to 12 stimulants that trigger responses from innate and adaptive immune cells. In essence they reveal effects of BMI, age and smoking on cytokines from whole blood cultures. The effect depends on the stimulus. They then further show that smoking cessation restores innate immune differences but that some aspects of adaptive immunity do not revert (dependent on pack years) and that this may be due to epigenetic modifications.

There are no technical deficiencies in the reported outcomes. However, it is felt that the conclusions are not altogether surprising when considering prior publications in the same field.

Mario Bauer et al for example published in Epigenetics in 2016 that “Tobacco smoking differently influences cell types of the innate and adaptive immune system-indications from CpG site methylation”

Giulia Piaggieschi et al published in Front Immunol in 2021 “Immune Trait Shifts in Association With

Tobacco Smoking: A Study in Healthy Women”

There are many other examples. It is likely that multiple factors acting in concert explain immune heterogeneity in health and disease, which is supported by the observation in the current study that smoking only explains between 4 and 9 percent of inter-individual variance.

One specific point is that the authors should be careful categorising stimulants as “innate” or “adaptive” as lymphocytes for example can also express innate receptors and vice versa.

Author Rebuttals to Initial Comments:

We acknowledge and thank the reviewers for their positive assessment of our work and constructive criticism that have helped us improve our manuscript. Please find below a point by point answers to their questions and comments.

Referee expertise:

Referee #1: Genetics, smoking, clinical

Referee #2: Genetics, disease

Referee #3: Respiratory immunology

Referees' comments:

Referee #1 (Remarks to the Author):

In this article, the authors subject blood from 1,000 healthy donors to immune stimulation directed towards a range of innate and adaptive responses, and assess cytokine responses. They complement this work with methylation and genomic profiling to present an impressively comprehensive assessment of environmental factors affecting the immune response. Most interestingly, they are able to dissect out the effects of smoking on innate and adaptive immune responses, showing short-term reduction in innate immunity in current smokers and longer-term reduction in adaptive immunity in ex-smokers. This has clear clinical implications for the risk of developing infections, and cancers.

We thank the reviewer for their positive assessment of our article recognizing the high impact of our results and its clear clinical implications for risk of developing infections and cancer.

I would highly commend the authors for data transparency, providing as they do full experimental data in their supplementary information, as well as genomic and methylation data, and analysis code. I would personally prefer to see the methylation data in an open repository such as GEO rather than the authors' institutional repository, but I understand they may be constrained by prior agreements.

We appreciate the reviewer commending our efforts at data transparency as a major goal of our work is to make our datasets accessible to the scientific community. However, as also hinted by the reviewer, we have to balance our desire to share data openly with both the restrictions of our donor consent and local and European regulations including GDPR. Indeed, according to our institutional legal team, DNA methylation data is considered as identifiable data, because it includes genetic data (59 SNPs). Therefore, GEO is no longer permitted by French legislation due to geographical location in the USA of stored personal data. It is for this reason that the methylation data is available through

our institute repository, and in fact in our experience this improves and speeds up data access with respect to the European Genome-Phenome Archive (EGA), which in the past we have used for sharing SNP data. In both cases, data requests must be submitted to our Data Access Committee (DAC); however, having the data shared through our institutional repository removes an additional layer of administration that comes with use of EGA. Approval by the DAC is a requirement that ensures that the research participants are informed about all use of their data and that such usage is consistent with the informed consent signed by the participants (GDPR requirements). The major new dataset generated from this study (the cytokine dataset of 12 whole blood stimulations), which are not identifiable, as well as the table reporting the 136 variables related to the donors of the cohort are included as supplementary data files, which is the simplest way to make this data accessible to other researchers interested in the work. The code to reproduce the work has been made public through GitHub under GPL3 open-source license and tables of values corresponding to the main figures have been now added as source data files.

It would be helpful if the authors could expand on the associations between smoking and age. This cohort is designed with 100 men and 100 women in each decade of life. With a wide spectrum of ages represented we would expect years smoking and total cigarettes smoked to strongly correlate with age. The patterns of expression shown for smoking (Figure 2) and age (extended data figure 4) do look fairly similar. It does appear that the authors are using multivariate linear models adjusted for age, but are not taking the same approach of adding covariates as they did with, for example, CEACAM6. Could they expand a little on this and justify that the smoking analyses presented are all appropriately corrected for age?

The reviewer is correct in that there are some smoking-related variables correlations with age. As we know from previous studies that age and sex are often associated with variable immune responses, we did correct for these effects, by passing them as covariates in the models. We followed the same approach for testing whether the smoking status was associated with cytokine levels while considering the effect of age, as the one that we used to test whether smoking was associated with cytokines considering the number of cells. The only difference is that we did not include possible interactions: we compared the model $\text{lm}(\text{cytokine} \sim \text{tested variable} + \text{age} + \text{sex} + \text{batchID})$ with the model $\text{lm}(\text{cytokine} \sim \text{age} + \text{sex} + \text{batchID})$ with a Likelihood Ratio Test (LRT). Following the reviewer's concerns, we have now performed additional analysis considering not only the age variable but also the interaction of tested variables with the age variable in our models (New Extended Data Fig.3). Namely, we have compared the model $\text{lm}(\text{cytokine} \sim \text{tested variable} * \text{age} + \text{sex} + \text{batchID})$ with the model $\text{lm}(\text{cytokine} \sim \text{age} + \text{sex} + \text{batchID})$ with a LRT. The results are very similar to the ones obtained without considering interactions, and some variables are associated with even higher significance (e.g., total number of cigarettes smoked, number of years smoking, number of years since last smoke) in SEB, CD3+CD28, *E. coli* and LPS stimulations with respect to what we observed previously. Interestingly, by including these interactions, smoking-related variables become significantly associated with IL2 responses in the BCG stimulation. This, we believe, represents further validation of our results, as BCG-induced IL2 may reflect a long-lived antigen specific T cell response to BCG vaccination, which the cohort received at birth due to mandatory vaccination in France prior to 2007. We now show this analysis in Extended Data Figure

3 and have made corresponding additions to the main text: *“As potential interactions may exist between our tested variables and age, we performed the same analysis considering age and smoking interactions in the models. The results are very similar to the ones obtained without considering interactions, and some smoking-related variables are associated with even higher significance in SEB, CD3+CD28, E. coli and LPS stimulation conditions (Extended Data Fig. 3). Interestingly, by including these interactions, smoking-related variables are significantly associated with IL2 responses after BCG stimulation. This IL2 response may reflect a long-lived antigen-specific T cell response to BCG vaccination, which all of the cohort received at birth, further strengthening the smoking associations with T cell immunity.”* We have also re-performed Fig. 1c on the 956 individuals of the cohort who gave consent to share their data publicly in order to ensure easy reproducibility of our results (the previous analysis was performed on the 1,000 donors for which data cannot be entirely shared publicly) and improved the layout of Fig. 1b and 1c. Accordingly, we have added in the methods: *“Unless otherwise stated, all displayed results have been performed on the 956 individuals of the cohort who gave consent to share their data publicly in order to ensure easy reproducibility of the results”*. Overall, we think these combined changes have improved the interpretability of our results and the identification of a BCG induced IL2 smoking association has further strengthened the novelty of our main findings.

Extended Data Fig.3: Heatmaps showing $-\log_{10}(\text{BY adj. pval})$ of association for the eCRF variables associated with at least one cytokine in each stimulation considering smoking status and age interactions in the compared models. These are colored according to the color key on the side of each heatmap and stars are shown depending on the strength of association (* < 0.05; ** < 0.01; *** < 0.001).

The authors present a detailed analysis of genomic and epigenetic array-based data. They do not mention HLA, which is perhaps the best known determinant of immune response variability. Do they have these data (or are they able to get it)? If not, I believe tools exist to infer HLA type from SNP arrays. Even in this “genomically homogenous” cohort I would expect some variation in HLA, and would not be surprised if it were a major factor in the immune response.

We thank the reviewer for this important point. While the reviewer is correct that HLA is a very well-known determinant of immune response variability, it is mostly relevant for antigen specific responses, which is not our major focus here. However, we obtained HLA data, which was inferred from the SNP array data used to genotype this cohort (previously described in Patin et al., 2018).

Following the reviewer's advice, we have tested for associations between HLA types (for those that are present at > 5% frequency in the MI cohort) and induced cytokines responses, following the same procedure that we used for the other donor variables. By doing so, we detected only one significant association, which was between the major histocompatibility complex, class II, DQ beta 1 "HLA.DBQ1.1P" and IL6 in the non-stimulated condition (Figure 1 for reviewers). This is an interesting observation but do not affect the main results of our work which is focused on induced immune responses. However, we have added these lines to the results to highlight that we have tested this point: *"In addition, as Human Leukocyte Antigen (HLA) is a well-known determinant of immune response variability, mostly relevant for antigen-specific responses, we tested associations between previously identified HLA types⁸ and induced cytokine responses following the same procedure that we used for the other donor variables. By doing so, we detected only one significant association, which was between the major histocompatibility complex, class II, DQ beta 1 "HLA.DBQ1.1P" and IL6 in the non-stimulated control condition. No associations were observed with induced cytokine responses after stimulation."*

Figure 1 for reviewers: Heatmaps showing $-\log_{10}(\text{BY adj.pval of LRT})$ of associations between the HLA variables and induced cytokines considering age, sex and batchId as covariates. These are colored according to the color key on the side of each heatmap and stars are shown depending on the strength of association ($* < 0.05$; $** < 0.01$; $*** < 0.001$).

Regarding CEACAM6, the authors note that this is expressed on immune cells including neutrophils and macrophages. When they say the effects negate the changes in CXCL5 after innate immune

stimulation, are they confident that this is a genuine regulatory effect rather than a shift in immune cell populations?

We thank the reviewer for this interesting question. Given that our stimulation period is 22 hours, we think it is too short for shifts in immune cell populations. Furthermore, the regression models that included cell numbers as covariates (Fig.3a) did not identify any associations between variation in cell numbers and the bacteria-induced CXCL5 smoking association. For these two reasons, we think that the CEACAM6 effect is genuinely related to the biology of immune cells rather than a shift in immune cell populations. We have added details to highlight that point in the main text: “We interpret this effect as a biological interaction rather than a shift in immune cell populations, due to the short stimulation period of 22 hours and that regression analysis including cell numbers as covariates (Fig.3a) did not identify any cell number associations with the bacteria-induced CXCL5 smoking association.”

There is a substantial body of literature examining the effects of smoking on DNA methylation. Have the authors validated their smoking-associated DNA methylation markers in external datasets? Regarding the selection of 11 CpGs which eliminate IL2 and IL13 effects, can the authors comment on multiple testing correction? I understand that their 850k methylation data was reduced to 129 CpGs using appropriate correction (BY method) but I am not clear that any correction was applied to reduce the 129 to 11 which eliminate the effects of smoking on immune stimulation?

Regarding the selection of 11 CpGs that eliminate IL2 and IL13 effects, we first identified 2,416 CpG sites to be directly (i.e., not acting through cellular count differences) associated with smoking status while correcting for multiple testing. From this reduced list, 129 CpG sites were identified to be significantly associated with IL2 levels after SEB stimulation (BY adjusted p value of $LRT < 0.001$). The association tests of the smoking status with the induced cytokines considering each of these 129 probes were also BY corrected for all the tests corresponding to the whole heatmap shown in Extended Data Figure 7b (BY adjusted p-value of $LRT < 0.001$). We have clarified that by adding “(BY adj.pval of $LRT < 0.001$)” in the sentence: “We observed that 11 CpGs, when passed as covariates in the models, eliminate the association of smoking with IL2 and IL13 (BY adj.pval of $LRT < 0.001$) (Extended Data Fig. 7b).” of the main text. These smoking-methylation associations correspond to 5 genes, which have been previously reported as differentially methylated in smokers compared to non-smokers. To support this point, we have added the Christiansen et al. 2021 reference, which explored smoking status and DNA methylation in 1407 human whole blood samples across 4 independent UK population-based cohorts and reported our identified loci in their top list as well as referenced previous studies also identifying those: “*AHRR, F2RL3, GP15, PRSS23 and RARA CpG sites that were previously identified as candidate smoking-related loci in whole blood*^{26,27}.”

I don't quite know what to make of the authors' genomic eQTL analysis. No insights from this are mentioned in the Discussion. How would the authors describe the impact of this work? Do they see

this leading to identifying people at risk of a reduced immune response based on genotyping and attempting to intervene therapeutically? What would be the roadmap to this? This is a lot of work and it would be good to make the impact clearer to the reader.

We thank the reviewer for highlighting this point. In line with our most novel and interesting findings related to smoking-cytokine associations, we used and discussed the genetic associations as a relevant metric for comparing the proportion of cytokine variance explained by the different associated factors. As a specific example, the effect of current smoking on bacterial-induced CXCL5 levels is 9%, a level that is equivalent to common cis-eQTL effects, which are known to have an impact on disease risk. Following the reviewer's question, we have now added additional text to the discussion to reinforce and clarify this point: *"The variance explained by the smoking status for some cytokines upon stimulation, reach a level equivalent to the one explained by age, sex and genetic variants, which are known to have consequences for disease risk."* We also have added details about the novelty of these genetic associations (Figure 2 for reviewers) and modified the text accordingly *"The results (Table 1) are consistent with the trans-pQTLs identified from an independent cohort⁷, for common cytokines (IL6, IL10 and TNFa) and stimulations (LPS and C. albicans). In addition, we identified 22 pQTLs, that were not identified by this study, nor are present in the Somalogic database²⁰: CXCL5 (rs10013453, trans; rs10779330, trans), IFNG (rs2564594, trans; rs3775291, trans; rs4833095, trans), IL12A (rs143060887, cis), IL12B (rs143060887, trans), IL17A (rs10004195, trans), IL1B (rs10863358, trans; rs3764613, trans; rs3775291, trans), IL2 (rs1801274, trans; rs4833095, trans; rs6815814, trans), IL23A (rs10779330, trans; rs5743614, trans), IL6 (rs10034903, trans; rs11936050, trans; rs3775291, trans; rs72636686, trans), IL8 (rs4833095, trans), TNF (trans; rs3775291, trans)."*

We also tested the significance of interactions between the smoking status and each of the SNPs that we have found to be associated with the cytokine levels in the different immune stimulations. Significant interactions (BY adj pval < 0.05) were observed for IL1b, IL6, IL8 and IL17 in response to BCG stimulation. For the strongest interaction, smoking status can remove differences in response between individuals of different genotypes (New Extended Data Fig 8). While not changing our major findings, this is very interesting in showing that genotype effects can be modulated by the smoking status. We have added a supplementary figure (Extended Data Fig. 8) and modified the text to highlight this point: *"We also assessed potential interactions between the identified SNPs and smoking status. No significant associations were observed in E. coli, LPS, SEB and CD3+CD28, but BCG stimulation showed significant genetic-smoking interactions (BY adj-pval < 0.05) for rs61934597 and IL8, IL17, IL1b, IL6, and rs72636686 and IL8, IL1b and IL17 (Extended Data Fig. 8a). For the strongest interaction, between rs72636686 and smoking status for IL8 levels, smoking status can remove differences in response between individuals of different genotypes (Extended Data Fig. 8b), showing genotype effects can be modulated by the smoking status."* However, regarding the potential clinical implications of these QTL results, while this is a very interesting question, we are not aware of any clinical use of eQTLs so far and believe that this discussion is beyond the scope of our study.

Figure 2 for reviewers: Barplot of pQTL associations from the SomaLogic plasma protein database. **a**, Significance of associations between genetic variants listed in Table 1 and the corresponding cytokines. **b**, Significance of associations between variants in linkage disequilibrium ($r^2 < 0.2$) with variants listed in Table 1. Dashed line: FDR adjusted p-value = 0.05, dotted line, FDR adjusted p-value = 0.01.

Extended Data Fig. 8: Smoking status and SNPs interactions. **a**, Heatmaps showing $-\log_{10}(\text{BY adj. pval of LRT})$ of interactions between genetic variants listed in Table 1 and Smoking status for each induced cytokine considering age, sex and batchId as covariates. These are colored according to the color key on the side of each heatmap and stars are shown depending on the strength of association (* < 0.05; ** < 0.01; *** < 0.001). **b**, Boxplot showing IL8 levels depending on the genotype for rs72636686 and the smoking status.

Referee #2 (Remarks to the Author):

Summary of the key results

In the submitted work, Saint-André et al. investigated the relationships between cytokine responses and immune stimulations with respect to environmental factors in the *Milieu Intérieur* (MI) project. Even though the title only stated the effects of smoking on host immune response, the authors investigated and nominated a wide range of variables, such as age, sex, genetic background and CMV infection status, and their impact on cytokine responses.

We thank the reviewer for recognizing the wide range of variables that we analyzed in our study. The reason to focus the title on smoking, despite additional analysis of a wide range of variables, is because this is the main novelty and most important message of our study.

Originality and significance: if not novel, please include reference

I don't know the literature in the field of cytokine responses with respect to environmental factors well. But a brief search suggested that previous studies have shown smoking alters methylation as well as cytokine responses. This has been shown in population cohorts such as UKB (Amador et al. 2021), TwinsUK cohort (Christiansen et al. 2021) and Lothian Birth Cohort 1936 (Corley et al. 2019).

We agree with the reviewer that the association between smoking and methylation have been previously described, including in some of our own studies (Bergstedt et al., 2022). However, the novelty of the present study concerns the first identification of an association between smoking status and T cell induced cytokines, which we show is epigenetically mediated through DNA methylation. Regarding the specific studies cited, they were quite different from ours both in scope and results obtained: Amador *et al.* 2021 studied how associations between genetic variants and smoking may affect obesity-related traits, Christiansen *et al.* 2021 showed associations between smoking and CpG sites, and Corley *et al.* 2019 showed a relationship between smoking-epigenetic associations and aging cognitive phenotypes. None of these studies explored cytokines or immune phenotypes/functions. However, we thank the reviewer for highlighting these articles and have included Christiansen *et al.* 2021 as a relevant citation for the smoking-methylation associations in our revised manuscript: "AHRH, F2RL3, GP15 and RARA CpG sites that were previously identified as candidate smoking-related loci in whole blood^{25,26}."

In the abstract, the authors stated "Our findings describe new factors associated with cytokine secretion variability, identify a new role for smoking on immune response regulation". Please can the authors describe exactly what new factors and new role for smoking on immune response regulation

did they identify. As far as I can read:

- No new data was generated for the MI project.
- No new methods were introduced for analyzing the dataset apart from looking at different environmental factors that were previously measured in the study.
- No novel biological mechanisms were identified through the study, only confirmation of previous findings.

We thank the reviewer for allowing us to clarify these points, which clearly highlights some misunderstanding.

- Data generated:

The entire cytokine dataset, which is the basis of this study, is newly generated and is presented in this manuscript for the first time (see Table S1): it consists of 13 cytokines measured in 12 stimulation conditions for 956 donors (individuals from the 1,000 who gave consent to share their data publicly), so a total of 149,136 new data points are reported in our manuscript. We realize that given how we introduced and cited our previous study on 25 donors, this may have caused this confusion and we apologize for that. We have now clarified this by moving this reference to the methods section. In addition, 136 Case Report Form (CRF) variables (presented in Table S2) are published for the first time as a table included in this study (only sub-samples were tested in previous MI studies) and will enable other researchers to perform further analyses from this data.

- Methods:

To make these new observations the analysis consisted of integrating this new dataset with 136 CRF variables, 76 immune cell subsets, selected CpG sites, and 5×10^6 SNPs in an integrative manner. Substantial bioinformatic work has been performed to enable testing the potential associations of such a number of covariates with appropriate pre-processing, statistical corrections (identification of robust associations through LRT and BY correction), filter for artefactual associations (filtering for minimal occurrence in levels of categorical variables and for induced cytokines only) and compare effect of associated variables. All the scripts developed to perform this work will be made available to the community.

- Biological mechanisms:

The new biological mechanisms identified include past smoking associations with T cell induced cytokines, that are mediated through long lived T and B cells and specific epigenetic changes, as well as active smoking associations with a bacterial induced inflammatory cytokine, that is associated with elevated levels of plasma CAECAM6. Such regulatory effects of smoking on cytokines were never shown before.

We hope these clarifications will help the reviewer to better appreciate the novel data and findings reported in our manuscript. We have also reformulated the abstract to make it more precise on the “new factors associated with cytokines” and “new role for smoking”, as requested by the reviewer: *“Individuals widely differ in their immune responses, with age, sex and genetic factors playing major roles in this inherent variability. However, the variables that drive such differences in cytokine*

secretion, which is a crucial component of the host response to immune challenges, remain poorly defined. Here we investigated 136 variables and identified smoking, cytomegalovirus latent infection and body mass index as major contributors to cytokine response variability, with comparable effects in strength as age, sex and genetics. We find that smoking influences both innate and adaptive immune responses. Notably, its effect on innate responses is quickly lost after smoking cessation and is specifically associated with plasma levels of CEACAM6, whereas its effect on adaptive responses persists long after individuals quit smoking and is associated with epigenetic memory. This is supported by the association of the past smoking effect on cytokine responses with DNA methylation at specific signal transactivators and metabolism regulators. Our findings identify three novel variables associated with cytokine secretion variability and reveal new roles for smoking in the short and long-term regulation of immune responses. These results have potential clinical implications for the risk of developing infections, cancers or auto-immune diseases.

Data & methodology: validity of approach, quality of data, quality of presentation

The main text is clearly written, but data presentation and method description have room for improvement with more details and consistency (suggested improvements listed below).

We thank the reviewer for highlighting these points, which we have corrected and answered in a point by point below.

- Line 51-53 could be better written. I couldn't find anything describing adjusting genetic factors in their Method or justify the statement of "a homogenous genetic background". Are these individuals genotyped? If yes, a genetic PC would help to justify the statement.

As described in the methods (lines 425-427), the cohort was recruited in a way to avoid genetic stratification: "to avoid genetic stratification in the study population, the recruitment of donors was restricted to individuals whose parents and grandparents were born in Metropolitan France." Metropolitan France refers to the mainland part of France in Western Europe and excludes the French overseas departments and territories. So the donors of the cohort originate from a similar genetic background. Indeed, the individuals are genotyped (description in methods page 30, lines 596-606), which permitted the pQTL analysis. In specific response to this question, we performed the same association tests as for the eCRF variables on the first 20 principal components of the PCA on the individual genotypes and found no significant associations of these PCs with the induced cytokines at the p-value threshold used throughout the manuscript (BY adj-pval < 0.01). Only a slight association between IL10 and PC1 in the BCG stimulation was observed ($0.01 < \text{BY adj-pval} < 0.05$) (Figure 3 for reviewers). To avoid confusion, we have now clarified this point in the methods section: "Additionally, we formally checked how the genetic background of the donors could affect cytokine levels and found that the first 20 principal components out of the Principal Component Analysis (PCA) on the individual genotypes showed no significant associations with cytokine responses at the p-value threshold BY adj-pval < 0.01."

Figure 3 for reviewers: Heatmaps showing $-\log_{10}(\text{BY adj.pval of LRT})$ of association between the genetic PCs and induced cytokines considering age, sex and batchId as covariates. These are colored according to the color key on the side of each heatmap and stars are shown depending on the strength of association (* < 0.05; ** < 0.01; *** < 0.001).

- It is unclear in various places what regression model was used. For example, Line 100-101 compared the effect sizes of smoking status to age and sex, but did not provide details of the

regression models used in Extended Data Fig. 3 and 4. For sex, is it $\text{lm}(\text{cytokine} \sim \text{sex} + \text{batchID})$ or $\text{lm}(\text{cytokine} \sim \text{age} + \text{sex} + \text{batchID})$? And how can the authors compare effect sizes when one is adjusted for age and sex $\text{lm}(\text{cytokine} \sim \text{sex} + \text{age} + \text{batchID} + \text{smoking})$ or does Figure 2 show $\text{lm}(\text{cytokine} \sim \text{smoking} + \text{batchID})$. In either case more clarity is needed.

We thank the reviewer for this question. The heatmap in Fig. 1c shows the significance of the comparison of the model $\text{lm}(\text{cytokine} \sim \text{tested variable} + \text{age} + \text{sex} + \text{batchID})$ with the model $\text{lm}(\text{cytokine} \sim \text{age} + \text{sex} + \text{batchID})$ using a Likelihood Ratio Test (LRT). When significant, this test shows that the tested variable better explains the cytokine levels than the model considering age, sex and batchID only. The effect size plots in Fig. 2 display the estimates of the tested variable from the model $\text{lm}(\text{cytokine} \sim \text{tested variable} + \text{age} + \text{sex} + \text{batchID})$. The significance of association corresponding to each estimate is not used in the heatmaps in Fig. 1c, it is used to color in black or grey the confidence intervals in Fig. 2. Similarly, the effect size plots of original Extended Data Fig. 4 were generated from the same code used for generating the heatmaps of original Extended Data Fig. 3 and display the estimates of the tested variable. For previous Extended Data Fig. 3 and 4 the effect of age and sex were tested using batchID as covariate: $\text{lm}(\text{cytokine} \sim \text{sex} + \text{batchID})$ for sex and $\text{lm}(\text{cytokine} \sim \text{age} + \text{batchID})$ for age. Effect sizes are not easy to compare between age, sex and smoking status. Indeed, sex and smoking status are categorical variables, respectively comparing one sex to the other and current or past smokers to non-smokers, while age here is numerical with effect sizes reflecting the effect of 1 year increase in age. To compare effects, we estimated the percentage of variance explained (r^2) by each predictor in the model containing all the variables we found associated (Fig. 5). We removed the sentence *“with effect sizes that are similar to the strongest age or sex effect sizes”* (line 100-101), which may have created this confusion and apologize for that. We have re-done the analysis adding, respectively, age or sex in the models testing the effects of sex or age. Significance of LRT and effect sizes are displayed respectively in new Extended Data Figures 4 and 5 and the results are very similar to previous Extended Figures 3 and 4. We have now accordingly adjusted/clarified these points in the figure legends and methods: *“Such categorical variables or numerical ones were tested for associations with the log-transformed induced cytokine levels in each stimulation through Likelihood Ratio Tests (LRTs), using age, sex and the technical variable “batchID” (corresponding to two batches of TruCulture tubes produced at different periods of time) as co-variates: the LRT compared the models $\text{lm}(\text{cytokine} \sim \text{variable} + \text{age} + \text{sex} + \text{batchID})$ with $\text{lm}(\text{cytokine} \sim \text{age} + \text{sex} + \text{batchID})$, followed by Benjamini-Yekutieli (BY) multiple testing correction. For Extended data Figures 4 and 5 the models compared were $\text{lm}(\text{cytokine} \sim \text{age} + \text{sex} + \text{batchID})$ with $\text{lm}(\text{cytokine} \sim \text{sex} + \text{batchID})$ for age and $\text{lm}(\text{cytokine} \sim \text{sex} + \text{batchID})$ with $\text{lm}(\text{cytokine} \sim \text{age} + \text{batchID})$ for sex. P-values of association tests were represented using ggplot2 3.2.1 in R 3.6.0.”*

Extended Data Figure 4: Heatmaps showing associations of age and sex variables corrected for sex or age respectively and for batchId, on induced cytokines in each stimulation condition. Significance of the BY adj-pval Likelihood Ratio Test is marked with stars (* < 0.05; ** < 0.01; * < 0.001).**

Extended Data Figure 5: Effect size plots for sex (a) and age (b) corrected for age or sex respectively, and batchId, on induced cytokines in each stimulation condition for the 1,000 individuals of the *Milieu Intérieur* cohort. Significant effect sizes are in black, others are in grey. Significance of the BY adj-pval Likelihood Ratio Test is marked with stars (* < 0.05; ** < 0.01; *** < 0.001).

- The authors need to describe how they define innate versus adaptive stimulation and which stimulus belongs to which.

We thank the reviewer for this comment. We have now clarified innate versus adaptive stimulations in the text and defined which stimulus belongs to which as follows: *“The stimulations are classified into 4 categories: microbial (Bacillus Calmette-Guérin (BCG), Escherichia coli (E. coli), lipopolysaccharide (LPS), and Candida albicans (C. albicans)) and viral (influenza and Polyinosinic-polycytidylic acid (PolyIC)) agents, that are predominantly recognized by receptors on innate immune cells; T-cell activators (Staphylococcus aureus Enterotoxin B superantigen (SEB) and anti-CD3+CD28 antibodies (CD3+CD28)), which induce adaptive immune responses; as well as cytokines (TNF α , IL1b, and IFN γ).”*

- The quality control steps for processing the methylation steps are not described. This is important as methylation arrays can be noisy and subject to batch effects. If it is the same as Bergstedt et al. 2022, it also needs to be mentioned.

The reviewer is correct that methylation arrays can be noisy and subject to batch effect. The quality control steps for processing the methylation data used in this study are the same as the ones performed in Bergstedt et al., 2022. This information has been added to the revised manuscript: *“CpG methylation profiles were generated using the Infinium MethylationEPIC BeadChip (Illumina, California, USA) on genomic DNA treated with sodium bisulfite (Zymo Research, California, USA) for 958 individuals of the MI cohort as described in Bergstedt et al., 2022.”*

- The authors used the term “factor” for different variables (genetic factors, environmental factors, blood factors, etc). I think they all mean slightly different things. Did the authors perform a factor analysis to define the “factors” in the clinical variable association analysis? It would be better to use more precise words to describe them for clarity. E.g. instead of “blood factors” just call them proteins.

We apologize for this confusion and have renamed “soluble blood factors” as “soluble blood proteins”, “genetic factors” as “genetic variants” and most remaining “factors” as “variables” to better reflect their meaning.

- It is unclear which scripts in the provided GitHub repository are related to this project.

We apologize for this confusion. All relevant scripts are provided on GitHub (<https://github.com/ViolaineSaint-Andre/>) under the defined project folder titled “MI_13Cytokines_12Stim” and labeled after the figure numbers to ease the connection with the work presented in the manuscript.

Appropriate use of statistics and treatment of uncertainties

I have a few questions/suggestions regarding the statistics used in this manuscript:

- The authors used BY-adjust p-value to nominate significant associations, but this also needs to be adjusted for multiple-testing among 13 cytokines too. It would also be beneficial to state the actual BY-adjust p-value for the readers to understand the level of dependencies for all the environmental variables considered.

We apologize if this was not clear. For heatmaps similar to those on Figure 1c, we used BY correction to correct for multiple testing for the whole heatmaps, so it takes into account the tests made for all the variables tested (136 for Figure 1c) with all the induced cytokines in a specific stimulation. We have now clarified this point in the methods by adding: “*Benjamini-Yekutieli (BY) multiple testing correction applied to the whole heatmaps, so taking into account the tests made for the 136 CRF variables with all the induced cytokines in a specific stimulation.*”, and added tables of the BY adjusted p-values corresponding to the main Figures in the source data attached to the figures.

- A supplementary table needs to be provided for all reported associations presented in the main figures and in the text.

We have now included source data tables by figure that includes all reported associations.

The principal component analysis needs to be described in the Method section. Are each cytokines standardized similar to what they did in the heatmap analysis?

We have now included additional information on the principal component analysis in the methods section: “*The PCA on Extended Data Figure 1 was created in R 4.2.1 using the FactoMineR 2.8 package. The data was log transformed and by default scaled to unit variance and missing values were imputed by the mean of the variable.*”

- The authors seem to use different models for claiming significance in their heatmap and effect size analysis. The statistical tests should be the same in these two presentations.

The models used for computing significance in the heatmaps and effect sizes are the same and were described above (see for example Fig1c_and 2 script on GitHub for further technical details).

- Are the p-values presented in the boxplots also multiple-testing corrected and adjusted for age, sex and batch?

The p-values of the Wilcoxon tests are indeed adjusted for multiple testing (the R `wilcox.test` function does correct for multiple testing by default). These are not adjusted for age and sex, as we thought it is important to show the raw data as well, especially as differences can be seen without correction. However, the reviewer is right that it can also be important to see boxplots of corrected data. In order to address this point, we have now regressed out the cytokine concentration values on age, sex and batchId, and created boxplots on the residuals. By doing so, our original observations remain, even if the p-values are a bit less significant in the specific case of SEB and CD3+CD28. These plots have been now added to Extended Data Fig. 6. For the sake of homogeneity, we have also added scatterplots corrected for age and sex to Extended Data Fig.6 and modified the methods accordingly: *“Adjusted p-values on the boxplots were computed with the `wilcox.test` function, correcting for multiple testing. Versions of the boxplots and scatterplots made on the residuals after regression on age, sex and batchID are displayed on Extended Data Fig. 6d-f.”*

Extended data Fig.6: Smoking effect on innate and adaptive immune responses, represented by LPS and CD3+CD28 stimulations respectively and same plots on residuals after regression on age, sex and batchId. Boxplots of CXCL5 concentration in LPS stimulation (a) and of IL2 concentration in CD3+CD28 stimulation (b) for never, past and current smokers. Significance of the Wilcoxon test is indicated with stars above the boxes on the left (between never and past smokers), in the middle (between never and current smokers) and on the right (between past and current smokers) (N.S: non-significant, * < 0.05; ** < 0.01; *** < 0.001). Plots showing CXCL5 concentrations in LPS stimulation or IL2 and IL13 concentrations in CD3+CD28 stimulation (c) depending on the numbers of years smoking for current smokers (red) or past smokers (blue). Grey areas depict the confidence intervals of the regression lines. Similar boxplots and scatterplots are displayed on residuals of CXCL5 in *E. coli* and LPS stimulations or IL2 and IL13 in SEB and CD3+CD28 stimulations after regression on age, sex and batchId variables (d-f).

- There is no discussion of how genetic background will affect the results. Considering the authors have stated that genetic background does make a difference in cytokine responses, genetic PCs should be included as covariates in their linear models.

We thank the reviewer for highlighting this important point. However, we should clarify that when we say that genetic background (i.e., variation in genetic ancestry) affects cytokine response, we

refer to strong variation in genetic background. In light of the relative genetic homogeneity of our cohort, population level genetics has indeed a negligible impact on cytokine responses. The association tests of the first 20 genetic PCs with induced cytokines (Figures 3 for reviewers) support this point: the first genetic PCs of the MI cohort are not associated with any of the cytokines which have associations with the CRF variables that are discussed in the article. We have added additional text to the methods to clarify this point: *“Additionally, we formally checked how the genetic background of the donors could affect cytokine levels and found that the first 20 principal components out of the Principal Component Analysis (PCA) on the individual genotypes showed no significant associations with cytokine responses at the p-value threshold BY $adj-pval < 0.01$.”*

- Genome-wide PCs should also be included as covariates in the pQTL analysis.

As genetic PCs were not associated with variation in cytokine levels, we did not see the need to include them in the pQTL analysis. Additionally, when conducting pQTL analyses with the first 2 genetic PCs in the models, this did not modify our results.

Conclusions: robustness, validity, reliability

As the authors mentioned in the conclusion, there is no replication cohort in their study and it is currently achievable with population cohorts such as UK Biobank.

Unfortunately no cytokine data after immune stimulation exists in the UK Biobank (<https://www.ukbiobank.ac.uk/enable-your-research/about-our-data/past-data-releases>), so this study cannot be used as a replication cohort, as our associations were only observed after specific immune stimulation. While we appreciate the importance of replication studies, when identifying novel findings such as those reported in our study, it is often not possible to find other such studies that would allow their replication.

Suggested improvements: experiments, data for possible revision

Major

- Replicate main findings in an independent population cohort

We would happily try to replicate our findings in an independent population cohort if such an existing dataset was available, however we are not aware of any. We detected the novel cytokine response associations with smoking only in the context of immune stimulation (both for bacterial and superantigen induced responses), and, as indicated above, the UK biobank does not have any induced cytokine datasets. The only population cohort study that measured cytokines after immune stimulations is the 500FG cohort, however they do not have T cell agonist stimuli in their study, only specific microbes, and do not have CXCL5 cytokine measured in these stimulations. Therefore, addressing this specific point would require the creation of a completely new cohort.

While we understand the reasons for this request, requiring a replication cohort almost by definition prohibits the publication of new and exciting findings.

- A conditional analysis should be undertaken for the various smoking related phenotypes e.g. current smoker, previous smoker, number of years smoked, number of cigarettes smoked a day etc. as it is unclear whether these variables are correlated and which is the primary signal of association. This is important as it has different biological implications.

We are not sure to fully understand the reviewer's point here. Current and past smokers are exclusive groups. "Number of years smoked" and "Total number of cigarettes smoked" variables are relevant for smokers and past smokers only (not for non-smokers), and the variable "Numbers of years since last smoke" is only relevant for past smokers. Each of these variables was thus tested separately, in distinct subsets of the donors.

- Since the smoking effect on cytokine is the main focus, can the authors perform a further conditional analysis with $\text{lm}(\text{cytokine} \sim \text{age} + \text{sex} + \text{batchID} + \text{CMV} + \text{all other non-smoking related factors})$ and report if the smoking has an independent effect from all other environmental factors.

We respectfully disagree with the reviewer on this point. The inclusion of hundreds of variables in the model would not likely be relevant here, as it will introduce noise and multi-collinearity. Many difficulties tend to arise when there are more than five independent variables in a multiple regression equation. Without penalization this could also lead to overfitting. In a first attempt, we applied regularized regression methods, such as glmnet, but these gave a lot of false positive results, most likely due to the fact that the CRF covariates are very heterogenous in their nature. For these reasons, we tested the association of each variable through linear regressions independently and designed a model to include all of the variables that we found to be associated with the induced cytokines (including smoking status) to show the percentage of variance explained by each of them when considered all together in the same model (Fig. 5).

- Since genotyping information is available for this cohort, it would be good to see a SNP x tobacco interaction analysis, similar to Pisecka et al.

We thank the reviewer for raising this point. We have now tested the significance of interactions between the smoking status variable and each of the SNPs we found to be associated with the cytokine levels in the different immune stimulations. Significant interactions (BY adj pval < 0.05) were observed for IL1b, IL6, IL8 and IL17 in BCG but not in E. coli, LPS, SEB and CD3+CD28 stimulations, so they do not impact our major findings. We have added a supplementary figure (New Extended Data Fig. 8) and modified the text to highlight this point: "*We also assessed*

potential interactions between the identified SNPs and smoking status. No significant associations were observed in E. coli, LPS, SEB and CD3+CD28 but BCG stimulation showed significant genetic-smoking interactions (BY adj-pval < 0.05) for rs61934597 and IL8, IL17, IL1b, IL6, and rs72636686 and IL8, IL1b and IL17 (Extended Data Fig. 8a). For the strongest interaction, between rs72636686 and smoking status for IL8 levels, smoking status can remove differences in response between individuals of different genotypes (Extended Data Fig. 8b), showing genotype effects can be modulated by the smoking status.”

Extended Data Fig. 8: Smoking status and SNPs interactions. **a**, Heatmaps showing $-\log_{10}(\text{BY adj.pval of LRT})$ of interactions between genetic variants listed in Table 1 and Smoking status for each induced cytokine considering age, sex and batchId as covariates. These are colored according to the color key on the side of each heatmap and stars are shown depending on the strength of association (BY adj.pval of LRT < 0.01; (* < 0.05; ** < 0.01; *** < 0.001). **b**, Boxplot showing IL8 levels depending on the genotype for rs72636686 and the smoking status.

- The description of novel pQTL signals is inaccurate. A search on Open Targets suggested that rs35045 was previously reported in Sun et al. 2018. Furthermore, what are the LD relationships between the reported novel pQTLs in relationship with the previously top reported pQTL SNP in respect to the same protein. The authors need to perform a conditional analysis to convince the readers they are indeed independent novel regulatory SNPs compared to previously reported pQTL signals. For example, the claimed “new pQTL” for IL6 (rs35345753) is in LD with the lead pQTL SNP (rs2069840) reported in Pietzner*, Wheeler* et al., 2021 ($r^2 = 0.2$).

We apologize for the confusion caused here by the use of the term “novel” as compared to the genetic associations previously identified in the MI study. However, in order to test the novelty of our pQTL results, we studied the SomaLogic plasma protein pQTL database (Sun et al., Nature 2018, <http://www.phpc.cam.ac.uk/ceu/proteins/>) for both cis- and trans-pQTLs listed in Table 1. This

dataset reference known pQTL associations for CXCL5, IFNg, IL1b, IL2, IL6, IL10 and IL12a. Significant associations were identified between the variants rs352045 (cis), rs2393969 (trans), rs10822168 (trans) and the protein CXCL5 (respective FDR adjusted $p = 3.02e-10$, $p=0.01$ and $p = 0.022$), between rs35345753 (cis), rs62449491 (cis) and IL6 (respective FDR adjusted $p = 4.17e-3$ and $p = 0.017$) and between rs3775291 (trans) and IL12A (FDR adjusted $p = 0.049$) (Figure 2a for reviewers). In order to test associations for SNPs in linkage disequilibrium (LD) with the SNPs originally referenced in Table 1, we used a dataset of LD from the ensemble database with similar ancestries as the *Milieu Intérieur* cohort (1000GENOMES:phase 3:CEU: Utah residents with Northern and Western European ancestry). To be inclusive, SNPs with a r^2 greater than 0.2 were selected as associated alleles and underwent the same analysis as the one performed with the SNPs of reference. SNPs that came out as significant are those in LD with the SNP referenced in Table 1 that is significantly associated with the corresponding protein (rs352045, rs10822168 and rs62449491 and rs10779330) (Figure 2b for reviewers). We also compared the SNPs listed in Table 1 with the pQTLs performed on the 500FG cohort for common cytokines (IL6, IL10 and TNFa) and stimulations (LPS and *C. albicans*) (Li et al., 2016). Altogether, this screening of multiple databases supports novel associations for 22 pQTLs that were not identified by the 500FG consortium, nor are present in the Somalogic database: CXCL5 (rs10013453, trans; rs10779330, trans), IFNG (rs2564594, trans; rs3775291, trans; rs4833095, trans), IL12A (rs143060887, cis), IL12B (rs143060887, trans), IL17A (rs10004195, trans), IL1B (rs10863358, trans; rs3764613, trans; rs3775291, trans), IL2 (rs1801274, trans; rs4833095, trans; rs6815814, trans), IL23A (rs10779330, trans; rs5743614, trans), IL6 (rs10034903, trans; rs11936050, trans; rs3775291, trans; rs72636686, trans), IL8 (rs4833095, trans), TNFa (trans; rs3775291, trans). It is always difficult to be exhaustive when listing identified QTLs. However, the databases and studies we screened are to our knowledge those best referencing the identified pQTLs for the cytokines in our study. Further conditional analyses could be performed in order to describe these new reported associations but would take too much space in this article, and our main results are not focused on these associations. We mainly describe these genetic findings here in order to be able to account for and compare the genetic effects on the induced cytokines to the novel smoking status effects.

The results of this additional analysis have been added to the main text: *“The results (Table 1) are consistent with the trans-pQTLs identified from an independent cohort⁷, for common cytokines (IL6, IL10 and TNFa) and stimulations (LPS and C. albicans). In addition, we identified 22 pQTLs, that were not identified by this study, nor are present in the Somalogic database²⁰: CXCL5 (rs10013453, trans; rs10779330, trans), IFNg (rs2564594, trans; rs3775291, trans; rs4833095, trans), IL12A (rs143060887, cis), IL12B (rs143060887, trans), IL17A (rs10004195, trans), IL1b (rs10863358, trans; rs3764613, trans; rs3775291, trans), IL2 (rs1801274, trans; rs4833095, trans; rs6815814, trans), IL23A (rs10779330, trans; rs5743614, trans), IL6 (rs10034903, trans; rs11936050, trans; rs3775291, trans; rs72636686, trans), IL8 (rs4833095, trans), TNFa (trans; rs3775291, trans).”*

Corresponding text has been added to the methods: *“In order to test the novelty of our pQTL results, we studied the SomaLogic plasma protein pQTL database²⁰, for both cis- and trans-pQTLs listed in Table 1. This dataset allowed testing associations for CXCL5, IFNg, IL1b, IL2, IL6, IL10 and IL12a. Significant associations were identified between the variants rs352045 (cis), rs2393969*

(trans), rs10822168 (trans) and the protein CXCL5 (respective FDR adjusted $p = 3.02e-10$, $p=0.01$ and $p = 0.022$), between rs35345753 (cis), rs62449491 (cis) and IL6 (respective FDR adjusted $p = 4.17e-3$ and $p = 0.017$) and between rs3775291 (trans) and IL12A (FDR adjusted $p = 0.049$). In order to test associations for SNPs in linkage disequilibrium (LD) with the SNPs originally referenced in Table 1, we used a dataset of LD from the ensemble database with similar ancestries as the Milieu Intérieur cohort (1000GENOMES:phase_3:CEU: Utah residents with Northern and Western European ancestry). To be inclusive, SNPs with a $r^2 > 0.2$ were selected as associated alleles and underwent the same analysis as the one performed with the SNPs of reference. SNPs that came out as significant are those in LD with the SNP referenced in Table 1 that is significantly associated with the corresponding protein. In addition, we also screened eQTL results. We compared our pQTL results with the eQTLs reported in our previous work based on nanostring transcriptomic data for common cytokines (CSF2, IFN γ , IL1b, TNF α , IL2, IL6, IL8, IL10, IL12p70, IL13, IL17, IL23) and stimulations (E.coli, C. albicans, Influenza, BCG, and SEB)9, which identified 2 main loci: the TLR1/6/10 locus and the CR1 locus. Association of variants referenced in Table 1 were found in the GTEx consortium database for rs1518110 and IL10 (FDR adjusted $p = 4.3e-9$), for rs352045 (cis) and rs2564594 (cis) and CXCL5 (respective FDR adjusted $p = 9.2e-23$ and $4.1e-22$) in whole blood and for rs143060887 (cis) and IL12A (FDR adjusted $p = 0.000076$). Significant associations between rs352045 and CXCL5 and between rs1518110 and IL10 were also found in the eQTLgen catalogue.”

Figure 2 for reviewers: Barplot of pQTL associations from the SomaLogic plasma protein database. **a**, Significance of associations between genetic variants listed in Table 1 and the corresponding cytokines. **b**,

Significance of associations between variants in linkage disequilibrium ($r^2 < 0.2$) with variants listed in Table 1. Dashed line: FDR adjusted p-value = 0.05, dotted line, FDR adjusted p-value = 0.01.

Minor

- Extended Data Fig.1 would be more clear if the authors chose a better color scheme among 4 different stimulation categories (different shades of red, blue, black and green say).

We have redone the PCA with a different color code as indicated and replaced Extended Data Fig.1 with this new version.

Extended Data Fig.1: Principal Component Analysis (PCA) of individuals for the 13 cytokines in the 12 stimulation conditions. Each dot represents one individual in a specified stimulation condition. Contribution of

the cytokines to the first 2 dimensions are represented with arrows.

- It would be helpful to see separate PCA within each stimulations, and color them by age and sex. Based on the Extended Data Fig. 1, there seems to be a clear stratification in PolyIC, Influenza and Null, are they separated by smoking status or any other non-biological factors, e.g. batch, season?

We have performed separate PCAs per stimulation and have color-coded the plots by age, sex, smoking status, season and batchID as requested (Figure 5 to 9 for reviewers). The visual stratification effect cannot be explained by smoking, season nor batchID for Influenza, PolyIC and the Null stimulation conditions. However, there is some level of stratification due to sex and age for these stimulations. This is consistent with the associations of age and sex shown in Extended Data Fig. 4. (Season effects may be related to the batchID effect, as the batch correspond to different sets of TruCulture tubes used over time).

Figure 5 for reviewers: PCA plots of individuals in each stimulation colored by age intervals: 1) [20-30), 2) [30-40), 3) [40-50), 4) [50-60), 5) [60-70).

Figure 6 for reviewers: PCA plots of individuals in each stimulation colored by sex: 1) men, 2) women

Figure 7 for reviewers: PCA plots of individuals in each stimulation colored by smoking status: 0) non-smokers, 1) past smokers, 2) current smokers

Figures 8 for reviewers: PCA plots of individuals in each stimulation colored by season: 1) winter (Jan-Mar), 2) spring (Apr-Jun), 3) summer (Jul-Sept), 4) autumn (Oct-Dec)

Figure 9 for reviewers: PCA plots of individuals in each stimulation colored by batchId: two different series of TruCulture tubes.

- Fig. 1 has a missing legend for panel c. The color scale seems to be different for each stimulation panel. This should be unified with an added color legend.

The color scale is different on each panel as each stimulation is analyzed separately. The color scales corresponding to each heatmap were indeed missing. We now show these on the side of

each heatmap and described them in the Figure 1 legend: “These are colored according to the color key on the side of each heatmap and stars are shown depending on the strength of association ($* < 0.05$; $** < 0.01$; $*** < 0.001$).” We also have created a Figure on which the same color scale is applied to all the plots (Figure 4 for reviewers). However, we prefer to keep different color scales on the original Figure, as each stimulation condition is independent of each other and therefore we believe it is more informative to have a separate scale for each.

Figure 4 for reviewers: Fig 1c with the same scale for all stimulations. Heatmaps showing $-\log_{10}(\text{BY adj.pval of LRT})$ of association for the eCRF variables associated with at least one cytokine in each stimulation. These are colored according to the color key on the side of each heatmap and stars are shown depending on the strength of association ($* < 0.05$; $** < 0.01$; $*** < 0.001$).

- Don't really understand the grouping scheme for environmental factors. It is unclear to me why "cooked meals" and "heart rate" belong to the same group.

We originally grouped variables based on our interpretation of correlations between the different variables. As shown in a correlation plot (Figure 10 for reviewers), abdo. circ and systolic measure mostly correlate with BMI, and all the smoking-related variables correlate with each other. It is however less strong for beer, heart rate and cooked meals, so we agree with the reviewer that this may be confusing and have removed the original grouping and adapted the text and legend of Figure 1 accordingly.

Figure 10 for reviewers: Correlation plot of the eCRF variables associated with at least one induced cytokine in at least on stimulation condition.

- The SI Tables are missing legends.

The legends for the SI Tables are in the SIGuide.txt file, as requested in the Nature guidelines to authors.

- Extended Data Fig.2 has confusing names for the cytokines (e.g.e GM_CSF and ena_78).

We apologize for this inconsistency, which has been corrected in a new version in Extended Data Figure 2.

- The formatting of Table 1 could be improved, with respect to the tables and exponentials.

Table 1 has been reformatted and adjusted p-values are now written as exponentials.

References: appropriate credit to previous work?

- The results of CEACAM6 - CXCL5 association in smokers should be described more and reference earlier studies if it is already known.

We have not seen previously reported associations on CEACAM6-CXCL5 in smokers, however we have included additional text and references on previous reports of separate CEACAM6-smoking and CXCL5-smoking associations: *“While previous studies have suggested CXCL5³² and CEACAM6³³ can be elevated in smokers, most studies focused on pulmonary sites and in patients with respiratory disease (eg. cancer, COPD, asthma) which can confound the results. Our study identifies a strong link between these previously proposed disease biomarkers and response to immune challenges in smokers versus non-smokers. Furthermore, our findings in healthy donors open avenues for further exploration into understanding how smoking acts as a risk factor for cancers beyond the lungs.”*

- Similarly, for the methylation analysis, the authors need to describe what is known and reference previous studies.

We have added additional smoking-methylation reference in the text in which our reported associations figure in the top list: *“AHRR, F2RL3, GP15 and RARA CpG sites that were previously identified as candidate smoking-related loci in whole blood^{25,26}.”*

- For the pQTL analysis, the authors need to describe why they think their results are novel.

As explained above, we used the term “novel” as compared to the genetics association already described in the MI cohort. We understand this is confusing and have performed a new analysis using the cytokine data referenced in the SomaLogic Database, which is to our knowledge the most exhaustive database of pQTLs for cytokines (Figure 2 for reviewers). We have included this data in the new version of the manuscript: *“The results (Table 1) are consistent with the trans-pQTLs identified from an independent cohort⁷, for common cytokines (IL6, IL10 and TNF α) and stimulations (LPS and C. albicans). In addition, we identified 22 pQTLs, that were not identified by this study, nor are present in the Somalogic database²⁰: CXCL5 (rs10013453, trans; rs10779330, trans), IFN γ (rs2564594, trans; rs3775291, trans; rs4833095, trans), IL12A (rs143060887, cis), IL12B (rs143060887, trans), IL17A (rs10004195, trans), IL1 β (rs10863358, trans; rs3764613, trans;*

rs3775291, trans), IL2 (rs1801274, trans; rs4833095, trans; rs6815814, trans), IL23A (rs10779330, trans; rs5743614, trans), IL6 (rs10034903, trans; rs11936050, trans; rs3775291, trans; rs72636686, trans), IL8 (rs4833095, trans), TNF α (trans; rs3775291, trans)._”

*Corresponding text has been added to the methods: “In order to test the novelty of our pQTL results, we studied the SomaLogic plasma protein pQTL database²⁰, for both cis- and trans-pQTLs listed in Table 1. This dataset allowed testing associations for CXCL5, IFN γ , IL1b, IL2, IL6, IL10 and IL12a. Significant associations were identified between the variants rs352045 (cis), rs2393969 (trans), rs10822168 (trans) and the protein CXCL5 (respective FDR adjusted $p = 3.02e-10$, $p=0.01$ and $p = 0.022$), between rs35345753 (cis), rs62449491 (cis) and IL6 (respective FDR adjusted $p = 4.17e-3$ and $p = 0.017$) and between rs3775291 (trans) and IL12A (FDR adjusted $p = 0.049$). In order to test associations for SNPs in linkage disequilibrium (LD) with the SNPs originally referenced in Table 1, we used a dataset of LD from the ensemble database with similar ancestries as the Milieu Intérieur cohort (1000GENOMES:phase_3:CEU: Utah residents with Northern and Western European ancestry). To be inclusive, SNPs with a $r^2 > 0.2$ were selected as associated alleles and underwent the same analysis as the one performed with the SNPs of reference. SNPs that came out as significant are those in LD with the SNP referenced in Table 1 that is significantly associated with the corresponding protein. In addition, we also screened eQTL results. We compared our pQTL results with the eQTLs reported in our previous work based on nanostring transcriptomic data for common cytokines (CSF2, IFN γ , IL1b, TNF α , IL2, IL6, IL8, IL10, IL12p70, IL13, IL17, IL23) and stimulations (*E. coli*, *C. albicans*, Influenza, BCG, and SEB)⁹, which identified 2 main loci: the TLR1/6/10 locus and the CR1 locus. Association of variants referenced in Table 1 were found in the GTEx consortium database for rs1518110 and IL10 (FDR adjusted $p = 4.3e-9$), for rs352045 (cis) and rs2564594 (cis) and CXCL5 (respective FDR adjusted $p = 9.2e-23$ and $4.1e-22$) in whole blood and for rs143060887 (cis) and IL12A (FDR adjusted $p = 0.000076$). Significant associations between rs352045 and CXCL5 and between rs1518110 and IL10 were also found in the eQTLgen catalogue.”*

Clarity and context: lucidity of abstract/summary, appropriateness of abstract, introduction and conclusions

As I mentioned before, I feel the authors need to describe better what is novel in the present work. Current descriptions of “new factors” and “new roles” are too vague.

We have modified the abstract to better highlight the novelty of our results and provide improved clarity on the novelty of our findings, changes from the original abstract are underlined: “Individuals widely differ in their immune responses, with age, sex and genetic factors playing major roles in this inherent variability. However, the variables that drive such differences in cytokine secretion, which is a crucial component of the host response to immune challenges, remain poorly defined. Here we investigated 136 variables and identified smoking, cytomegalovirus latent infection and body mass index as major contributors to cytokine response variability, with comparable effects in strength as age, sex and genetics. We find that smoking influences both innate and adaptive immune responses. Notably, its effect on innate responses is quickly lost after smoking cessation and is specifically associated with plasma levels of CEACAM6, whereas its effect on adaptive responses persists long”

after individuals quit smoking and is associated with epigenetic memory. This is supported by the association of the past smoking effect on cytokine responses with DNA methylation at specific signal transactivators and metabolism regulators. Our findings identify three novel variables associated with cytokine secretion variability and reveal new roles for smoking in the short and long-term regulation of immune responses. These results have potential clinical implications for the risk of developing infections, cancers or auto-immune diseases."

Referee #3 (Remarks to the Author):

This is an extremely detailed and well executed study of the effects of various parameters on the outcome of cigarette smoking cessation, particularly with regards to the immune system.

The positives of this study include the cohort of 1000 patients that are matched with regards to age, sex and ethnicity. This cohort, developed by the Milieu Intérieur (MI) project, has previously been interrogated for variations in immune homeostasis, with respect to age, sex, cytomegalovirus (CMV) latent infection and smoking.

Another positive is the ability to link with sociodemographic and clinical details etc. The current study is comprehensive, examining the production of 13 cytokines in response to 12 stimulants that trigger responses from innate and adaptive immune cells. In essence they reveal effects of BMI, age and smoking on cytokines from whole blood cultures. The effect depends on the stimulus. They then further show that smoking cessation restores innate immune differences but that some aspects of adaptive immunity do not revert (dependent on pack years) and that this may be due to epigenetic modifications.

There are no technical deficiencies in the reported outcomes. However, it is felt that the conclusions are not altogether surprising when considering prior publications in the same field.

We thank the reviewer for highlighting many positive aspects of our study including its extreme detail and well execution, as well as comprehensive nature and no technical deficiencies. We are happy that the reviewer considers that the conclusions are in line with expectations based on prior publications, however we disagree on his feeling that the conclusions are not surprising, in particular with the 2 references listed as examples. We show for the first time that smoking acts both on short term and long-term cytokine response regulation. We describe that the effect on innate responses is quickly reversible, while the effect on adaptive responses is persistent long after individuals stop smoking and is associated with specific epigenetic memory in long-lived T and B cells.

Mario Bauer et al. for example published in Epigenetics in 2016 that “Tobacco smoking differently influences cell types of the innate and adaptive immune system-indications from CpG site methylation”

Bauer *et al.* showed that the cg05575921 site within *AHRR* is hypomethylated in CD3+ T cells and granulocytes of active smokers, but they did not examine any impact on immune function such as cytokine responses. Nor did they examine any effect of past smoking, as we performed in our study, which revealed the most interesting and novel findings.

Giulia Piaggieschi et al published in Front Immunol in 2021 “Immune Trait Shifts in Association With Tobacco Smoking: A Study in Healthy Women”

Piaggieschi et al. reported associations between smoking and immune cell phenotypes. Again, they did not study immune function such as cytokine responses.

There are many other examples. It is likely that multiple factors acting in concert explain immune heterogeneity in health and disease, which is supported by the observation in the current study that smoking only explains between 4 and 9 percent of inter-individual variance.

We agree that multiple factors likely act in concert to explain immune heterogeneity, however we maintain that explaining 5-9% of variance due to a single environmental factor is a major contribution to our understanding of immune inter-individual variability. This is the equivalent level of variability explained by common cis-genetic associations, which are known to have clinical implications, and it is on average higher than the widely reported age and sex effects on immune response (see Piasecka et al., PNAS for example). The reviewer raises an important and interesting point about potential interactions. Along this line, we performed new analysis to test for SNP-smoking interactions as our cohort is well-powered for such testing. We have added a supplementary figure (Extended Data Fig. 8) and modified the text to highlight these additional new results: “*We also assessed potential interactions between the identified SNPs and smoking status. No significant associations were observed in E. coli, LPS, SEB and CD3+CD28 but BCG stimulation showed significant genetic-smoking interactions (BY adj-pval < 0.05) for rs61934597 and IL8, IL17, IL1b, IL6, and rs72636686 and IL8, IL1b and IL17 (Extended Data Fig. 8a). For the strongest interaction, between rs72636686 and smoking status for IL8 levels, smoking status can remove differences in response between individuals of different genotypes (Extended Data Fig. 8b), showing genotype effects can be modulated by the smoking status.*”

Extended Data Fig.8: Smoking status and SNPs interactions. **a**, Heatmaps showing $-\log_{10}(\text{BY adj.pval of LRT})$ of interactions between genetic variants listed in Table 1 and Smoking status for each induced cytokine considering age, sex and batchId as covariates. These are colored according to the color key on the side of each heatmap and stars are shown depending on the strength of association (BY adj.pval of LRT < 0.01 ; $(* < 0.05$; $** < 0.01$; $*** < 0.001$). **b**, Boxplot showing IL8 levels depending on the genotype for rs72636686 and the smoking status.

One specific point is that the authors should be careful categorising stimulants as “innate” or “adaptive” as lymphocytes for example can also express innate receptors and vice versa.

We thank the reviewer for this comment and have now clarified innate versus adaptive stimulations in the text : *“The stimulations are classified into 4 categories: microbial (Bacillus Calmette-Guérin (BCG), Escherichia coli (E. coli), lipopolysaccharide (LPS), and Candida albicans (C. albicans)) and viral (influenza and Polyinosinic-polycytidylic acid (PolyIC)) agents, that are predominantly recognized by receptors on innate immune cells; T-cell activators (Staphylococcus aureus Enterotoxin B superantigen (SEB) and anti-CD3+CD28 antibodies (CD3+CD28)), which induce adaptive immune responses; as well as cytokines (TNFa, IL1b, and IFNg). ”*

* Nature Research's authors website (<https://www.nature.com/authors>) contains information about and links to policies and resources.

This email has been sent through the Springer Nature Manuscript Tracking System NY-610A-SN&MTS

Confidentiality Statement:

This e-mail is confidential and subject to copyright. Any unauthorised use or disclosure of its contents is prohibited. If you have received this email in error please notify our Manuscript Tracking System Helpdesk team at <http://platformsupport.nature.com>.

Details of the confidentiality and pre-publicity policy may be found here
<http://www.nature.com/authors/policies/confidentiality.html>

Privacy Policy | Update Profile

DISCLAIMER: This e-mail is confidential and should not be used by anyone who is not the original intended recipient. If you have received this e-mail in error please inform the sender and delete it from your mailbox or any other storage mechanism. Springer Nature Limited does not accept liability for any statements made which are clearly the sender's own and not expressly made on behalf of Springer Nature Ltd or one of their agents. Please note that Springer Nature Limited and their agents and affiliates do not accept any responsibility for viruses or malware that may be contained in this e-mail or its attachments and it is your responsibility to scan the e-mail and attachments (if any).

Springer Nature Ltd. Registered office: The Campus, 4 Crinan Street, London, N1 9XW. Registered Number: 00785998 England.

Reviewer Reports on the First Revision:

Referees' comments:

Referee #1 (Remarks to the Author):

I thank the authors for a very clear and comprehensive rebuttal. They have addressed all of my comments clearly and the paper is much improved. I do not have any further substantive comments to add.

As a minor point on re-reading I noted a number of figures where the statistical tests used should be more explicitly stated in the figure legend. Several figures mention "significant associations (LRT adj.p.value < 0.01)" or similar; they should spell out that you have used Likelihood Ratio Tests more explicitly and without abbreviation. Figure 4 b-e show correlations with R values and p-statistics; the correlation method used (e.g. Pearson) should be stated here. Figure 5 needs a much clearer explanation of the statistics used within the figure legend.

Referee #2 (Remarks to the Author):

I appreciate the authors for providing a detailed clarification of their methods and discussing their results. In response to the authors' comments, I have only a few remaining points:

1. Genetic principal component analysis:

(a). I appreciate the authors' detailed response in justifying the statement regarding 'a homogenous genetic background.' As the authors pointed out, while the majority of the PCs are not associated with cytokine levels, a few do exhibit such associations (e.g., IL10 and PC1). Consequently, the claim of 'no significant associations with cytokine responses,' as included in the method section, was not accurate.

(b). The color bar in Fig. 3 for reviewers is not explained, I assume it is the effect size? Maybe it would be helpful to include them in the supplement.

(c). Additionally, it would be visually helpful to include a supplementary figure illustrating the top three genetic PCs of the individuals studied.

2. pQTL analysis

(a). I don't find the reason "further conditional analysis could be performed in order to describe these new reported associations but would take too much space in this article" convincing enough to warrant not performing the necessary analysis to establish novelty. If the author wanted to claim "new associations" in comparison to previous studies, then conducting a conditional analysis is warranted.

(b). Given the increasing number of pQTL studies, asserting novel pQTLs is challenging. Instead, I

recommend that the authors focus on “context/response-specific pQTL”, where a pQTL is only specific to a particular condition.

(c). The claim “The results are consistent with the trans-pQTLs identified in an independent cohort ...” (Line 231-233) lacks supporting supplementary material. A correlation plot of effect sizes reported in the two studies would support the claim.

(d). The summary statistics of the pQTL studies should be provided as an online resource.

(e). It would be easier for the readers if the authors report the actual SNP and LD in Line 243-244.

(f). Additionally, displaying the actual statistics (effect size and p-value) for the reported interaction (Line 260-263) would enhance clarity.

Referee #3 (Remarks to the Author):

UNAVAILABLE FOR RE-REVIEW

Referee #4 (Remarks to the Author):

This manuscript submitted by Violaine Saint-Andre et al “Smoking affects adaptive immunity with persistent effects”. Utilizes the Milieu Interior resource to examine how cytokine production upon innate or adaptive immune stimuli is influenced by heritability and environmental factors. This is a logical and important extension of the work of this group to help us better understand the variability in immune responses across individuals. The most notable finding is imparted by the title, showing that cigarette smoking influences the immune responses as measured by cytokine production, in response to both innate and adaptive stimuli, but that the effect of smoking on the adaptive immune system persists after smoking cessation, through epigenetic mechanisms. The prior reviewers have made important suggestions to enhance the manuscript and ensure the analysis is appropriate, and the authors responses and revisions are appropriate.

Overall, this work is clearly written, and includes an extensive amount of information that will be useful to future studies of the human immune response and variation across individuals. It would be ideal to validate the findings with another cohort and in more diverse populations as the authors note- but the size and scope of MI makes the findings important, and it is reasonable to publish without that additional data- with hopes others will provide that validation. Although the observation that smoking has important biological impact including on the immune response, this work with its depth and breadth makes an important contribution and includes novel or more complete insights into the process through which smoking influences the immune response.

Author Rebuttals to First Revision:

Thank you for the additional comments which we have addressed below in a point by point manner.

Referee #1 (Remarks to the Author):

I thank the authors for a very clear and comprehensive rebuttal. They have addressed all of my comments clearly and the paper is much improved. I do not have any further substantive comments to add.

Thank you for acknowledging our work and our effort to address all of your comments.

As a minor point on re-reading I noted a number of figures where the statistical tests used should be more explicitly stated in the figure legend. Several figures mention "significant associations (LRT adj.p.value < 0.01)" or similar; they should spell out that you have used Likelihood Ratio Tests more explicitly and without abbreviation. Figure 4 b-e show correlations with R values and p-statistics; the correlation method used (e.g. Pearson) should be stated here. Figure 5 needs a much clearer explanation of the statistics used within the figure legend.

As requested we have added details in the figure legends: we have spelled LRT as "Likelihood Ratio Tests", have added the correlation method used in the legend of Figure 4 ("R values and p-statistics of Pearson correlation are reported on each graph."), and have added details on the statistics used in the legend of Figure 5 ("The R² contributions averaged over orderings among regressors are represented on each plot."), as well as in the "Computation of variance explained" section of the methods ("The R² contribution averaged over orderings among regressors was computed using the "lmg" type in the calc.relimp function of the relaimpo R package.").

Referee #2 (Remarks to the Author):

I appreciate the authors for providing a detailed clarification of their methods and discussing their results. In response to the authors' comments, I have only a few remaining points:

Thank you for acknowledging our work. Below is a point by point response to your specific comments.

1. Genetic principal component analysis:

(a). I appreciate the authors' detailed response in justifying the statement regarding 'a homogenous genetic background.' As the authors pointed out, while the majority of the PCs are not associated with cytokine levels, a few do exhibit such associations (e.g., IL10 and PC1). Consequently, the claim of 'no significant associations with cytokine responses,' as included in the method section, was not accurate.

We apologize if this point lacked clarity. We have modified the corresponding sentence in the methods: *“Additionally, we formally checked how the genetic background of the donors could affect cytokine levels. Although PC1 had a significant association with IL10 (BY adj-pval < 0.05), we found that the first 20 principal components of the Principal Component Analysis (PCA) on the individual genotypes showed no significant associations with cytokine responses at the pvalue threshold (BY adj-pval < 0.01) we use throughout this study.”*

(b). The color bar in Fig. 3 for reviewers is not explained, I assume it is the effect size? Maybe it would be helpful to include them in the supplement.

We apologize for this lack of clarity. The color bar in Figure 3 for reviewers indicates $-\log_{10}(\text{BY adj-pvalue})$ of the Likelihood Ratio Tests, as for similar heatmaps reported in the main Figures. As imposed by *Nature* editorial guidelines, we are limited to 10 Supplementary Figures. Therefore, we did not consider the Figure 3 for reviewers to be important enough for inclusion in the manuscript. Indeed, this figure mostly shows no associations except for one of the tests, which we have explained in the methods section.

(c). Additionally, it would be visually helpful to include a supplementary figure illustrating the top three genetic PCs of the individuals studied.

Following the reviewer’s suggestion, we have now added a supplementary panel to Extended Data Figure 9 (shown below) to illustrate the genetic homogeneity of the 1,000 individuals of the *Milieu Intérieur* cohort. For comparison purposes, a PCA performed on 261,827 independent SNPs and 1,723 individuals, which include the 1,000 MI donors together with 723 individuals from a selection of 36 populations of North Africa, the Near East, western and northern Europe is shown, similarly to what was performed in Supplemental Figure 20b of Patin et al., 2018. PC1 versus PC2, PC1 versus PC3 and PC2 versus PC3 are displayed, as well as a barplot of the proportion of variance explained by the first 20 components of the PCA. The corresponding text has been added to the methods: *“To illustrate the homogeneity of the genetic structure of the 1,000 individuals of the MI cohort, a PCA was performed with EIGENSTRAT³⁵ on 261,827 independent SNPs and 1,723 individuals, which include the 1,000 MI donors together with 723 individuals from a selection of 36 populations originating from North Africa, the Near East, as well as western and northern Europe³⁶ is shown, similarly to what was previously performed⁸. PC1 versus PC2, PC1 versus PC3 and PC2 versus PC3 are displayed as well as a barplot of the variance explained by the first 20 components of the PCA (Extended Data Figure 9b).”* and Figure legend of Extended Data Figure 9b: *“PCA performed on 261,827 independent SNPs and 1,723 individuals, which include the 1,000 MI donors together with 723 individuals from a selection of 36 populations from North Africa, the Near East, as well as western and northern Europe. PC1 versus PC2 (top left), PC1 versus PC3 (bottom left) and PC2 versus PC3 (top right) are displayed as well as a barplot (bottom right) of the variance explained by the first 20 components of the PCA.”*

Extended Data Figure 9b

2. pQTL analysis

(a). I don't find the reason "further conditional analysis could be performed in order to describe these new reported associations but would take too much space in this article" convincing enough to warrant not performing the necessary analysis to establish novelty. If the author wanted to claim "new associations" in comparison to previous studies, then conducting a conditional analysis is warranted.

(b). Given the increasing number of pQTL studies, asserting novel pQTLs is challenging. Instead, I recommend that the authors focus on "context/response-specific pQTL", where a pQTL is only specific to a particular condition.

We have coupled our responses to the two points above. The Somalogic and Olink databases are the main resources of plasma pQTLs, which have identified pQTLs for some of our tested cytokines at steady state (Sun et al., 2018; Ferkingstad et al., 2021; Pietzner et al., 2021; Gudjonsson et al., 2022; Koprulu et al., 2023). However, to our knowledge only one study, the 500FG study, tested for response pQTLs for some of our tested cytokines using PBMCs (Li et al., 2016). Among the common tested cytokine-stimulation pairs, both studies identify pQTLs for IL6 and IL1b in PolyIC, TNFa in *C. albicans* and IL1b in LPS. Interestingly, the pQTL (rs3775291) we identify for IL1b and IL6 in PolyIC is located in the TLR3 exon locus while the pQTLs reported by the 500FG study for these cytokines (rs28393318 for IL1b and rs6831581 for IL6) are located in the TLR1/6/10 locus. Following the reviewer’s question, we have performed conditional analysis for rs28393318 (rs6831581 was not tested in our dataset) and rs3775291, by passing rs28393318 as a covariate in our pQTL identification with MatrixEQTL. We show that the association we report between IL1b and rs3775291 is maintained, indicating that it is independent of the one reported in the 500FG study (see new SI Table S4 containing the summary statistics for both analyses, which top lines are reported in the Table below). As we detail in the manuscript, this SNP is of particular interest as it is associated with age-related macular degeneration and resistance to viral infections. We apologize if this was not clear enough in our original manuscript but we only report potential novelty for response pQTLs, as most of these were not tested elsewhere. In addition, the reviewer is correct that given the increasing number of pQTL studies, asserting novel pQTLs is challenging, which is why we used the terms “potential new” and “to our knowledge”. We have now amended the text to include these additional details: *“We tested a total of 5,699,237 high quality imputed Single-Nucleotide Polymorphisms (SNPs) for associations with the cytokines induced in each stimulation, adjusting for age, sex, technical variables and major immune cell population counts (SI Table 3) and report 44 response pQTLs (Table 1). The Somalogic and Olink databases are the main resources of plasma pQTLs, which have identified pQTLs for some of our tested cytokines at steady state^{19–23}. However, to our knowledge only one study, the 500FG study, tested for response pQTLs for some of our tested cytokines using PBMCs²⁴. Among the common tested cytokine-stimulation pairs, both studies identify pQTLs for IL6 and IL1b in PolyIC, TNFa in *C. albicans* and IL1b in LPS. Interestingly, the pQTL (rs3775291) we identify for IL1b and IL6 in PolyIC is located in the TLR3 exon locus while the pQTLs reported by the 500FG study for these cytokines (rs28393318 for IL1b and rs6831581 for IL6) are located in the TLR1/6/10 locus. We have performed conditional analysis between the two locus , by passing rs28393318 as a covariate in our pQTL identification, and show that the association we report between IL1b and rs3775291 is maintained, indicating that it is independent of the one reported in the 500FG study (SI Table S4). This is consistent with PolyIC signaling through the TLR3 pathway. Among the potential new trans pQTLs we identified, 3 are for CXCL5.”*

without rs28393318 as additional covariate						
snps	gene	statistic	pvalue	FDR	beta	R2
rs3775291	ENSG000001	10.48793411	1.76244e-24	8.02872e-17	0.50882287	0.09963504
with rs28393318 as additional covariate						
snps	gene	statistic	pvalue	FDR	beta	R2
rs3775291	ENSG000001	10.63269437	4.35504e-25	1.98392e-17	0.52489353	0.10175398

(c). The claim “The results are consistent with the trans-pQTLs identified in an independent cohort ...” (Line 231-233) lacks supporting supplementary material. A correlation plot of effect sizes reported in the two studies would support the claim.

As the effect sizes and p-values were not computed using the same covariates in the models and the reported SNPs are not the same for the two studies (and sometimes not present in our microarrays), comparing these values is very challenging. Instead, we assessed whether we identified pQTLs for the same stimulation-cytokine pairs, and if these came from the same locus. Interestingly we identified a different pQTL for IL6 and IL1 β in Poly IC stimulation which makes biological sense as PolyIC stimulation induces immune response mediated through the TLR3 pathway. We have made this clearer in the text: *“We tested a total of 5,699,237 high quality imputed Single-Nucleotide Polymorphisms (SNPs) for associations with the cytokines induced in each stimulation, adjusting for age, sex, technical variables and major immune cell population counts (SI Table 3) and report 44 response pQTLs (Table 1). The Somalogic and Olink databases are the main resources of plasma pQTLs, which have identified pQTLs for some of our tested cytokines at steady state^{19–23}. However, to our knowledge only one study, the 500FG study, tested for response pQTLs for some of our tested cytokines in PBMCs²⁴. Among the common tested cytokine-stimulation pairs, both studies identify pQTLs for IL6 and IL1 β in PolyIC, TNF α in *C. albicans* and IL1 β in LPS. Interestingly, the pQTL (rs3775291) we identify for IL1 β and IL6 in PolyIC is located in the TLR3 exon locus while the pQTLs reported by the 500FG study for these cytokines (rs28393318 for IL1 β and rs6831581 for IL6) are located in the TLR1/6/10 locus. We performed conditional analysis between the two loci, by passing rs28393318 as a covariate in our pQTL identification, and show that the association we report between IL1 β and rs3775291 is maintained, indicating that it is independent of the one reported in the 500FG study (SI Table S4). This is consistent with PolyIC signaling through the TLR3 pathway. Among the potential new trans pQTLs we identified, 3 are for CXCL5.”.*

(d). The summary statistics of the pQTL studies should be provided as an online resource.

As requested we have now compiled the summary statistics of the pQTLs as a new supplementary table (SI Table 3).

(e). It would be easier for the readers if the authors report the actual SNP and LD in Line 243-244.

We thank the reviewer for raising this point. The SNP in question is rs10013453 and its LD value with rs4833095 is $r^2=0.69$. As this LD value is indeed pretty high, we have tested for colocalization of the two pQTL signals, using SuSiE (Wang et al., 2020) (susie_rss function of the susieR R package) and coloc (coloc.susie function from the coloc R package). Our results argue in favor of non-independent effects (posterior probability that both traits are linked to the same SNP (PPH4) > 0.8), so we have removed this sentence in the text to avoid confusion and corrected this in Table1. We also have made minor changes and reordered Table 1 in alphabetical order of cytokines.

(f). Additionally, displaying the actual statistics (effect size and p-value) for the reported interaction (Line 260-263) would enhance clarity.

Following the reviewer's request, the effect size and adj p-value for rs72636686 and smoking interaction have now been added to the sentence and included to supplementary figures: *"For the strongest interaction (effect size= 1.58 [1.42 - 1.75], BY adj-pvalue=3.8e-13) (Extended Data Figure 8b), between rs72636686 and smoking status for IL8 levels, smoking status can remove differences in response between individuals of different genotypes (Extended Data Fig. 8c), showing genotype effects can be modulated by the smoking status."* and a figure panel representing this effect size has been added to Extended Data Figure 8, with its corresponding legend: *"b, Effect size plot for the interaction of the SNP rs72636686 with the Smoking status, corrected for age, sex and batchId, on induced cytokines in the BCG stimulation condition. Significant effect sizes (p-val < 0.01) are in black, others are in grey. Those that also have BY adj.pval of the Likelihood Ratio Test < 0.01 are labelled with a red star."*

Extended Data Figure 8b

Referee #3 (Remarks to the Author):
UNAVAILABLE FOR RE-REVIEW

Referee #4 (Remarks to the Author):

This manuscript submitted by Violaine Saint-Andre et al "Smoking affects adaptive immunity with persistent effects". Utilizes the Milieu Interior resource to examine how cytokine production upon innate or adaptive immune stimuli is influenced by heritability and environmental factors. This is a logical and important extension of the work of this group to help us better understand the variability in immune responses across individuals. The most notable finding is imparted by the title, showing that cigarette smoking influences the immune responses as measured by cytokine production, in response to both innate and adaptive stimuli, but that the effect of smoking on the adaptive immune system persists after smoking cessation, through epigenetic mechanisms. The prior reviewers have made important suggestions to enhance the manuscript and ensure the analysis is

appropriate, and the authors responses and revisions are appropriate.

Overall, this work is clearly written, and includes an extensive amount of information that will be useful to future studies of the human immune response and variation across individuals. It would be ideal to validate the findings with another cohort and in more diverse populations as the authors note- but the size and scope of MI makes the findings important, and it is reasonable to publish without that additional data- with hopes others will provide that validation. Although the observation that smoking has important biological impact including on the immune response, this work with its depth and breadth makes an important contribution and includes novel or more complete insights into the process through which smoking influences the immune response.

Thank you for the appreciation of our work.

This email has been sent through the Springer Nature Manuscript Tracking System NY-610A-SN&MTS

Confidentiality Statement:

This e-mail is confidential and subject to copyright. Any unauthorised use or disclosure of its contents is prohibited. If you have received this email in error please notify our Manuscript Tracking System Helpdesk team at <http://platformsupport.nature.com>.

Details of the confidentiality and pre-publicity policy may be found here

<http://www.nature.com/authors/policies/confidentiality.html>

Privacy Policy | Update Profile

DISCLAIMER: This e-mail is confidential and should not be used by anyone who is not the original intended recipient. If you have received this e-mail in error please inform the sender and delete it from your mailbox or any other storage mechanism. Springer Nature Limited does not accept liability for any statements made which are clearly the sender's own and not expressly made on behalf of Springer Nature Ltd or one of their agents. Please note that Springer Nature Limited and their agents and affiliates do not accept any responsibility for viruses or malware that may be contained in this e-mail or its attachments and it is your responsibility to scan the e-mail and attachments (if any).

Springer Nature Ltd. Registered office: The Campus, 4 Crinan Street, London, N1 9XW. Registered Number: 00785998 England.

Reviewer Reports on the Second Revision:

Referees' comments:

Referee #2 (Remarks to the Author):

The authors have addressed all my questions now.